# Revealing the meteorological drivers of the September 2015 severe dust event in the Eastern Mediterranean

Philipp Gasch[1], Daniel Rieger[1], Carolin Walter[1], Pavel Khain[2], Yoav Levi[2], Peter Knippertz[1], and Bernhard Vogel[1]

[1]Karlsruhe Institute of Technology, Institute of Meteorology and Climate Research, Karlsruhe, Germany
[2]Israel Meteorological Service, Bet Dagan, Israel

*Correspondence to:* Philipp Gasch (philipp.gasch@kit.edu)

**Abstract.** In September 2015 one of the severest and most unusual dust events on record occurred in the Eastern Mediterranean. Surprisingly, operational dust transport models were unable to forecast the event. This study details the reasons for this failure and presents simulations of the event at convection permitting resolution using the modelling system ICON-ART. The results allow for an in-depth analysis of the influence of the synoptic situation, the complex interaction of multiple driving atmospheric systems and the mineral dust radiative effect on the dust event. A comparison of the results with observations reveals the quality of the simulation results with respect to structure and timing of the dust transport. The forecast of the dust event is improved decisively. The event is triggered by the unusually early occurrence of an active Red Sea trough situation with an easterly axis over Mesopotamia. The connected sustained organized meso-scale convection produces multiple cold-pool outflows responsible for intense dust emissions. Complexity is added by the interaction with an intense heat low, the inland penetrating Eastern Mediterranean sea-breeze and the widespread occurrence of super-critical flow conditions and subsequent hydraulic jumps in the vicinity of the Dead Sea Rift Valley. The newly implemented mineral dust radiation interaction leads to systematically more intense and faster propagating cold-pool outflows.

## 1 Introduction

Mineral dust aerosol plays an important role for the environment. The transport of mineral dust within the atmosphere has an effect on physical processes, chemical composition and biological systems on various temporal and spatial scales (Carslaw et al., 2010; Shao et al., 2011a; Boucher et al., 2013). Therefore, a need exists to correctly represent mineral dust and its effects in atmospheric models.

In the Eastern Mediterranean the impact of mineral dust on the environment, human population and traffic is important due to the unique environmental setting and great population density. Dust events in the Eastern Mediterranean (EM) are usually associated with strong south-westerly and southern flows in the region, although events have been reported under easterly flow conditions (Dayan et al., 1991; Levi and Rosenfeld, 1996). As a result, the most important remote dust source regions for the EM are situated in northeastern Africa and the southern Arabian peninsula (Ganor, 1991; Kubilay et al., 2000). According to a subjective classification conducted by Dayan et al. (2008), the synoptic scale systems associated with dust transport towards

the EM are Cyprus Lows (60%), Sharav cyclones (12%) and Red Sea troughs (12%). Consequently, the number of dust events in the EM is correlated with cyclone activity in the region (Kishcha et al., 2016, and references therein). Long-range mineral dust transport towards the EM peaks during the transitional seasons in spring and autumn (Offer and Goossens, 2001), when low pressure systems and their associated fronts occur most frequently in the region (Singer et al., 2003, and references therein;

Dayan et al., 2008). Resulting from the climatology of dust events in the EM, many studies investigate and simulate dust events in connection with low pressure systems and long-range transport of mineral dust from the Sahara towards the EM (Vogel et al., 2006; Spyrou et al., 2013; Rémy et al., 2015; Kishcha et al., 2016).

In September 2015 an exceptional dust event occurred in the EM. The impact of the event on the EM region was severe, with five people reported to have died, hundreds hospitalized and daily life as well as traffic in the region disrupted (NASA Earth

Observatory, 2015; The Weather Channel, 2015; Times Of Israel, 2015). The event was remarkable with respect to magnitude, timing, duration and dust transport direction. In Israel, dust surface concentrations were more than 100 times the normal, exceeding 5000 $\mu$g m$^{-3}$ at some stations. The Israeli Meteorological Service stated that it was the first time in 75 years that a dust storm reached Israel in early September and lasted beyond one day (Alpert et al., 2016). In an analysis of measurements from Cyprus, Mamouri et al. (2016) also report dust surface concentrations close to 8000 $\mu$g m$^{-3}$, with maximum aerosol

optical thickness (AOD) values above 5 retrieved by the Moderate Resolution Imaging Spectroradiometer (MODIS). Based on a climatological comparison of the observed AOD values Mamouri et al. (2016) classify the event as 'record-breaking'. They report an observed multi-layered dust plume structure, indicating a complex event evolution with multiple dust emission sequences.

Satellite images for the visible part of the electromagnetic spectrum (VIS) from MODIS illustrate the evolution of the dust

storm (Fig. 1). Starting from 06 September 2015 high dust concentrations were observed over Syria. On 07 September the full Mesopotamia region and the northern part of the EM was covered by a thick dust layer. Large amounts of dust were transported into the southern EM region during the following night, consequently covering the complete EM on 08 September 2015. The dust plume remained detectable in the region over the course of the next seven days (Alpert et al., 2016; Mamouri et al., 2016). As discussed above, the propagation direction of the record dust storm into the Eastern Mediterranean from the east is very

unusual and so is the duration of the event.

Adding to the extraordinariness, operational global dust transport models were unable to forecast the event as also noted by Mamouri et al. (2016). All predictions provided through the World Meteorological Organization's Sand and Dust Storm Warning Advisory and Assessment System (SDS-WAS, http://sds-was.aemet.es) initialised at 12 UTC 07 September failed to simulate significant dust concentrations in the EM region for 12 UTC 08 September. The simulated values of the dust optical

depth in the EM are between $0.1 - 0.4$ in the multi-model mean with a standard deviation of $0.1 - 0.2$. The forecast failure is highly problematic due to the severe impact of the event.

From EUMETSAT (European Organisation for the Exploitation of Meteorological Satellites) SEVIRI (Spinning Enhanced Visual and Infrared Imager) satellite observations the development of organized meso-scale convective systems (MCS) which produced cold-pool outflows (CPO) over Mesopotamia is detectable. The CPOs and their interaction with a heat low have been

suggested as important drivers for the observed dust emissions by Kerkmann et al. (2015). Solomos et al. (2017) model the

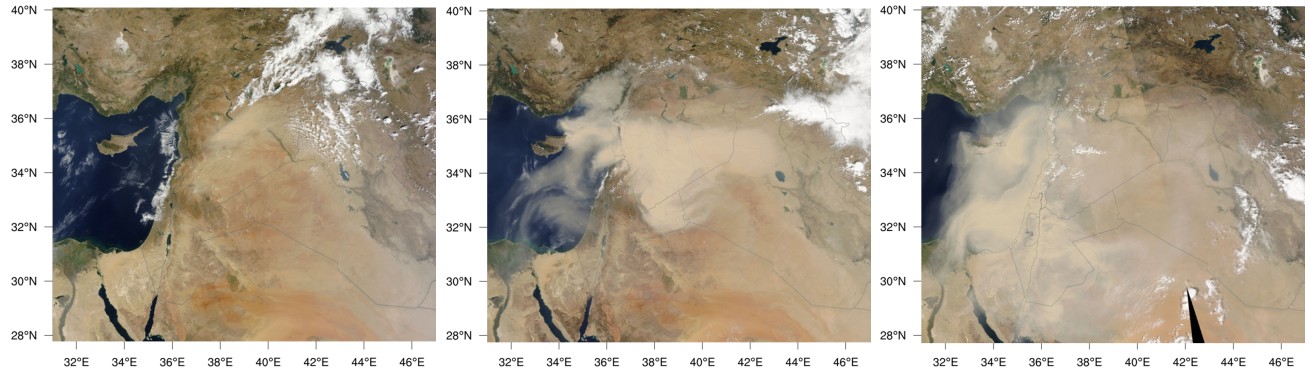

**Figure 1.** MODIS VIS satellite images of the EM region, from left to right at 08.15 UTC 06 September, 10.35 UTC 07 September and 11.18 UTC 08 September 2015 (NASA Worldview, 2016).

event at convection permitting resolution, however, the spatial extent of their convection permitting domain does not cover the full MCS region. Consequently, their model fails to reproduce the observed CPO outflow structures and connected dust plumes realistically (see discussion in Sec. 3.2 and 3.3).

A number of studies have shown that the inability to represent organized meso-scale convection and the related CPOs in
models with parametrized convection can lead to a substantial underestimation of dust emissions. Analysing simulations for summertime West Africa, Marsham et al. (2011) show that only convection permitting models are able to resolve an afternoon peak in a parameter termed uplift potential, which is closely related to dust emission. Heinold et al. (2013) model off-line dust emission and estimate the contribution of different meteorological systems using large-domain convection permitting simulations during a 40-day period also for summertime West Africa. Corroborating Marsham et al. (2011), they find that
approximately 40% of dust emissions can be linked to CPOs, highlighting the need for convection permitting resolution modelling. In another in-depth study of CPOs, Pantillon et al. (2016) analyse year-long convection permitting simulations in the Sahara region. They conclude that in this region the contribution of convective CPOs to dust uplift potential is in the order of one fifth of the annual budget, with substantially higher proportions up to one third over the summer months. In summary, CPOs have been identified as important systems contributing to dust emission (Knippertz et al., 2007; Marsham et al., 2011; Heinold
et al., 2013; Pantillon et al., 2016) and their occurrence has been documented for all major dust source regions worldwide (Knippertz, 2014).

Therefore, the existence of CPOs detectable in satellite observations is suspected as the main reason for the failure of operational dust transport models, as all of them operate at resolutions with fully parametrized convection. In order to overcome this problem the ICON-ART (ICOsahedral Nonhydrostatic - Aerosol and Reactive Trace gases) global modelling system (Zängl
et al., 2015; Rieger et al., 2015) is used in this study. It is capable of local grid refinements, in this study the finest nest has a convection permitting grid spacing of 2.5 km. The ART extension (Rieger et al., 2015) allows for on-line simulation of dust processes in ICON.

The mineral dust radiation interaction has been implemented in ICON-ART as a part of this study, as it has been shown to be of great importance for atmospheric processes (Pérez et al., 2006; Heinold et al., 2008; Bangert et al., 2012; Rémy et al., 2015). The mineral dust radiative effect is calculated on-line based on Mie-calculations and feeds back on atmospheric parameters, which in turn can influence mineral dust processes.

Combining the above topics, the influence of the radiative effect of dust contained in CPOs is investigated as a part of this study. The modification of the atmospheric radiation budget by CPOs can influence their evolution and lifetime. A first mechanism is intensively studied by Redl et al. (2016), showing that surface temperatures in CPOs can be higher than in the surrounding air masses at night. This counter-intuitive behaviour is due to increased cloud coverage inside the CPO and due to the dynamical breakup of the stable night-time inversion in the surface layers by increased turbulent mixing. The result is a downward transport of energy and reduced cooling of the surface, which in turn radiates this energy into space. The overall effect is a loss of energy from the lower boundary layer in the initial stages of cold-pool development. Redl et al. (2016) also investigate a second effect of higher humidity within the cold air-mass leading to increased downwelling longwave radiation in the order of $5 \mathrm{~W} \mathrm{~m}^{-2}$ and thereby warming of the lowest layer. They state that this effect becomes increasingly important in the later stages of CPO development after the dynamical effects diminish. The reduced stratification of the CPO in the lowest layers can result in increased vertical mixing, turbulence and drag, leading to a faster decay. A third effect, which is not included in the model of Redl et al. (2016), is the emission of mineral dust due to the high wind speeds and its subsequent interaction with radiation. This is expected to reduce incoming shortwave radiation during daytime and increase downwelling atmospheric longwave radiation during night-time. Mineral dust can thereby feedback on boundary layer dynamics, which in turn can alter dust processes (Heinold et al., 2008; Rémy et al., 2015). The missing radiative effect of mineral dust is also a limitation in the above discussed study of Heinold et al. (2013). Kalenderski and Stenchikov (2016) include and investigate the mineral dust radiative effect in their convection permitting simulation of a CPO in the Red Sea region. Extensively comparing their model results to observations, they find generally good agreement in modelled dust plume structure. They report significant reductions up to $301.4 \mathrm{~W} \mathrm{~m}^{-2}$ in incoming solar radiation and longwave radiation increases up to $9.0 \mathrm{~W} \mathrm{~m}^{-2}$ at the surface due to mineral dust. However, Kalenderski and Stenchikov (2016) do not systematically investigate the mineral dust radiative feedback on the CPO structure.

In this paper, we present results from convection permitting simulations of the September 2015 severe dust event in the EM with ICON-ART including the mineral dust radiative effect. Consequently, the research questions addressed are as follows: (1) Is the forecast of the dust event improved by running convection permitting simulations? (2) How does the synoptic situation relate to its exceptional character? (3) What are the meteorological drivers responsible for pick-up and long-range transport of mineral dust during this event? (4) How does the mineral dust radiative effect influence the dust event in general and the evolution of the CPOs in particular?

## 2 Model description

ICON is a non-hydrostatic modelling system developed jointly by the German Weather Service (DWD) and the Max Planck Institute for Meteorology (Zängl et al., 2015). It solves the full three-dimensional non-hydrostatic and compressible Navier-Stokes equations for all domains. Thereby, ICON can serve as a unified global numerical weather prediction model and climate modelling system, enabling seamless prediction from the global to local scale with a unified set of model physics. Its major advantages over previous model generations used at DWD and especially important for atmospheric tracer studies are the exact local mass conservation achieved by solving a prognostic equation for density, and the mass-consistent tracer transport achieved by transporting time-averaged mass fluxes computed from the dynamical core and diagnostic reintegration of the mass continuity equation. ICON allows for flexible local grid refinements (nests) with two-way interactions between the respective grids. Furthermore, it features a better scalability on massively parallel computer architectures. Since January 2015 ICON is used for operational weather forecasting at DWD. The ART module is an extension of ICON developed at the Institute of Meteorology and Climate Research at the Karlsruhe Institute of Technology. An overview of the module is given by Rieger et al. (2015). ART is capable of simulating a variety of aerosol species, e.g. volcanic ash, sea salt and radioactive substances. In addition, atmospheric chemistry processes, e.g. two species of very short-lived bromocarbons, a linearised ozone chemistry and photolysis are also available in ART. For the tracer transport simulations the modelling capabilities of ICON are of crucial importance because the same physical parametrization packages can be used from a global to regional scale. Thereby, inconsistencies in tracer concentrations arising from differences in tracer advection and physical parametrizations between the driving model and the high-resolution model can be avoided.

The size distribution of mineral dust is represented by three modes in ART. For each mode the integral values of specific number and mass are the prognostic variables. The distribution of specific number and mass with particle size during transport is described using log-normal distributions for each mode with the diagnostic median diameter of the mass distribution and constant geometric standard deviation as parameters (Mode A, d = 1.5 $\mu$m, $\sigma$ = 1.7; Mode B, d = 6.7 $\mu$m, $\sigma$ = 1.6; Mode C, d = 14.2 $\mu$m, $\sigma$ = 1.5). The processes which affect mineral dust number and/or mass concentrations in ART are gravitational settling (sedimentation), deposition due to turbulent diffusion and wet deposition due to washout. The scheme used for emission of mineral dust in ART is described in Rieger (2016). It is based on an emission scheme introduced by Vogel et al. (2006). Compared to the original version three improvements were implemented, these are (1) the global availability of soil properties (size distribution, residual soil moisture), (2) accounting for the soil dispersion state, and (3) a tile approach used to account for soil type heterogeneity at coarse resolutions. For the equations used the reader is referred to the work by Rieger (2016). The scheme parametrizes the threshold friction velocity above which dust emission can occur according to Shao and Lu (2000). They base their description on a physical balance between aerodynamic drag and lift causing upward directed forces and cohesion and gravity causing downward directed forces on particles. The effect of surface roughness and soil moisture are accounted for through parametrizations by Raupach et al. (1993) and Fécan et al. (1999) respectively, which modify the threshold friction velocity. The regional distribution of soil types used for dust emission in ART is taken from the Harmonized World Soil Database (HWSD) with a resolution of 30 arc seconds (Nachtergaele and Batjes, 2012). The fraction of erodible

soil is determined assuming that certain land use classes from the GlobCover2000 dataset (Arino et al., 2008) contribute to mineral dust emission whereas others do not, with snow generally prohibiting emissions. The land use classes which can contribute to emission are regions with sparse vegetation, bare areas, closed to open grass- and shrub-lands, furthermore mosaic forest/grassland and shrub-land. In order to retain the high spatial resolution of the dataset for ART a tile approach is used for calculating dust emissions. The tile approach calculates the overall emission in every grid box as a weighted average of the emissions from different soil types based on their fractional coverage of the grid box. Determination of land use classes is difficult in the Levantine region. It has undergone drastic change in recent years due to on-going conflict (Gleick, 2014), problems in transboundary water management (Voss et al., 2013) and a drought period (Notaro et al., 2015). Therefore, widespread, previously cultivated areas exist which are now available to erosion and known to be efficient dust sources (Solomos et al., 2017). The physical parametrization for mineral dust emission in ICON-ART relies on soil type and land cover information from external datasets as stated above. Hence, the recent land cover changes as described above are not reflected in the datasets used by ICON-ART. Therefore, this can result in an underestimation of dust emission.

## 2.1 Mineral dust radiative effect

As a part of this study the on-line dust radiative effect has been implemented in ICON-ART. It is now possible to include the radiative effect of the current, local dust concentration from ART at every grid point and time step in ICON instead of the previously used dust climatology. Through its feedback on radiative fluxes the dust influences atmospheric state, thereby providing a feedback loop back to dust processes again (Tegen et al., 2006; Heinold et al., 2008; Shao et al., 2011a). The implementation was done for the standard radiation scheme utilized by ICON which is the Rapid Radiative Transfer Model (RRTM) described by Mlawer et al. (1997). The ART dust radiation routine is called at every time step at which the RRTM is called by ICON.

Without ART, ICON uses a climatological distribution of aerosols (e.g. mineral dust, sea salt, stratospheric aerosol) to include their radiative effect. When using ART, any of these aerosol species can be calculated on-line and therefore its radiative effect can be included with much better accuracy. For aerosol species not simulated by ART the climatological values are still used and taken from ICON. Therefore, the radiative transfer parameters provided by ART to the RRTM are combined values from the local ART aerosol concentration plus the ICON climatology, which is used only for the aerosol species not simulated. For example, in our study we simulated mineral dust using ART, and therefore can include the on-line mineral dust radiative feedback. For the sea salt and stratospheric aerosol radiative effect, however, the climatological values from ICON are used.

The radiative transfers parameters needed consist of the optical depth, single scattering albedo and asymmetry parameter. In order to obtain the on-line mineral dust radiative feedback, the local radiative transfer parameters are calculated using the dust optical properties and the local dust mass concentration at every grid-point and for every level as detailed in Stanelle et al. (2010). The radiative transfer parameters are calculated in ART and provided to the RRTM, where they feedback on the atmospheric state in ICON.

For consideration of the on-line dust radiative effect the dust optical properties need to be determined. The mineral dust optical properties are computed with the help of Mie calculations using the complex refractive index of mineral dust (Bohren

and Huffman, 1983; Petty, 2006; Wagner et al., 2012). This study therefore assumes sphericity as well as a spatially invariant mineral composition of the mineral dust particles. Although this assumption is generally not fulfilled for single mineral dust particles (Otto et al., 2009; Kahnert et al., 2007), it introduces negligible errors for a population of randomly oriented non-spherical particles if only albedo and flux related quantities are calculated (Mishchenko et al., 1995, 1997). Due to the random orientation a collection of particles scatters light similar to a spherical particle because the individual differences disappear in the angular integration (Tegen and Lacis, 1996). Therefore Mie calculations are able to provide a good representation of scattering even for non-spherical particles.

The spatially invariant mineral composition of dust in ICON-ART means we assume similarity to Saharan dust everywhere. Studies have shown that mineral dust optical properties can depend on the source region (Petzold et al., 2009), which presents a great uncertainty for the radiative forcing as discussed in Myhre and Stordal (2001). For our region of interest, Nisantzi et al. (2015) find differences in the dust particle lidar ratios in a comparison of dust from the Middle East and the Sahara. Two problems prevent a more detailed description of the mineral dust optical properties for our study. First, there is a lack of observations of the refractive index for our dust source region and the variance within source regions can be considerable (Petzold et al., 2009). Second, to the best of our knowledge a dataset of the earth's crust mineralogical composition for our region is missing so far, making a more detailed availability of refractive indices futile. However, the influence of differences in the refractive indices is small compared to the influence of a varying size distribution (Myhre and Stordal, 2001) and this latter effect is represented in ICON-ART.

The values of the refractive index used to conduct the Mie calculations are the same ones used by Stanelle et al. (2010) for COSMO-ART and therefore not detailed further.

For the Mie calculations a code developed by Bond et al. (2006) was used, this in turn utilizes a subset developed by Mätzler (2002) for calculation of the Mie scattering coefficients and truncation of the series. The code was adapted to allow for processing of multiple wavelengths and averaging to the RRTM wavebands in a post-processing step.

The mineral dust optical properties are calculated for three modes and 30 RRTM radiation wavebands, respectively. Results of Mie calculations for the ART mineral dust modes are shown in Fig. 2 and are comparable to those published by other studies (Tegen and Lacis, 1996; Helmert et al., 2007; Rémy et al., 2015). Although the authors use slightly different size distributions and refractive index properties the results are very similar to the ones presented here.

The optical properties of mineral dust are highly dependent on its particle size distribution represented through three log-normal modes in ART. From the six prognostic variables in ART, the specific dust mass and number concentration for each of the three modes, a median diameter can be diagnosed for each mode. Due to different processes such as gravitational settling acting differently on the specific dust mass and number concentrations in ART, the diagnostic median diameter of each mode changes during transport (the standard deviation of each mode is kept constant). In a physical sense, the most important effect is large particles settling out faster due to sedimentation, which results in mostly smaller particles being transported to distant regions within each mode. Therefore, the median diameter of each mode is expected to decrease during transport. Consequently, a new polynomial parametrization of the optical properties per waveband was introduced to account for the change in median diameter during the transport processes. In a post-processing step a third order polynomial is fitted to the results of multiple Mie

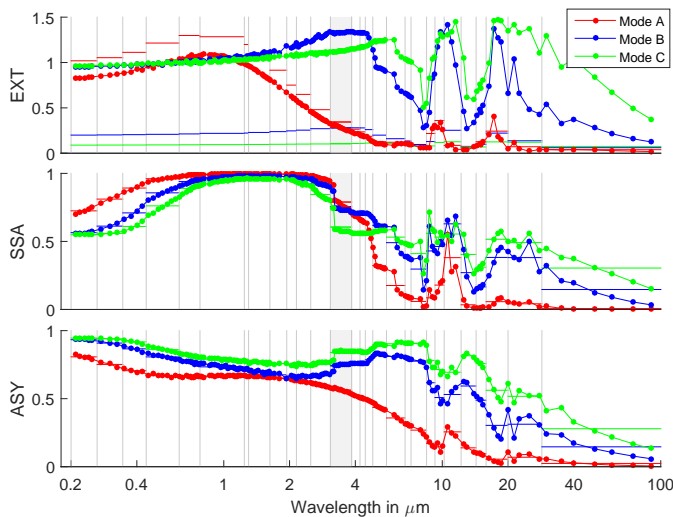

**Figure 2.** Results from Mie calculations as used by ICON-ART. Points are normalized with respect to value at 550nm, horizontal lines show absolute values. Shown are from top to bottom: Extinction coefficient (EXT) in $m^2\,g^{-1}$, single scattering albedo (SSA) and asymmetry parameter (ASY) for all modes. The borders of the RRTM radiation scheme wavebands are adumbrated as grey lines in the background. The filled grey band represents the waveband present both in the longwave and shortwave part of the RRTM.

calculations for median diameters between $0.25-1.25$ times the initial median diameter of the count number distribution. This is done for every mode and every RRTM waveband respectively. The polynomial fit parameters are initialized in ICON-ART and used for determination of optical properties at every grid point with the diagnosed median diameter being the independent variable.

The specifics of the mineral dust radiative effect implementation are available upon request and detailed in Gasch (2016).

## 2.2    Model set-up

In this study ICON-ART is run in a set-up with one global domain and four nests, with two-way feedback for the meteorological parameters enabled for all domains. Each domain presents a stand-alone model run which obtains its lateral boundary conditions from the coarser domain. For the global domain an R2B6 grid is used, this corresponds to an effective grid spacing

of 40 km (Zängl et al., 2015). For every nest, the grid point distance is halved, thereby ending at R2B10 with an effective grid spacing of 2.5 km. The finest domain is circular shaped and centred at $35°\,N, 40°\,E$ with a radius of approximately 1000 km, thereby containing 507100 grid points. In order to realistically represent organized moist, deep convection a convection permitting model resolution is required. A grid spacing of 2.5 km is generally assumed to be sufficient to permit the development of convection in a non-hydrostatic model. Therefore the convection parametrization, including the parametrization for shallow

convection, is switched off for the finest grid. In our setup for the finest resolution, the advection/fast physics time step is 18 seconds with a sub-stepping of the dynamics at 4.5 seconds. RRTM is called every 288 seconds for the finest resolution.

On the global domain the model consists of 90 levels extending up to the mesosphere, with the lowest level being at 20 m and the highest level at 75 km. For the nests the simulated atmosphere extends into the stratosphere up to 22.5 km containing 60 vertical levels.

For the cloud micro-physical processes a the two-moment cloud scheme is used (Seifert and Beheng, 2006), as this was found to lead to more realistic features of the meso-scale organized convection. The two-moment scheme utilizes a parametrization developed by Seifert and Beheng (2001) which predicts number and mass concentrations for six different hydro-meteor species. These are cloud droplets, rain drops, cloud ice, snow, graupel and hail. For this parametrization an extension was developed by Rieger (2016), which includes the aerosol effect on cloud formation through using the current, local aerosol mass and number concentrations from ART. The aerosol - cloud microphysics interaction is not included in this study as it creates a new set of research questions and the focus in this study is on the mineral dust radiation interaction. The combined effects of the mineral dust radiation interaction and its impact on cloud microphysics are investigated and quantified in a separate publication for a different event (Rieger et al., 2017).

ICON-ART is initialized with analysis fields from the Integrated Forecasting System (IFS) of the European Centre for Medium-Range Weather Forecasts (ECMWF). A limitation with initializing from the IFS analysis datasets is that the IFS has a horizontal grid spacing of approximately 13 km and is therefore non convection-permitting. Thus, when re-initializing ICON-ART with the current meteorological fields any previously existing organized convection is terminated. The IFS initialization data for soil moisture was modified in a region along the Syrian-Iraqi border which showed high soil moisture values and spatial inhomogeneities without preceding rain or changes in soil properties. Therefore, in a region from $37.5° \, \text{N} - 41.5° \, \text{N}$ and $32.5° \, \text{E} - 35° \, \text{E}$ the soil moisture index in the four layers provided by the IFS is set to the average value of the region between $36.5° \, \text{N} - 38° \, \text{N}$ and $32° \, \text{E} - 34° \, \text{E}$. This is done in order to prevent a possible effect of soil moisture on dust emission in this region, where dust emission is likely to be under-estimated due to the recent changes in land use conditions (see Sec. 2). The region modified is an important dust source region and emission fluxes for mineral dust increased due to the reduction of the soil moisture content.

The ICON-ART mineral dust dust concentrations are passed on to the next run whenever a reinitialization of the meteorological fields from an IFS analysis is performed. No assimilation of mineral dust concentrations from observations takes place. Hence, the runs are performed as free runs for the mineral dust concentrations with a frequent update of the meteorological background conditions from IFS analysis in order to benefit from the data assimilation performed therein. A two week spin-up simulation is performed starting on 23 August 2015 in order to achieve a realistic background concentration for mineral dust on the global domain. Another reinitialization from an IFS analysis is performed at 00 UTC 04 September to obtain a realistic background concentration of mineral dust in the finer domains for the simulations starting at 18 UTC 05 September. From 00 UTC 06 September onwards, a multitude of partially overlapping events take place which need to be simulated correctly in order to obtain a realistic dust distribution in the EM. Therefore, the time chosen for initialization is of crucial importance due to the aforementioned termination of organized convection. Various options were examined, the times chosen for initialization are 18 UTC 05 September and 12 UTC 06 September.

For investigation of the dust radiative effect two simulations are performed. The first is the simulation including the on-line radiation interaction with mineral dust from ART which is called 'ARI' (aerosol radiation interaction) in the following. The second is a simulation in which the mineral dust concentrations are multiplied by zero in the ART routine calculating the radiative effect of dust. This simulation is called 'CTRL' (control) in the following, it contains no mineral dust influence on radiation at all. In this study only the mineral dust radiation interaction is simulated on-line by ART, for all other aerosols the default climatologies are used in all runs.

## 3  Simulation results and validation

For the global grid, ICON-ART produces results comparable to those from other global models. However, due to its flexible nesting capability, it allows for convection permitting simulations for the finest resolution. As is shown in this section, ICON-ART is thereby able to resolve the meteorological drivers of the event in great detail. The results show that the dust event consists of multiple stages and is created by the interaction of different meteorological systems. A comparison of model results to available satellite observations highlights the ICON-ART simulation quality.

### 3.1  Synoptic situation

For the event simulated, the synoptic conditions in the Middle East are distinctively different from the normal summer situation as the ICON-ART model results show. An overview of the modelled synoptic situation is provided in Fig. 3 at 18 UTC 06 September, as at this point in time all relevant synoptic features are detectable.

The usually dominant Persian trough (Bitan and Sa'Aroni, 1992) is not present. Instead a low-level synoptic pattern termed Red Sea Trough (RST) with an eastern axis is apparent below 800 hPa, which extends northward from the Red Sea towards central Syria under easterly flow conditions. The direction of the trough axis and its position over the Red Sea make it plausible to attribute it to the RST category rather than to the Persian trough category. Furthermore, a characteristic strong southerly flow exists in the Mesopotamia region, which is crucial for the sustained lifetime of the meso-scale organized convection as it advects hot and moist air from the Persian Gulf. The RST is accompanied by a mid-tropospheric trough between $700 - 400$ hPa extending far southward into the EM under westerly flow conditions. The trough axis runs approximately along the Dead Sea Rift Valley, curving east towards Egypt in the southern part. East of this trough (downstream) high wind speeds up to $20$ m s$^{-1}$ exist in a streak running from south-west towards north-east. The streak is situated exactly upstream of the meso-scale convective system (MCS) development region. As suggested in previous studies (Krichak et al., 2012; Vries et al., 2013), this marked jet could provide significant moisture transport for the MCS from the Red Sea and Africa in the form of an atmospheric river with high values of atmospheric humidity. At upper tropospheric levels above $400$ hPa a short-wave trough exists in the region above Syria. This trough advects positive vorticity and cold air into the region at high altitudes, thereby creating quasi-geostrophic forced ascent and potential instability, respectively. In addition, the strong wind-shear between lower and upper levels in the Mesopotamia region produces the conditions necessary for MCSs by enabling a separation of up- and down-draft. In combination with the orographic lifting by the Zagros mountain range, the position of the slowly eastward moving trough

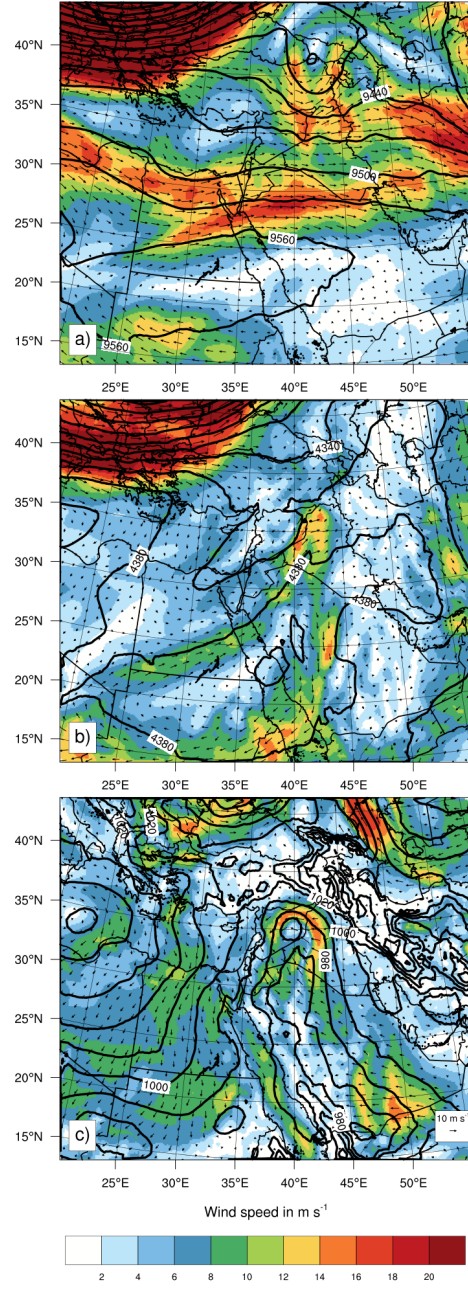

**Figure 3.** Synoptic situation on 06 September at 18 UTC as simulated by ICON-ART for the global domain. Shown are the a) 300 hPa, b) 600 hPa and c) 900 hPa level. Black lines denote the height of the respective pressure level in geopotential metres. Wind speed is colour coded and wind velocity is shown as vectors.

enables the development of meso-scale organized convection and determines its position throughout the event. The combination of a low-level RST extending northward and upper tropospheric troughs extending southward into the EM has been termed active RST in previous studies due to its high potential for severe weather (Krichak et al., 2012; Vries et al., 2013). The timing of the active RST synoptic situation is exceptional, as the RST usually starts to occur only by late September or early October (Alpert et al., 2004; Tsvieli and Zangvil, 2005), although events have been documented in August and September (Osetinsky, 2006). The active RST has been linked to severe weather phenomena in the EM and the Arabian peninsula in connection with atmospheric rivers transporting large quantities of precipitable water from eastern Africa, although the specific moisture sources are still debated (Vries et al., 2013, and references therein). The clustering of convectively active days with meso-scale convective organization is often observed during active RST situations (Krichak et al., 2012; Vries et al., 2013). This clustering is also in agreement with a study by Miller et al. (2008), who investigate haboob characteristics in the Arabian peninsula.

As discussed in Sec. 3.2, the active RST enables the interaction of multiple dust emitting meteorological systems over the course of three days, which explains the extraordinariness of the event with respect to magnitude and spatial extent. The unusual transport direction of the dust plume from Syria and Iraq into the EM from the east is caused by the downstream flank of the RST. To our knowledge, the active RST synoptic situation has previously not been linked to severe dust events in the EM and the exceptional character of this event is emphasized by a comparison with climatological studies (Singer et al., 2003; Alpert et al., 2004; Dayan et al., 2008).

### 3.2 Course of events

In the following, a detailed analysis of the development stages and responsible atmospheric drivers, which lead to the severe dust event, is provided. We focus on the results from the convection permitting domain, as it yields remarkable improvements compared to the global domain. The simulated convection and its interaction with dust emission is investigated in depth. Unfortunately, the region where the MCS and the first stages of the CPOs occur are located in the Syria-Iraq border region, which is not covered by a meteorological observation network. Therefore, no surface observations are available and the event can only be analysed using satellite data, which nevertheless yields interesting results. Figure 4 provides a schematic depiction of the event stages as well as their horizontal extent and course which are referred to in the following.

**First cold-pool outflow, heat low and Eastern Mediterranean sea-breeze**

During the night from 05 to 06 September 2015 a convective system exists over the Turkey-Syria border region. It is fuelled by the inflow along the eastern side of the RST. The system moves towards the north-east along with the mean flow direction above 500 hPa. Due to the favourable position of the convective system in front of the mid-tropospheric through axis, it intensifies during the course of the night. However, in contrast to subsequent systems, it lacks the full meso-scale organization of convection, possibly due to less favourable wind-shear conditions. During the early morning hours of 06 September 2015, the convective system produces a first, weak cold-pool outflow (termed CPO1 in the following) which remains decoupled from the surface due to the stable nocturnal boundary layer. As soon as the sun rises, the downward mixing of momentum increases and dust is picked up (a comparison of ICON-ART modelling results with satellite observations is shown in Fig. A1). The

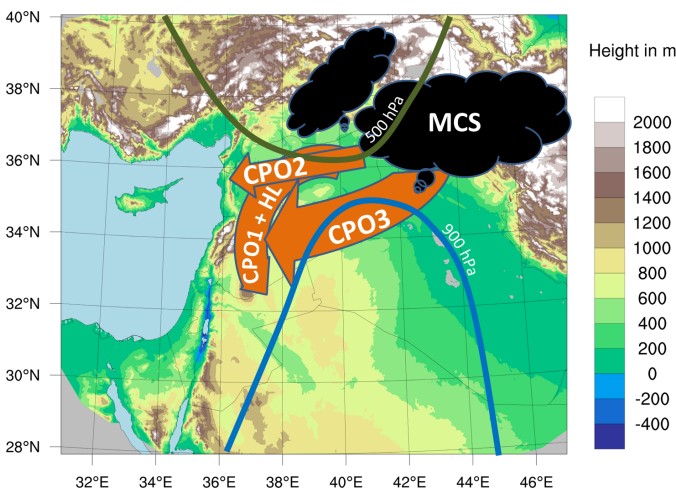

**Figure 4.** Schematic depiction the main dust event stages and atmospheric features. Colour-coded is the topography as used by the finest domain. Coloured lines represent average geopotential height of pressure levels. Labels refer to the description used in the text. Arrow size corresponds to event magnitude and arrowhead size to event speed.

high surface wind speeds are intensified and sustained during the day by a strong and shallow heat low (HL) forming in the developing boundary layer. The heat low is located in the tip of the RST and centred above Syria where temperatures during 06 September reach values above $45°$C. The maximum AOD value observed by MODIS is 2.78, compared to a dust optical depth (DOD) of 2.41 modelled by ICON-ART in good spatial agreement. Differences in the AOD distribution from MODIS

and DOD from ART over the EM are attributable to background aerosol (e.g. sea salt, black carbon), which is not represented in our simulation but measured by MODIS. It should be noted that MODIS can suffer from a systematic bias for AODs > 2.5, resulting in an AOD overestimation in the range from $0.5$ - $1.5$ as shown by Mamouri et al. (2016) through a comparison of MODIS and AERONET data in the region. Our analysis contrasts the simulation results by Solomos et al. (2017, their Fig. 4c), who model AOD values above 20 already before the onset of strong downward mixing of momentum. Furthermore, their

modelled bimodal maximum dust distribution was not observed by satellites and no closed cyclonic flow around the heat flow appears to have existed.

During the course of the day, the flow structure created by the CPO1 in combination with the heat low above Syria interacts with the inland penetrating sea breeze from the Mediterranean Sea, creating strong southward transport of dust towards Jordan. From 10 UTC onwards, the atmospheric instability created by boundary layer heating and upper level cold air advection

is released and deep convection starts to develop over the Syria-Iraq border region and Zagros mountain range in the RST inflow region. From SEVIRI Meteosat satellite images a second, convective cold pool outflow (CPO2) which travels west from the Zagros mountain range is detectable at 12 UTC. This CPO2 travels fast towards the west in the RST flow structure and supported by the heat low. The reinitialization of ICON-ART with IFS at 12 UTC impairs the CPO2 development due to the termination of convective structures. However, the main flow structures are still captured as the subsequent development shows.

The lifting caused by the gust front of CPO2 triggers initiation of deep convection over the Syria-Iraq border region which organizes into a MCS around 18 UTC. By this time the CPO2 has already travelled far into Syria. The MCS starts to develop a marked, third cold pool outflow (CPO3) from 20 UTC onwards. The CPO3 travels in the wake of CPO2, but in a more southerly direction. The strong third cold pool flow towards the south counters the inflow from southerly directions along the eastern flank of the RST, thereby lifting the warm and moist air masses. The above findings again contradict those of Solomos et al. (2017, their Fig. 7b), who in their model results find a northward travel direction of a small cold-pool structure. Based on the good agreement between ICON-ART and satellite observations this result is implausible. Furthermore, the intensity and spatial extent of their modelled CPO is much too small. A comparison of ICON-ART model results and SEVIRI RGB dust product observations as well as CALIPSO backscatter measurements is shown in Fig. A2 and Fig. A3. In addition, there are four cross-sections through ICON-ART results along which the event evolution and vertical structure can be tracked for all points in time discussed in this paper. The first cross-section runs from 35°N 32°E to 35°N 46°E along the 35°N circle of latitude, thereby providing insight into the east-west transport over Syria towards the northern EM (Fig. A4, left). The second cross-section runs from 32°N 34°E to 38°N 46°E along the main south-westward dust plume travel direction, thereby providing insight into dust transport towards the southern EM (Fig. A4, right). The third and fourth cross-sections run along both the CALIPSO overpass tracks (Fig. A5).

Past midnight on 07 September and explosive intensification of the MCS takes places due to the favourable atmospheric conditions. It develops a sharply defined, curved rainfall pattern in front of which the CPO3 is strongly intensified. In connection with the only slowly advancing upper atmospheric trough, which causes quasi-geostrophic forced ascent, and the orographic support from the Zagros mountain range, the MCS remains quasi-stationary over the next 12 hours. Due to the long duration and separation of the up- and down-draft region the MCS is able to produce an enormous amount of cool, moist air and a mighty CPO3 downstream a line shaped rainfall pattern. During the course of the night, the southerly direction of the CPO3 is deflected into a westerly flow direction by the RST flow structure and its inflowing air masses. The night-time spread of the CPO3 towards the west is crucial due to its subsequent interaction with the developing boundary layer mixing during daytime. During night-time the CPO is confined to a shallow layer of approximately 1 km close to the surface with wind speeds above $20 \text{ m s}^{-1}$ (Fig. A4, A5). Due to the stable stratification the dust plume does not cover the full CPO depth and dust concentrations are highest in the lowest hundred meters. With sunrise downward mixing of momentum increases. This leads to an increase of dust emissions and a greater dust plume depth which subsequently extends throughout the full CPO.

### 3.3 Meso-scale convective system and cold-pool outflows

An in-depth analysis of model results and comparison to satellite observations is conducted at 10 UTC 07 September (Fig. 5). The simulated MCS has passed its most intensive development stage approximately four hours earlier. The main MCS features are still visible, although it is in the stage of dissolution due to the shift of the upper atmospheric trough towards the east (Fig. 5a). The convection is organized along the orographic features of the Zagros mountain range, exhibiting a sharp line shaped rainfall distribution. Possibly, the convective structure is also shaped by a weak cyclogenesis taking place further east due to the favourable upper-tropospheric conditions. The cirrus cloud anvil extends far to the north-east as it is transported away by

the upper level flow. Downstream the line-shaped rainfall distribution near surface wind speeds increase strongly as the CPO3 reaches the surface (Fig. 5b). Wind speeds above 12 m s$^{-1}$ are modelled inside the CPO3 region. The southern edge is distinct and counters the inflow from the Persian Gulf region. The edge is also clearly recognizable in the 2m-dew point temperature (2m-DPT) field which is a good indicator for CPOs due to the change in air mass characteristic (Knippertz et al., 2007). The
difference in 2m-DPT between the CPO3 and the surrounding air masses is approximately 10 K (Fig. 5d). Dust has been picked up in large quantities over the previous hours with increasing boundary layer turbulence and DOD values above 2 are modelled in the CPO3 region (Fig. 5a). The maximum DOD value is 4.15, it is reached in an area close to the leading edge of CPO3 which shows the highest values of DOD.

Towards the north, the DOD exhibits a sharp gradient (northern boundary of CPO3 dust plume in Fig. 5). At closer inspection
of the wind and dew point temperature fields it becomes apparent that the decrease in DOD is not linked to the extent of the CPO3 in north-western Syria (Fig. 5b and d). This is due to the orography and the soil-type distribution in the region, a change in soil type towards less-emitting light clay and the increase in elevation are responsible for the reduced DOD towards the north. The northern edge of the aged CPO2 is marked by a line of convective clouds over the Taurus mountain range, here a second but less distinct gradient in DOD is visible. In the south-western part of Syria, entering Jordan, the remains of CPO2
are also still detectable by an elevated DOD above 1 and higher wind speeds up to 10 m s$^{-1}$. In addition, at a closer look the formation of small convective clouds above the nose of CPO2 can be seen in the ICON-ART model results. The aged CPO2 re-intensifies and the arc clouds develop further during the course of the day.

Comparing ICON-ART model results to the corresponding satellite observations shows that all modelled features are confirmed by the observations (Fig. 6). A comparison of the SEVIRI cloud cover with ICON-ART results shows that the cirrus
anvil of the MCS has a greater lateral extension in the model, whereas it is confined around 36° N in the observations (Fig. 6b). In addition, the MCS has moved further east by approximately 2° in reality. Taking the area with the brightest colour as a proxy for the strongest deep-convective area, the MODIS visible satellite image (Fig. 6c) shows good agreement with the line of highest integrated hydro-meteor content in ICON-ART (Fig. 6a).

The structure of the MCS determines the shape of the outflow, therefore differences in outflow structure can be identified
as well. Again, the CPO3 is more sharply confined between 34° N and 36° N in the observations, whereas it shows a bow shaped southward extension beyond 34° N in ICON-ART. The northward boundary of CPO3 is modelled accurately, with a northward deviation at the western tip. There is good agreement in the maximum dust plume optical thickness, with MODIS AOD measurements giving an AOD of 3.71 and the simulation a DOD of 4.15, although again the possible overestimation by MODIS should be kept in mind (Mamouri et al., 2016). Modelled DOD is higher in ICON-ART when compared to MODIS
AOD in the eastern part of CPO3 and lower in the western part of CPO3 (Fig. 6d). Unfortunately, no measurements by MODIS are available over the northern Mediterranean Sea where a substantial amount of dust is apparent in the visible satellite image. For the eastern part of CPO3 the MODIS AOD measurements seem doubtful when comparing to the MODIS visible satellite image.

The aged CPO2 spreading towards the north is detected by all satellite instruments, it is modelled with an offset towards
the north-east by ICON-ART (northernmost solid black line in Fig. 6). Noticeable are arc clouds forming above the Taurus

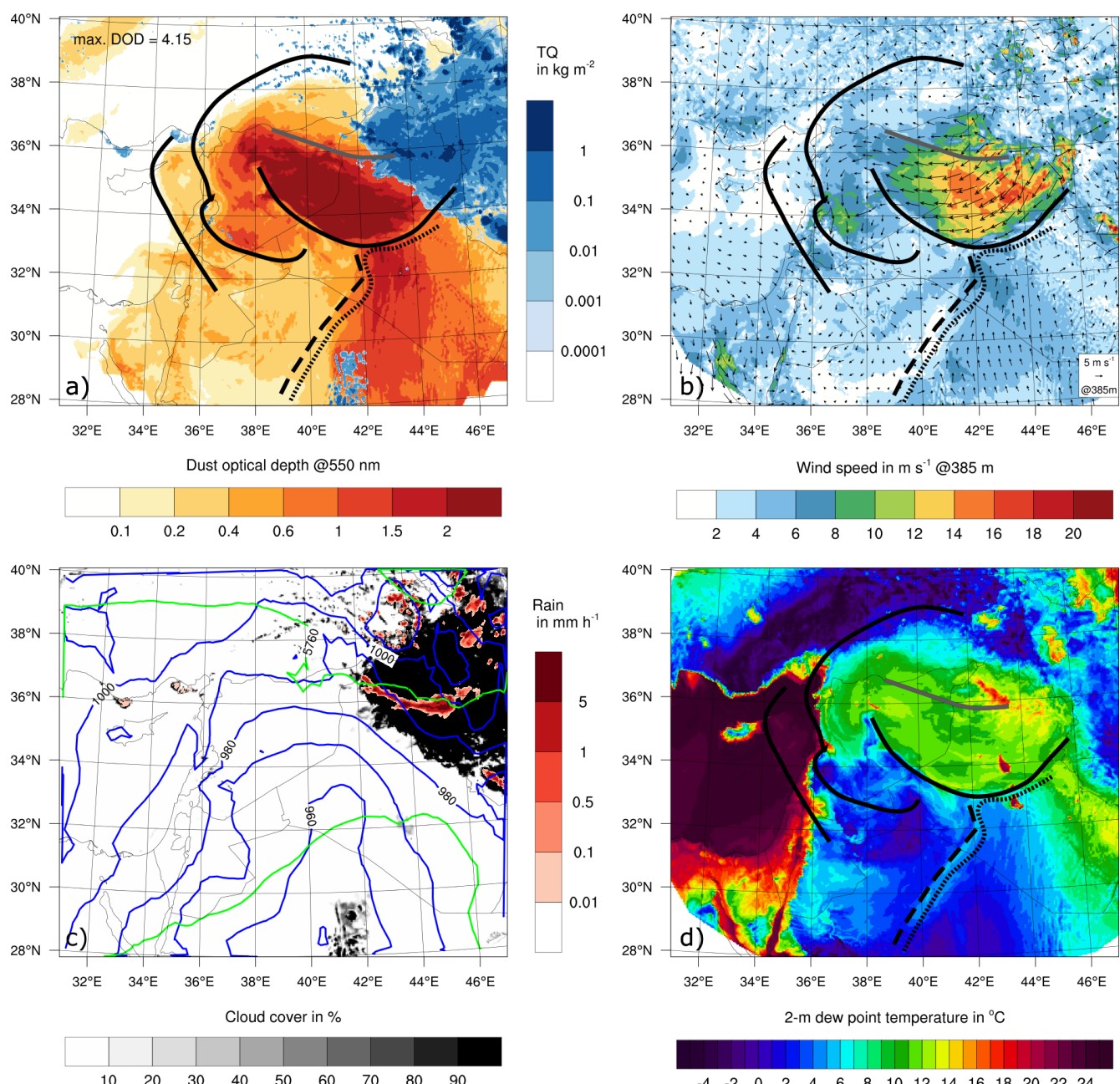

**Figure 5.** ICON-ART model results at 10 UTC 07 September. Displayed are: a) DOD at 550 nm overlain with column integrated hydro-meteor content TQ (cloud water, cloud ice and graupel). b) Wind speed at 385 m model height and wind velocity as vectors. c) Fractional cloud cover and surface rain rate colour coded. Blue lines show geopotential height of 900 hPa level at 10 gpm intervals, green lines show geopotential height of 500 hPa level at 20 gpm intervals. d) 2m-dew point temperature. From west to east solid black lines mark leading edges of CPO1, CPO2 and southern CPO3 boundary. Gray line marks northern dust plume boundary inside CPO3. Dashed line marks merged EM Sea-breeze front from 05 and 06 September. Dotted line marks frontal region of inflow.

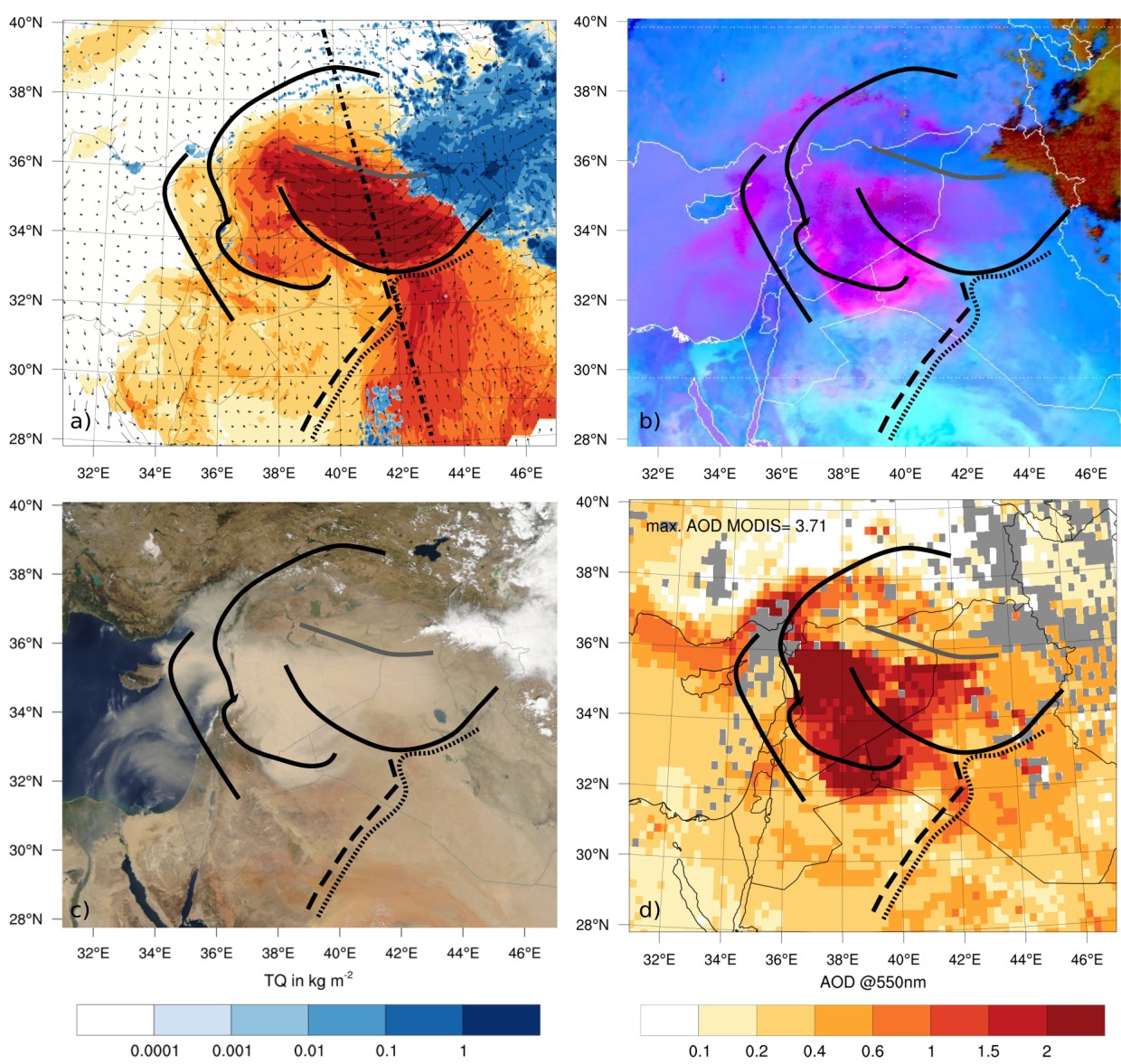

**Figure 6.** ICON-ART model results and satellite observations for 10 UTC on 07 September. Displayed are: a) ICON-ART DOD, column integrated hydro-meteor content and wind velocity. b) SEVIRI RGB dust product (Kerkmann et al., 2015). c) Aqua MODIS VIS satellite image, overpass at 10.35 UTC (NASA Worldview, 2016). d) Aqua MODIS AOD retrieval using DB2 algorithm (Levy and Hsu, 2015), overpass at 10.35 UTC. Chain dotted black line marks CALIPSO ground track at 10.35 UTC.

mountains in simulation and reality. The south-western boundary of the aged CPO2 is modelled correctly in the Golan Heights region. However, towards the west and south the CPO2 has advanced further in reality than in the model, as all measurements show a southern edge of the CPO2 which is just about to cross the border into Saudi-Arabia. Aerosol optical depths retrieved by MODIS are on the order of 2, whereas ICON-ART simulates values around 1. In this case, the MODIS measurements seem

more realistic when compared to the visible satellite picture. A possible reason for the underestimation of DOD magnitude in the model is reduced dust emission due to an unrealistic land use characterisation. Due to recent changes in the region an update of the land-use classes can be expected to yield significant improvements of modelled DOD magnitude (see Sec. 2) .

The vertical structure of the dust plume can be investigated at this point in time with the help of a CALIPSO overpass which occurred at 10.35 UTC. Results along the flight track are shown from south to north in Fig. 7. The southern inflow

region towards the MCS shows high values of the extinction coefficient in ICON-ART and attenuated backscatter in CALIOP measurements. The height of the dust plume is between $5 - 6$ km, with decreasing height towards the MCS. Just south of the main CPO3 dust plume, in a region from $32 - 33°$ N both show a decrease in dust concentrations at upper levels. However, a difference exists in the near surface values up to 2 km, where CALIOP reports high values of total attenuated backscatter but ICON-ART shows a minimum in extinction coefficient. The reason for this difference becomes clear when looking at the

CALIPSO ground track again in Fig. 6. The satellite passes the region where the merged one and two day old sea-breezes from the Mediterranean penetrate the RST inflow in a cyclonic rotational movement. The frontal structure is visible in Fig. 5 as a thin line in 2m-DPT, a gradient in DOD and a change in wind velocity along the front (as can be expected from a frontal structure). ICON-ART is able to simulate the re-intensifying sea-breezes, thereby enabling the above analysis, however no dust pick-up is connected with the frontal structure. This is due to the less emissive soil type in the region passed by the front. In

reality, however, dust was picked up by this frontal structure, this is detectable in all three satellite measurements (Fig. 6). As a result, CALIOP reports high values of attenuated backscatter in this region whereas ICON-ART simulates a minimum due to the clean air characteristics of the sea breeze.

The main CPO3 dust plume is distinct in both Figures with ICON-ART simulating an elevated nose of the CPO3. The above discussed differences in CPO3 structure are identifiable again. ICON-ART simulates a wider and more shallow outflow

whereas in reality it was more confined. In addition, the observations show an approximately one kilometre higher main dust plume compared to ICON-ART. Altitudes below 2 km are marked as no signal regions in the CALIPSO feature mask due to the attenuation of the lidar signal (not shown). On the northern side of the plume beyond $36°$ N a minimum captured by both is visible, once more illustrating the well simulated northern boundary of the CPO3 in this region. Further towards the north the overarching dust plume structure consisting of dust picked up by the aged CPO2 is represented very similarly in simulation

and measurement.

In summary, a northward deviation of the flow structure in ICON-ART can be identified both for CPO2 and CPO3 although the overall intensity and characteristics are simulated well. The combination of the northern deviation towards less dust emitting soil types and the wider, less well defined main outflow region reduce the amount of DOD over western Syria due to a less intense channelling of the CPO3. Furthermore, the re-intensification of the aged CPO2 is modelled by ICON-ART, however,

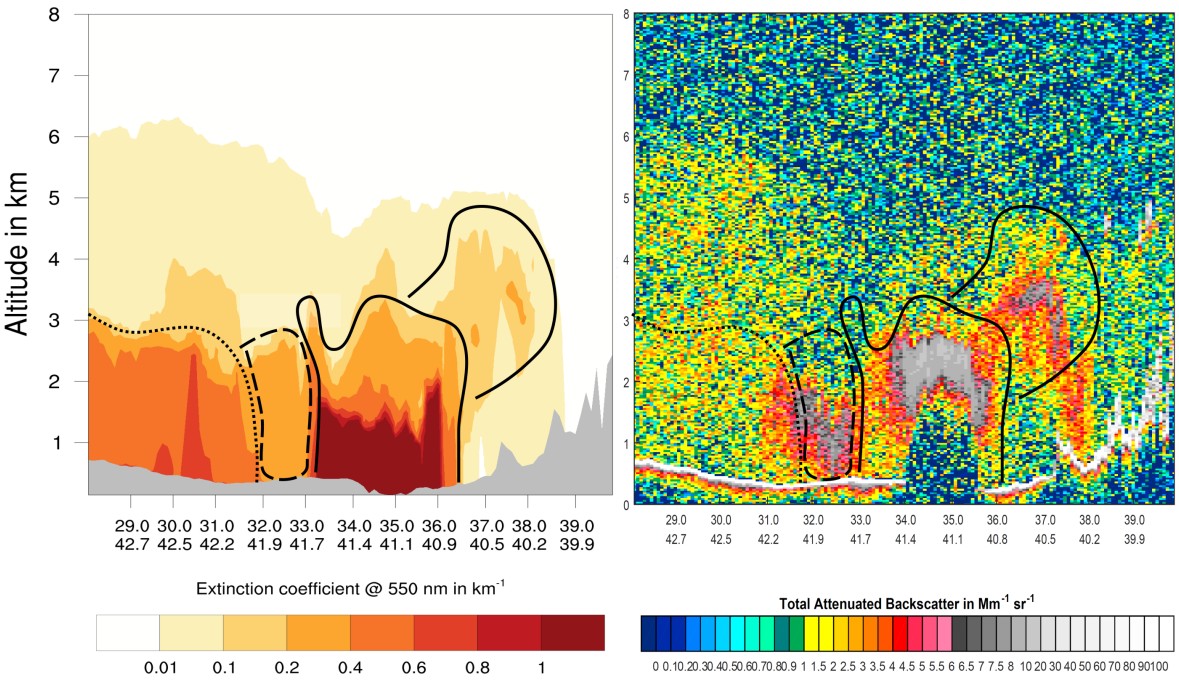

**Figure 7.** Vertical cross-section of ICON-ART model results for 10 UTC on 07 September and CALIPSO satellite observations from south to north along ground track in Fig. 6. Left side: ICON-ART mineral dust extinction coefficient. Right side: Total attenuated backscatter as measured by CALIOP. No quantitative comparison is possible as two different measures are displayed. Southern dotted black line marks inflow region. Dashed black line marks aged EM Sea-breeze penetrations region. Southern solid black line marks CPO3. Northern solid black line marks aged and lifted CPO2.

it was observed with greater magnitude in reality. The reduced amount of dust in the atmosphere in ICON-ART leads to a reduction in long range transport of dust towards the EM over night.

When comparing our results to those of Solomos et al. (2017, their Fig. 10), again large differences become apparent, both in spatial dust plume structure as well as vertical dust distribution. The driving MCS and related CPOs and their clearly marked 5  borders discussed above are not identifiable in their model results and they do not provide a detailed analysis of the flow structures.

### 3.4 Long-range transport to the Eastern Mediterranean

From 10 UTC onwards on 07 September, the MCS starts to dissipate and is in the course advected east along with the mean flow conditions above 500 hPa. The dust transport to the Eastern Mediterranean is complex, the previously identified plumes 10  are advected in different directions at different altitudes as visible in the SEVIRI RGB dust product animation (Kerkmann et al., 2015). This results in the multi-layered plume structure observed by Mamouri et al. (2016). The aged plume from CPO1

and HL is transported south-westward during the course of 07 September, consequently dust is detectable in Israel during 07 September already. The CPO3 airmass created during the night spreads south-west during daytime on 07 September. At all times it shows the characteristic features associated with CPOs such as an increase in surface wind speed, higher dew point temperatures and an arc cloud forming above the leading edge during the afternoon hours. The high surface wind speeds and turbulent mixing inside the CPO3s result in enormous dust emissions during daytime, consequently the dust is transported within the full boundary layer height up to 5 km (Fig. A4, A5).

With nightfall on 07 September the CPO2 and CPO3 merge. As the merged CPO is still located in the western, downstream flank of the RST the air mass and dust contained within it are advected fast towards the south-west over the course of the night. The Dead Sea Rift Valley is passed by the dust mass after midnight on 08 September, with the dust plume interacting with the complex orography. A comparison of simulated DOD with satellite observations for 11 UTC 08 September shows that the model represents the observed dust plume structure in the northern part of the EM (Fig. A6). The highest dust concentrations are present between Cyprus and Syria, although the dust plume has advanced approximately $2°$ further west in observations, reaching Cyprus. This shift can be explained by the northward deviation and less intense channelling of the CPO3 as well as the long forecast time. ICON-ART DOD values are one order of magnitude higher and show better spatial agreement than other global dust forecast simulation results in the northern part of the EM (see Sec. 1). Taking into account the $2°$ longitudinal offset in ICON-ART, the vertical structure of the dust plume arrival represents observations by Mamouri et al. (2016) (Fig. A7). A first, elevated plume extending from $2 - 4$ km with concentrations up to $1000 \ \mu g \ m^{-3}$ is noticeable during 07 September in both lidar measurements and model results. Mamouri et al. (2016) observe the arrival of the main dust plume past 19 UTC 07 September with concentrations up to $2000 \ \mu g \ m^{-3}$ at $0.75 - 1.5$ km height. ICON-ART shows the dust plume arrival past 21 UTC 07 September with concentrations up to $3000 \ \mu g \ m^{-3}$ at $0.5 - 2$ km height. During 08 September, dust concentrations increase up to $3500 \ \mu g \ m^{-3}$ and the plume thickness grows further, extending from $0.5$ km up to 3 km height in the model.

Dust transport into the southern EM is not simulated with the correct magnitude by ICON-ART despite the overall dust plume structure being captured, even when accounting for the MODIS AOD retrieval bias (Mamouri et al., 2016). MODIS measures AOD values consistently between $2-4$ over Israel, the Palestinian Territories and Jordan, and values above 5 over the southern EM. In this area, the contribution of the different plumes transported into the region along the Mediterranean coast from the north and across the Dead Sea Rift Valley from the north-east is especially complex due to the steep orography. Therefore, the transport into this region is analysed further in section 3.4.1 in order to investigate the differences found. Overall, a significant dust forecast improvement is achieved through convection permitting simulations with ICON-ART at 47 hour forecast time without data assimilation. During daytime on 08 September the dust plume is mostly stationary in the EM and influenced by the local circulation systems. In visible satellite pictures, the dust can be seen to remain in the EM at high concentrations over the course of the next four days. This period is not investigated as a part of this study as the scope of this work is the analysis of the generating mechanisms.

Due to the problems in simulating dust transport towards the southern EM with the correct magnitude, the next section investigates the timing and structure of the dust plume and CPO arrival in Israel in reality and simulation.

### 3.4.1 Environmental station data

In this section the simulated dust concentrations are compared to measurements from three stations in Israel. The selected stations are Afula, Jerusalem (Bar Ilan) and Beer-Sheva for PM10 measurements, as well as Sede Boker AERONET for AOD comparison (see Fig. 9c for a map of the station locations).

The stations all show individual dust concentration characteristics during the event, and the discussed features are present at other stations with similar characteristics as well. Beer-Sheva is chosen for its close proximity to Sede Boker at $45$ km distance, in a similar arid desert environment. Clearly four stations are not enough for a complete validation of the complex dust distribution in the region. The comparison shown here is meant to highlight differences between model and reality as well as to investigate dust transport features.

Due to its northern location the modelled main dust plume reaches Afula first at $00$ UTC $08$ September. Here, the highest values of DOD up to $1.5$ are modelled (upper row in Fig. 8). A few hours later, Jerusalem shows an increase and its peak optical depth. Even later, DOD increases in Sede Boker due to its location to the south of Israel, but the DOD values reach higher levels than in Jerusalem. Possibly, this is due to the lower altitude of the station as well as due to local dust emission.

A comparison of modelled DOD and AERONET AOD measurements shows a similar development of the optical depth for $07$ September, although with an offset of $0.3$. The offset is explainable by AERONET measuring the optical depth due to all aerosol species, whereas we only display DOD from ICON-ART, as well as a possible underestimation of dust background concentration in the model. Nevertheless, the main signal appears to be shaped by mineral dust processes captured by ICON-ART. The maximum modelled DOD for Sede Boker is $1.0$ on $08$ September, compared to $4.1$ measured by AERONET. The AERONET values appear realistic, as they are in good agreement with MODIS AOD measurements in the region. Thus, ICON-ART shows an underestimation of DOD by a factor of four.

When comparing PM10 measurements, a larger difference between model results and observations on the order of one magnitude becomes apparent. The large differences in absolute values of modelled and measured PM10 concentrations (second and third row in Fig. 8) are probably linked to the underestimation of dust transport in connection with CPO2, possibly due to an out-of-date description of soil properties in Mesopotamia (see Sec. 2 and Sec. 3.3). Furthermore, there is an inadequately modelled interaction of the dust plume with the complex orography of the Dead Sea Rift Valley as the next section shows.

Despite the large difference in absolute values, some insight into dust transport processes can be gained by comparing the course of the measured and modelled PM10 concentrations. The DOD in Afula reaches highest values during the night from $07$ to $08$ September, and PM10 measurements show a corresponding increase up to $985$ $\mu$g m$^{-3}$, which is missed by ICON-ART. A second, higher peak in concentrations with values up to $3422$ $\mu$g m$^{-3}$ is measured after noon on $08$ September in Afula at $57$ m surface elevation. This is linked to the onset of turbulent mixing needed to transport the highest dust concentrations from $1$ km dust plume height towards the surface. The downward mixing of dust leading to the second, higher maximum is captured by ICON-ART, as the results show a similar shape of the PM10 concentrations curve compared to the measurements with a peak at $12$ UTC at $532$ $\mu$g m$^{-3}$.

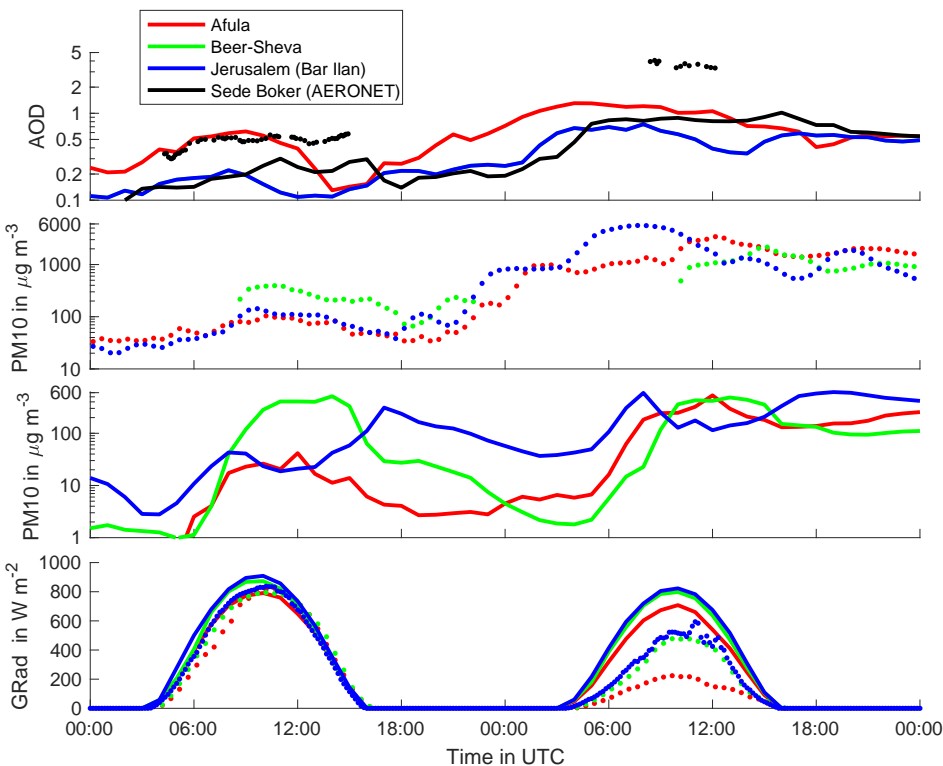

**Figure 8.** Observations and model results for three stations in Israel for 07 and 08 September. Points denote observations and solid lines model results. Please note the logarithmic y-axis scaling, which differs by a factor of ten between the second and third sub-plot. Shown are from top to bottom: 1. Modelled DOD and AERONET measurements for Sede Boker 2. Measured PM10 concentrations. 3. Modelled PM10 concentrations. 4. Global radiation (GRad), the values shown are for Afula, Beer-Sheva and Jerusalem (Givat Ram) due to data availability.

Highest overall PM10 concentrations are measured by the Jerusalem station with $5607 \, \mu g \, m^{-3}$. The main peak is the earliest and most pronounced peak of all stations at 08 UTC 08 September. This is also the case in ICON-ART, although with a shorter peak duration and at $615 \, \mu g \, m^{-3}$, one order of magnitude too low. The earlier and higher peak concentrations in Jerusalem are to due to the elevated location of the measurement station at 770 m, which is almost inside the dust plume. For this reason, PM10 concentrations are also more correlated with the shape of the DOD curve at this station. A secondary peak after sunset is visible in the measurements and model results.

PM10 measurements in Beer-Sheva show a first, pronounced peak during 07 September already, despite low values of AOD in Sede Boker. The first maximum is captured in magnitude and temporal duration by ICON-ART and can be linked to local dust emission. No measurements are available during the night, but on 08 September Beer-Sheva measured a maximum dust

concentration of 2155 $\mu$g m$^{-3}$, compared to 512 $\mu$g m$^{-3}$ modelled by ICON-ART. The underestimation by a factor of four between model and measurement is consistent with the AERONET measurements in Sede Boker.

ICON-ART fails to reproduce the measured low values of global radiation because for the radiative impact the absolute values of the dust concentration are of importance. The differences visible in the DOD above the stations manifest themselves in differences between the modelled amount of global radiation at the stations.

### 3.4.2 Hydraulic jump upstream the Dead Sea Rift Valley

In order to understand the problems of the model to forecast dust in Israel with the correct magnitude the transport processes upstream have to be investigated in more detail.

In Fig. 10, a cross-section of ICON-ART results along a cross-section through the Golan heights is displayed for 03 UTC 08 September. The location of the cross-section and corresponding meteorological situation is illustrated in Fig. 9. The arrival of the dust plume is simulated above a height of 1 km in Israel. This is in good agreement with measurements of the dust plume height in Israel which also report an arrival of the dust plume in 1 km height (Alpert et al., 2016).

Remarkable is a wave structure in the extinction coefficient on the lee side of the Golan Heights (Fig. 10). In the region of the wave structure, after passing the Golan Heights crest, the vertical depth of the dust plume decreases significantly to a few hundred meters. The plume expands and returns to its original height again in an abrupt expansion further to the west which is accompanied by a sharp decrease in flow speed. The reason for this behaviour is the existence of a flow phenomena termed 'hydraulic jump'.

Hydraulic jumps are connected to flows going from a sub-critical flow stage with Froude numbers $Fr$ smaller one, to a super-critical flow stage, with Froude numbers greater one. According to Drobinski et al. (2001) the Froude number can be calculated as:

$$Fr = \frac{U}{\sqrt{gh}} \cdot \sqrt{\frac{\Theta_v}{\Delta \Theta_v}}. \tag{1}$$

Thereby, $U$ denotes the wind speed and $\Theta_v$ the virtual potential temperature in the atmospheric boundary layer, chosen at a representative level and $\Delta \Theta_v$ the temperature inversion at the boundary layer top. Further, $g$ is the gravitational acceleration and $h$ the atmospheric boundary layer height.

Hydraulic jumps have been documented penetrating the Dead Sea Valley for flow conditions from the west (Metzger, 2016). The flow across the Golan Heights shows characteristic features of a super-critical flow with a subsequent hydraulic jump (see Fig. 10). These are:

1. A continuous decrease in mixing layer height detectable in the virtual potential temperature and extinction coefficient fields leading to a compressions of the sub-critical flow from east to west.

2. A sharp decrease in the vertical flow depth, connected to a rise in flow speed, after passing the orographic Golan Heights crest. This denotes the transformation of the flow state from sub- to super-critical.

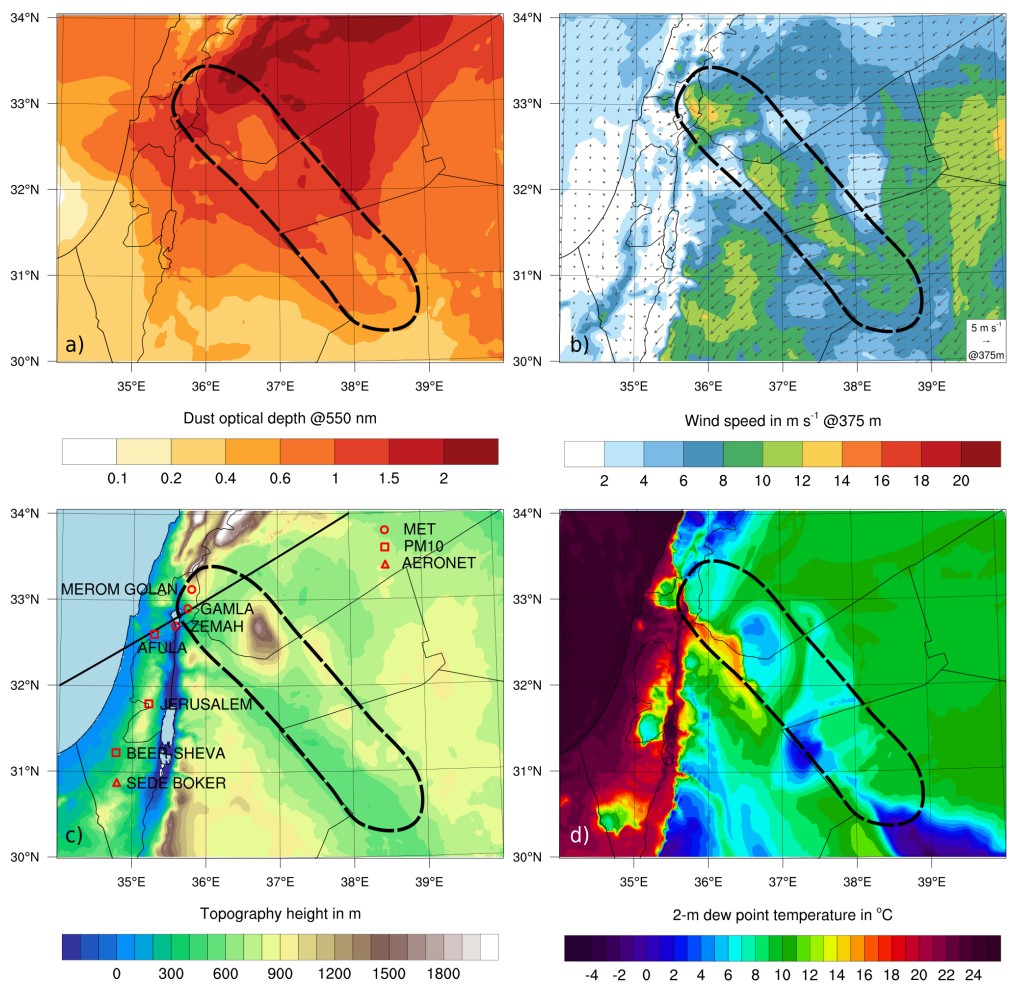

**Figure 9.** ICON-ART model results at 03 UTC 08 September, same as Fig. 5 but for the Dead Sea Rift Valley area. The bottom left plot shows the orography in the region colour-coded. The dashed black line marks the region where super-critical flow conditions and the subsequent hydraulic jump occur. The solid black line in c) shows location of ICON-ART vertical transect displayed in Fig. 10. In addition, location and names of stations used for comparison of results are displayed.

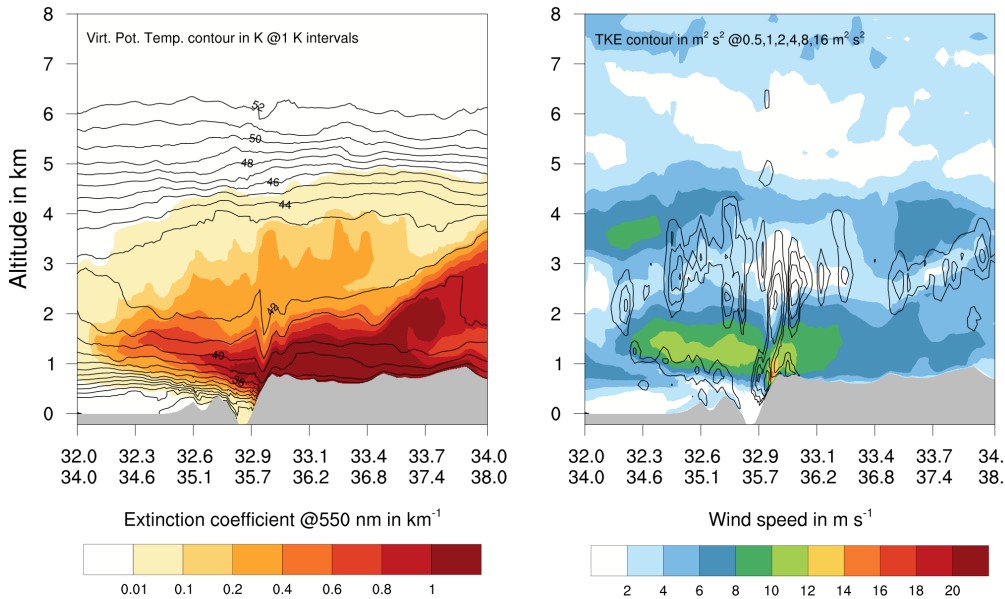

**Figure 10.** ICON-ART model results along a vertical cross-section from $32°$ N, $34°$ E to $34°$ N, $38°$ E at 03 UTC 08 September. Displayed on the left side is the colour-coded extinction coefficient overlain with virtual potential temperature contours. On the right side wind-speed is colour coded and overlain with turbulent kinetic energy contours.

3. A subsequent sudden increase in flow depth when the flow state reverts to the sub-critical stage again inside the Dead Sea Rift Valley. This is accompanied by a sharp decrease in flow speed and connected to an increase in atmospheric turbulence.

An approximate calculation of the Froude number upstream and downstream the Golan Heights crest height with average flow conditions as simulated by ICON-ART confirms the conversion of the flow state:

$$Fr_{upstream} = \frac{8 \text{ m s}^{-1}}{\sqrt{9.8 \text{ m s}^{-2} \cdot 1200 \text{ m}}} \cdot \sqrt{\frac{312 \text{ K}}{2 \text{ K}}} = 0.9,$$

$$Fr_{downstream} = \frac{12 \text{ m s}^{-1}}{\sqrt{9.8 \text{ m s}^{-2} \cdot 800 \text{ m}}} \cdot \sqrt{\frac{313 \text{ K}}{2 \text{ K}}} = 1.7.$$

Hydraulic jumps have been investigated and suggested as dust generating mechanisms (Cuesta et al., 2009). Their existence and interaction with the orography under strong easterly flow conditions has been proven for the Sahara (Drobinski et al., 2009). As the gust front of CPO3 reaches the Dead Sea Rift Valley almost along its full length, the flow reaches supercritical stage in the vicinity of many prominent orographic features (Fig. 9b). Consequently, the widespread existence of hydraulic jumps in connection with the upstream super-critical flow is assumed to be responsible for dust emissions on the eastern side of the Dead Sea Rift Valley. Dust emission in the region is assisted by the timing of the event, as in September the soil in the region is most erodible due to the preceding hot and dry summer.

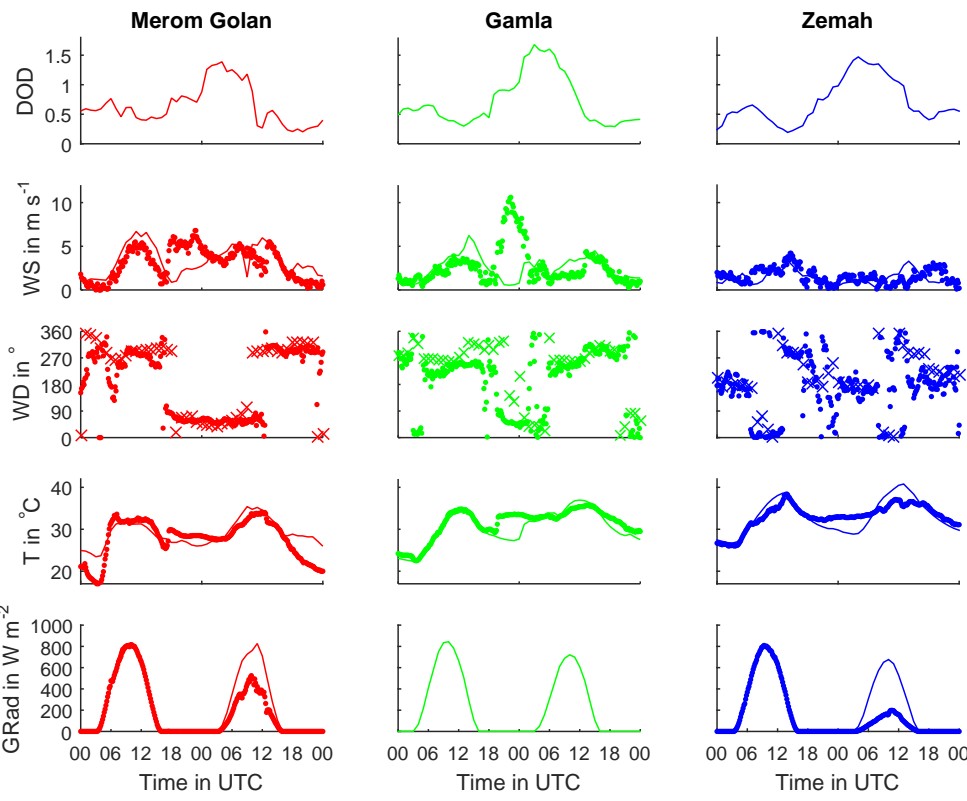

**Figure 11.** Observations and model results for three stations in the Golan Heights for 07 and 08 September. Points denote observations and solid lines/crosses model results. Shown are from top to bottom: 1. Modelled DOD 2. Wind speed (WS). 3. Wind direction (WD). 4. 2m-surface temperature. 5. Global radiation (GRad).

The existence of hydraulic jumps is modelled by ICON-ART and detectable along many cross-sections and at different times in the region. Some of them occur on the lee-side of much less pronounced orographic features such as the mountain structure visible at 33.5° N, 37.0° E in Fig. 10. Although ICON-ART simulates the flow phenomena, the location and magnitude of the hydraulic jumps need to be validated in order to evaluate the simulation quality.

### 3.4.3 Meteorological station measurements

The validation is done with the help of three meteorological stations approximately along the model cross-section through the Golan Heights. The locations and names of the stations used are marked in Fig. 9c. Unfortunately, no measurements are available in Jordan, where the existence of the hydraulic jumps is detectable in many places in ICON-ART and where the more erodible state of the soil is assumed to be responsible for high dust emissions.

The measurements of three meteorological stations are displayed together with the respective model results for their location in Fig. 11. The arrival of the main dust plume is simulated by ICON-ART past 18 UTC on 07 September. Subsequently, DOD values above 1.5 are reached during the night.

The development of the super-critical flow regime penetrating the Dead Sea Rift Valley is observed at two meteorological stations. It is detectable by marked, high wind speeds, a change in wind direction towards east and a sudden increase in temperature by approximately $4°C$. The higher Merom Golan station is first affected shortly before 18 UTC, whereas the lower Gamla station experiences high wind speeds only from 20 UTC onwards. Interestingly, confirming the super-critical flow theory, the highest values of wind speed are measured in Gamla around midnight. This is because the flow has more potential for gravitational acceleration from Merom Golan at 950 m elevation down to Gamla at 405 m elevation above sea level. Zemah, the station on the bottom of the Dead Sea Rift Valley at $-200$ m does neither report high wind speeds nor a change in wind direction at any point during the night as the super-critical flow does not reach the valley bottom.

The magnitude and timing of the super-critical flow regime is not captured correctly by ICON-ART, as a comparison of observations with model results shows. Although model results for Merom Golan show an increase in wind speed and a change in wind direction the results are far from the observed intensity and 2 hours late. For the lower Gamla station ICON-ART simulates the arrival of the super-critical flow with a delay of six hours and only a third of the observed intensity. As in the observations, the arrival is detectable in changes of wind speed, direction and surface temperature. The Zemah station and valley floor remains unaffected in ICON-ART as in reality. The possible reasons for the deviation of the simulation from reality include an unrealistic night-time boundary layer regime, incorrect atmospheric conditions upstream of the valley and a delayed arrival of the CPO3. As a result, possible mineral dust emissions due to the super-critical flow cannot be captured by ICON-ART. Through its destabilizing night-time boundary layer effect mineral dust itself can provide a positive feedback mechanism to higher near surface wind speeds and again higher dust emissions in the super-critical flow region. This is visible in the Zemah measurements, where the observations show much higher night-time surface temperature values than the model. As there is a reduced amount of dust in ICON-ART compared to reality, it underestimates the mineral dust radiative night-time warming effect.

As a result of the underestimated dust concentrations, ICON-ART is unable to capture the correct magnitude of reduction in global radiation due to mineral dust during daytime. Reductions in maximum global radiation for Zemah are modelled to be $125 \text{ W m}^{-2}$, whereas in reality $611 \text{ W m}^{-2}$ were observed between 07 and 08 September.

In summary, the existence of super-critical flow conditions in the region with connected hydraulic jumps is assumed to cause widespread and strong dust emissions on the eastern side of the Dead Sea Rift Valley. This contributes to the exceptional amount of dust in the southern part of the EM on 08 September. ICON-ART captures the special flow phenomena, albeit not with the correct magnitude and timing. The lack of a sufficiently developed super-critical flow and resulting high near-surface wind speeds prevents dust emission in Jordan and Israel in the model.

In combination with already underestimated dust emissions due to the recent land cover changes and soil degradation in the Mesopotamia region (see Sec. 2 and 3.3), this provides an explanation why dust transport into the southern EM is underesti-

mated by an order of magnitude. Nevertheless, ICON-ART provides a detailed understanding of previously unknown processes contributing to the historic dust event which makes these findings worthwhile to report.

## 3.5 Mineral dust radiative effect

The validation of the mineral dust distribution and transport characteristics with satellite and station measurements show overall
good agreement between ICON-ART and the observations, especially during the early stages of the event. The simulation therefore allows for investigation of the mineral dust radiative effect. Using ICON-ART, the radiative effect of mineral dust can be studied in detail through the differences between the simulation including mineral dust radiation interaction (ARI) and without it (CTRL). An analysis of the mineral dust radiative effect on surface conditions is conducted at two characteristic points in time during the dust storm development.

The first analysis is performed at 03 UTC 07 September as it shows interesting night-time features due to mineral dust (Fig.
12). The point in time chosen is just before sunrise in the region. The mineral dust affected areas show the expected behaviour in terms of its influence on radiative transfer. Net global radiation at night is increased by values between $5-50$ W m$^{-2}$, with higher values in regions with higher aerosol optical thickness (Fig. 12d). The increase is caused by an increase in down-welling long-wave atmospheric radiation (not shown). Consequently, the amount of energy radiated into space is reduced, which causes
a night-time warming of the surface. The reductions in net global radiation towards the east are caused by the daytime effect of mineral dust, as the sun is rising in this region already. The 2m-temperature field outside the CPO3 and MCS region (which themselves are discussed further below) shows the expected behaviour for mineral dust at night with an increase in surface temperature for ARI (Fig. 12c). Also visible are smaller scale irregularities showing local temperature differences located outside the region affected by mineral dust. These are introduced due to the slightly different location and timing of convection
in both runs. The magnitude of the temperature response to mineral dust radiative forcing is large when compared to other studies with values up to 5 K (Tegen et al., 2006; Highwood and Ryder, 2014; Rémy et al., 2015). A possible explanation are the high values used for the imaginary part of the mineral dust refractive index in the near infra-red spectral region, making the dust more absorptive. No homogeneously increased emissions are observed despite the decreased vertical stability outside the CPO3 area (Fig. 12b). The DOD is lower by more than 0.1 for most regions in the ARI run, although the maxima between
runs are similar. This results from the decreased emission during the previous day with a more stable boundary layer during daytime (not shown).

The DOD difference shows regions with an increased DOD at the leading edges of the CPOs in ARI (Fig. 12a). As the overall maxima of DOD are of comparable magnitude between ARI and CTRL, this can be attributed to a different propagation speed of the CPOs between runs. The CPOs in the ARI run show a higher propagation speed, which is not only detectable in the
DOD but also wind speed and 2m-DPT signal, especially at later stages (not shown). In ARI, the CPO with its steep DOD gradient arrives earlier compared to CTRL, therefore a higher DOD is shown for in the leading edge region at the same point in time. Furthermore, the leading edge is more sharply defined in the ARI simulation in general (not shown). The difference in DOD goes hand in hand with differences in dust emission, which are also apparent in Fig. 12.

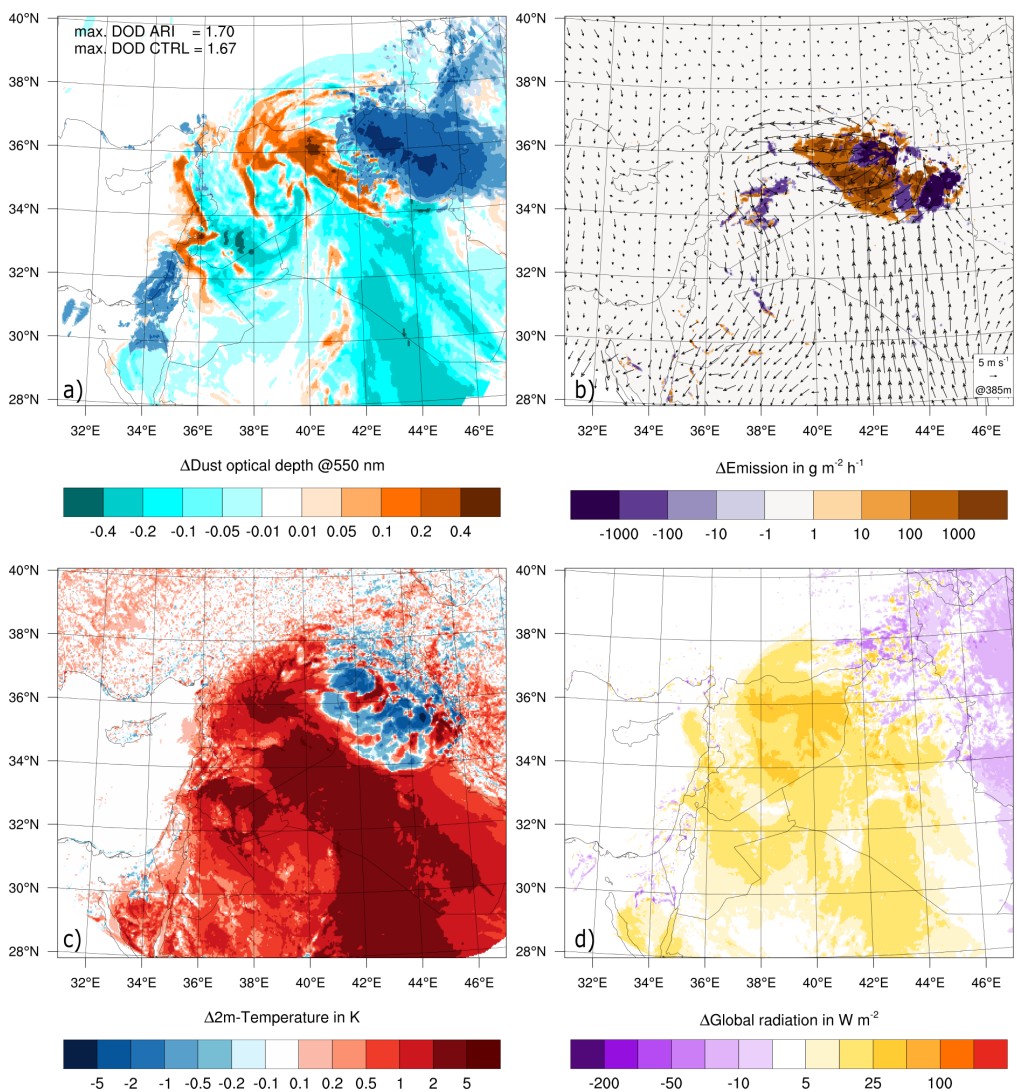

**Figure 12.** ICON-ART model results at 03 UTC 07 September, all results show ARI-CTRL. Displayed are: a) Column integrated hydrometeor content as in Fig. 5 and difference in DOD. b) Wind velocity and difference in hourly mineral dust emission. c) Difference in 2m-temperature. d) Difference in net global radiation at the surface.

Inside the CPO3 region a 2m-temperature cooling of more than 2 K can be observed for ARI (Fig. 12c). This is the opposite of the expected mineral dust night-time effect as discussed above. However, the mineral dust radiative forcing in this region remains positive (Fig. 12d). Dust emission increases in the ARI run despite a cooler surface, which again is not expected. All findings point towards a more intense CPO3 in the ARI run, which can explain all of the above observations.

A more intense and faster spreading CPO can have multiple reasons and further research is necessary in order to quantify the different contributions:

1. More intense convection leading to more rainfall which can evaporate, this in turn cooling the CPO more. The intensity of the convection can be increased due to a warmer inflowing airmass because of the mineral dust radiative surface heating at night.

2. More potential for evaporation due to a warmer surface boundary layer, also creating a cooler CPO.

3. A more stable stratified and thereby less turbulent CPO, preventing the loss of energy due to turbulent friction.

4. Travel of the CPO into a less stable night-time boundary layer due to mineral dust radiative surface heating. Therefore, less potential and kinetic energy of the CPO needs to be invested in order to lift the stable night-time boundary layer in front. Due to the reduced resistance propagation speed can increase.

The second analysis investigates the day-time radiative effect of mineral dust and is done for 10 UTC on 07 September (Fig. 13), this development stage is also analysed in section 3.3. The dominant effect is the reduction in incoming shortwave solar radiation, while the increase in down-welling atmospheric long-wave radiation is of lesser magnitude (not shown). Resulting reductions in net global radiation are more than $200 \, \mathrm{W \, m^{-2}}$ in the CPO3 region. The average value at the core of CPO3 in the region bordered by the $34°, 36°$ N circles of latitude and $40°, 42°$ E meridians is $-281 \, \mathrm{W \, m^{-2}}$ (Fig. 13d). The reduction in

incoming energy leads to a widespread reduction in 2m-temperature of more than 2 K. The average value in the aforementioned area is $-1.4$ K. The reduction in surface temperature is less than documented by other studies (Tegen et al., 2006; Helmert et al., 2007; Heinold et al., 2008). Possible explanations are the less absorptive character of mineral dust in ICON-ART with low values of the imaginary part of the refractive index in the short-wave region. Consequently, the dust mostly scatters radiation compared to absorbing it, thereby converting direct radiation to diffuse radiation. As the energy reaches the surface

nevertheless, reductions in surface temperature due to mineral dust are smaller than those found by two other studies. A detailed analysis for the 10 UTC situation supports this hypothesis. Decreases in direct shortwave radiation are on the order of $600 \, \mathrm{W \, m^{-2}}$ inside the CPO3 region. However, these are countered by increases in diffuse shortwave above $200 \, \mathrm{W \, m^{-2}}$, giving a net reduction in short-wave radiation of approximately $400 \, \mathrm{W \, m^{-2}}$ (not shown). Furthermore, the high amount of water vapour contained in the CPOs possibly absorbs and scatters large quantities of solar radiation without dust already,

thereby reducing the effect due to mineral dust. An observation which supports this hypothesis is that the reduction in 2m-temperature does not scale linearly with the difference in DOD between inflow and outflow region of the MCS.

The reductions in surface temperature have a stabilizing effect on the boundary layer (Heinold et al., 2008). As a result of the increased boundary layer stable stratification dust emissions decrease almost homogeneously throughout the dust affected

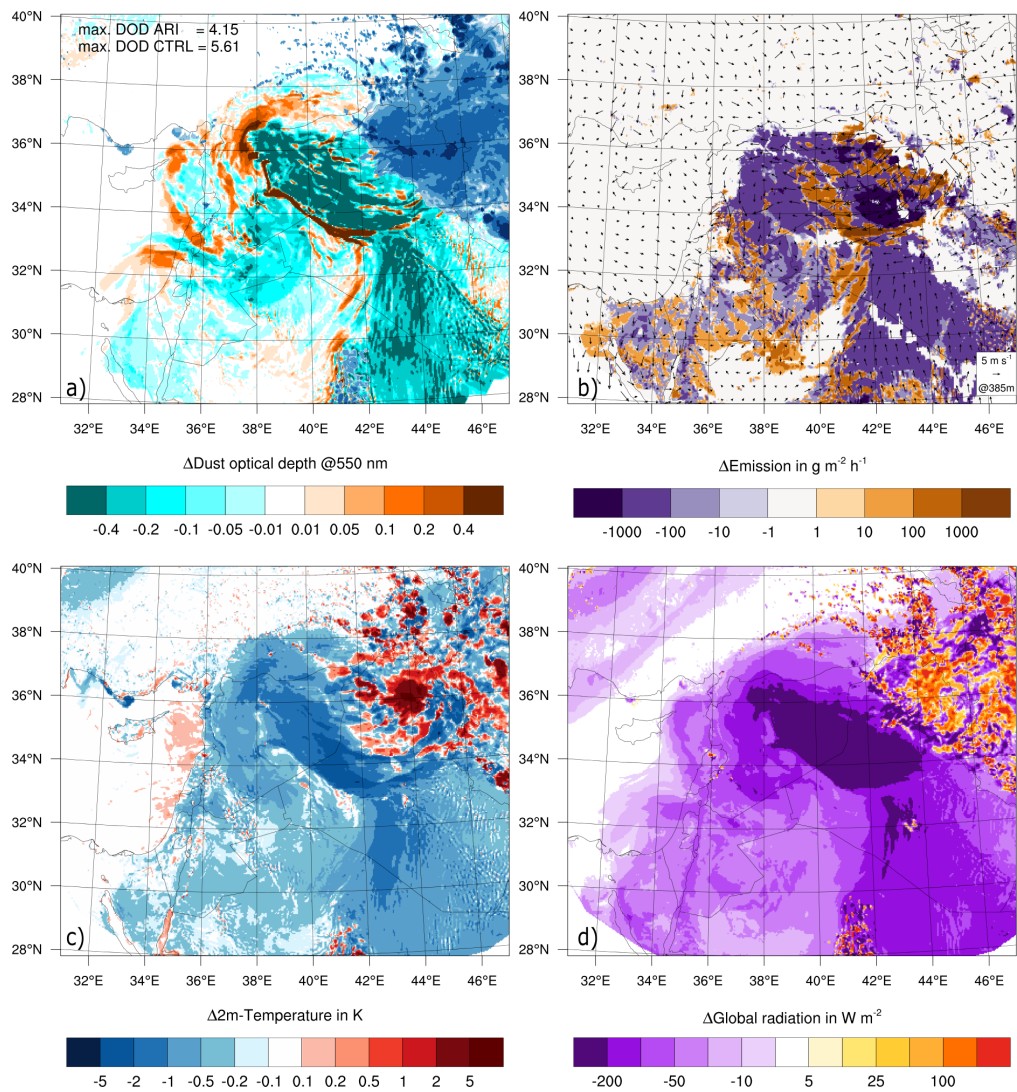

**Figure 13.** ICON-ART model results at 10 UTC 07 September, same as Fig. 12.

region (Fig. 13b). In addition, this results in a reduced maximum DOD of 4.15 in ARI compared to 5.61 in CTRL for the whole domain. During the late afternoon with maximum boundary layer development this difference increases. At 15 UTC ARI and CTRL show maximum DODs of 6.11 compared to 9.26, respectively. Another marked feature visible is the previously discussed difference in DOD at the CPO leading edges (Fig. 13a). As the feature is visible over more than 20 hours and in

connection with all CPOs, this confirms the observation of faster spreading CPOs in simulations including the mineral dust radiative feedback as proposed earlier. The streaks detectable inside the CPO3, where ARI shows higher values of DOD, can be attributed to shallow boundary layer convection, which develops earlier and more intensely in the CPO3 region in CTRL due to the lacking vertical stabilization (not shown). Consequently, due to the redistribution of dust by convection the DOD field is more inhomogeneous in the CTRL run (not shown).

**4   Conclusions**

This study presents a successful simulation of the September 2015 severe dust event at convection permitting resolution. An active Red Sea Trough situation and the related MCS and CPOs are shown to be responsible for the severe dust event in the Eastern Mediterranean. In addition, the interaction with an intense heat low, the inland penetrating Eastern Mediterranean sea-breeze and the widespread occurrence of super-critical flow conditions and subsequent hydraulic jumps are suggested as

important drivers for dust emission. The mineral dust radiation interaction has been implemented as a new module in ICON-ART. Based on Mie scattering calculations, which are conducted off-line to calculate the mineral dust optical properties, the radiative transfer parameters used by ICON are calculated on-line in ART to account for the mineral dust radiative effect. A new, size dependent parametrization of the mineral dust optical properties is proposed. Furthermore, to our knowledge this is the first study to investigate the mineral dust radiative effect on CPO structure. Summarizing, we are able to answer the

research questions presented in the beginning as follows:

(1) Is the forecast of the dust event improved by running convection permitting simulations?

The convection permitting simulation of the dust event with ICON-ART improves the forecast quality decisively. The driving meteorological systems and resulting dust emissions are captured in their horizontal, vertical and temporal structure as is shown by a comparison with satellite observations. The simulated DOD over Syria and Iraq is of realistic magnitude with

values above 2 throughout the main dust event region and maximum values above 6. The transport to the northern part of the EM and Cyprus is modelled with DOD values above 2 and in good spatial agreement with satellite observations at a $2°$ longitudinal offset towards the east. A comparison with lidar observations in Cyprus (Mamouri et al., 2016) shows very good agreement in vertical dust distribution. The simulated DOD in this region is less than observed, but one order of magnitude better than other operational global dust forecast models (see World Meteorological Organization dust forecast comparison,

http://sds-was.aemet.es), at a longer forecast time and without data assimilation. As the meteorological drivers are captured in detail, it is expected that the remaining underestimation of dust concentration is attributable to an out-of-date description of soil properties in the region due to the on-going conflict.

For the transport to the southern EM, a hydraulic jump is demonstrated to be of importance for dust emission in addition to the advection of the dense dust plumes into the region. It is captured by ICON-ART, albeit with reduced intensity compared to observations. Due to the out-of-date soil conditions in the Mesopotamia dust source region and an underdeveloped hydraulic jump phenomena, dust transport into the southern EM is underestimated by one order of magnitude by ICON-ART. Modelled DODs are in the range of $0.5 - 1.5$ over Israel and PM10 concentrations reach up to $600~\mu g~m^{-3}$ in Jerusalem. Nevertheless, the characteristic dust transport features are captured. The arrival of the main dust plume during the night of $08$ September is simulated at 1 km height and subsequent downward mixing increases surface dust concentrations.

(2) What is the synoptic situation enabling this extreme event and how does it relate to its exceptional character?

The event is triggered by an active RST situation. The occurrence of the active RST situation at the beginning of September is unusually early, thereby explaining the extraordinariness of the event with respect to timing. Furthermore, the RST situation enables the interaction of multiple dust emitting meteorological systems over the course of three days, which explains the extraordinariness of the event with respect to magnitude and spatial extent. In particular, the active RST situation favours a period of convectively active days with meso-scale organized convection and associated CPOs. In addition, the formation of an intense heat low above Syria is facilitated. The cyclonic flow around the RST provides the basis for the transport of dust towards southern and westerly directions on its downstream flank which explains the exceptional transport direction of the dust plume into the EM from the east.

(3) What are the meteorological drivers responsible for pick-up and long-range transport of mineral dust during this event?

During the early morning hours of $06$ September, a sharply defined CPO (CPO1) from an MCS over the Taurus mountain range interacts with a shallow but strong heat low forming in the boundary layer above Syria. Increased dust emissions occur as soon as turbulent mixing of the boundary layer sets in. Subsequently, the flow and dust plume created interact with the EM sea-breeze penetrating inland. Downstream of the upper tropospheric trough and with orographic support from the Zagros mountain range, a second MCS develops over the Turkey-Iraq-Iran border region from noon onwards. The MCS rapidly produces a CPO (CPO2) which travels west in the wake of CPO1 and the heat low, again producing substantial dust emission over central Syria. The lifting caused by the gust front of CPO2 triggers initiation of deep convection over the Syria-Iraq border region which organizes into an MCS around $18$ UTC. The MCS is again located in a dynamically favourable position downstream the quasi-stationary upper tropospheric trough. It produces another CPO (CPO3) from $20$ UTC onwards. CPO3 subsequently counters and lifts the inflow from the Persian Gulf along the RST flank, thereby fuelling the MCS and enabling its sustained lifetime of more than 12 hours. During night-time, the CPO3 gains momentum and spreads towards west. With sunrise and the onset of boundary layer mixing intense dust pick-up occurs in the CPO2 and CPO3 region. The aged plume from CPO1 and the HL is transported westward and south-westward along the EM coast, leading to a first arrival of dust in the region. During daytime on $07$ September, the MCS dissipates. The dust plume connected to CPO3 travels into a south-westerly direction supported by the flow on the downstream flank of the RST. The Dead Sea Rift Valley is passed by the merged dust plumes of CPO2 and CPO3 after midnight on $08$ September. During the night, the flow interacts with the complex orography.

As a result, widespread super-critical flow conditions, and the subsequent hydraulic jumps, occur in the lee of orographic features. This flow phenomena and the related dust emissions add to the extreme dust concentrations in the southern EM on 08 September. During daytime on 08 September, the dust plume is mostly stationary in the EM and influenced by the local circulation systems.

(4) How does the mineral dust radiative effect influence the dust event in general and the evolution of the CPOs in particular?

Besides the feedbacks of the mineral dust radiation interaction which have been identified in the literature before, a previously undocumented effect is found inside the CPO regions. Systematically more intense CPOs and a faster propagation of the CPOs in the mineral dust radiation interaction run are modelled.

In conclusion, this comprehensive case study has demonstrated the need to explicitly represent deep moist convection in dust storm forecasting. While Pantillon et al. (2016) propose a simple parametrisation to represent the climatological effects of haboobs in coarser resolution models, the forecasting of severe events like the one investigated here can hardly be successful without explicit convection. Given the substantial impact of the event and the potential benefit of an early warning, forecasting centres around the world should consider running higher-resolution dust forecasts for the most vulnerable regions. More research is also needed into the multi-scale interactions between Red Sea troughs, heat lows and convection. Moreover, the role of hydraulic jumps for dust emission and transport in the Dead Sea valley appears an interesting subject for further study, which would greatly benefit from a denser observational network.

**Appendix A**

*Author contributions.* The ICON-ART simulations were conducted and analysed by P. Gasch with the help of D. Rieger and C. Walter. B. Vogel supervised the work and contributed to the discussion and design of the experiment. The Mie-calculations for the mineral dust radiation interaction were conducted by P. Gasch and implemented in ART together with D. Rieger and C. Walter, based on input by B. Vogel. The Israeli station data was provided by P. Khain and Y. Levi who also aided the dust transport discussion as well as synoptic situation classification. P. Knippertz contributed to the discussion of meteorological drivers and event stages and the mentoring of the work leading up to this publication. P. Gasch prepared the figures and the manuscript, which was improved by all authors.

*Acknowledgements.* A special thanks goes to Christoph Kottmeier for his efforts in mentoring the work leading up to this publication. EUMETSAT and its training group (Jochen Kerkmann, Hans-Peter Roesli and Sancha Lancaster) is gratefully acknowledged for granting the right to use the MSG animation of the dust event. We would like to thank Tami Bond and Christoph Maetzler for making their MatLab based Mie-code publicly available. We thank Arnon Karnieli for his effort in establishing and maintaining the Sede Boker AERONET site. The MODIS Aerosol Optical Depth datasets were acquired from the Level-1 Atmosphere Archive and Distribution System (LAADS) Distributed Active Archive Center (DAAC), located in the Goddard Space Flight Center in Greenbelt, Maryland (https://ladsweb.nascom.nasa.gov/). The

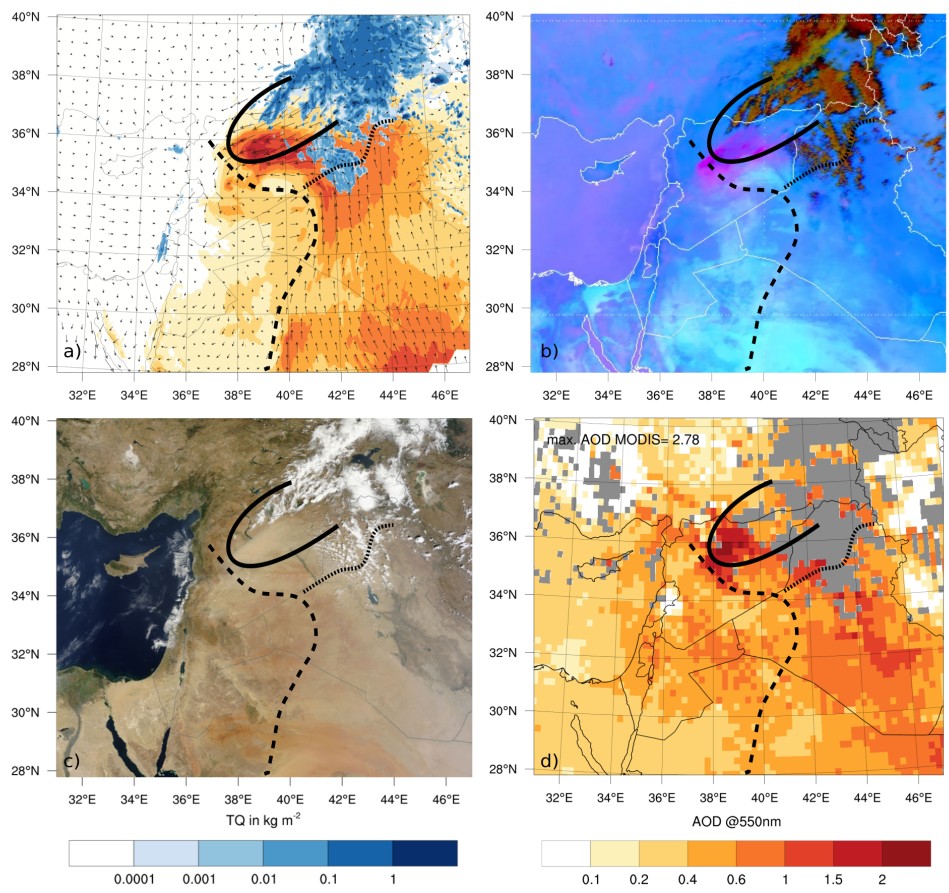

**Figure A1.** ICON-ART model results and satellite observations at 08 UTC 06 September. Solid black line marks CPO1, dashed black line EM Sea-breeze front from 05 September, dotted black line inflow frontal structure. Displayed are: a) ICON-ART DOD, column integrated hydro-meteor content and wind velocity. b) SEVIRI RGB dust product (Kerkmann et al., 2015). c) Terra MODIS VIS satellite image, overpass at 08.15 UTC (NASA Worldview, 2016). d) Terra MODIS AOD retrieval using DB2 algorithm (Levy and Hsu, 2015), overpass at 08.15 UTC.

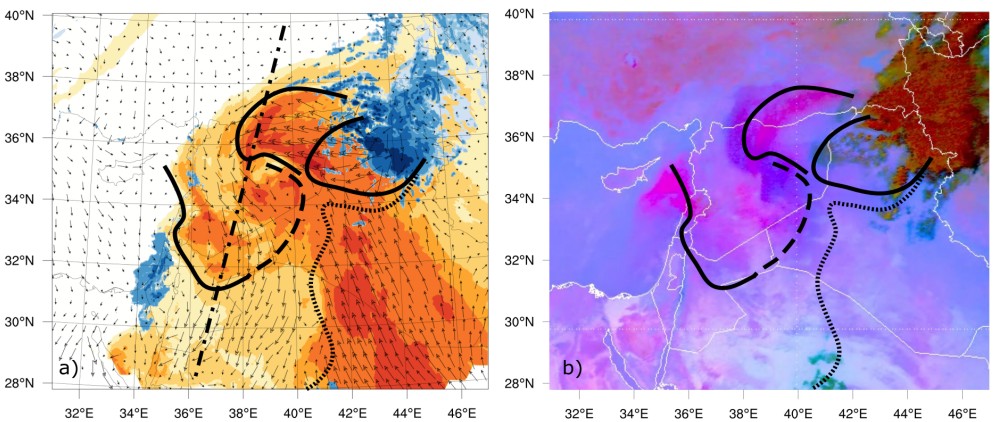

**Figure A2.** ICON-ART model results and EUMETSAT satellite observations at 00 UTC 07 September. The axis and label bars for DOD (ICON-ART and MODIS AOD) and column integrated hydro-meteor content are equal to the ones used in Fig. 5 and therefore not shown again. From west to east solid black mark leading edges of CPO1, CPO2 and CPO3. Dashed line marks EM Sea-breeze front from 06 September. Dotted line marks frontal region of inflow. Chain dotted black line marks CALIPSO ground track at 23.35 UTC 06 September. Left side: ICON-ART DOD, column integrated hydro-meteor content and wind velocity. Right side: SEVIRI RGB dust product (Kerkmann et al., 2015).

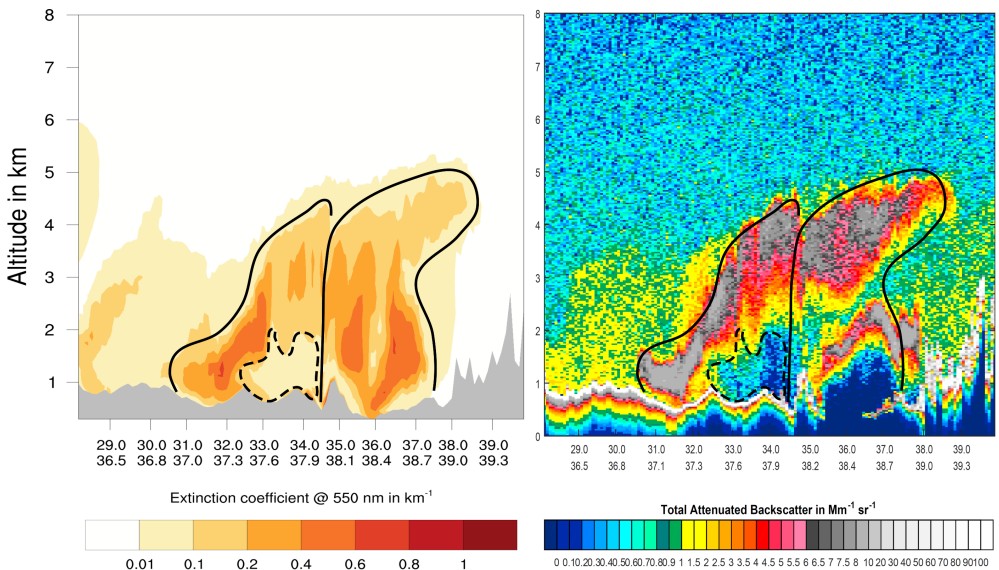

**Figure A3.** Vertical cross-section of ICON-ART model results at 00 UTC 07 September and CALIPSO satellite observations from south to north along ground track in Fig. A2. Left side: ICON-ART mineral dust extinction coefficient. Right side: Total attenuated backscatter as measured by CALIOP. No quantitative comparison is possible as two different measures are displayed. Southern solid black line marks CPO1 and HL region. Dashed line marks penetrating EM Sea-breeze. Northern solid black line marks CPO2 region.

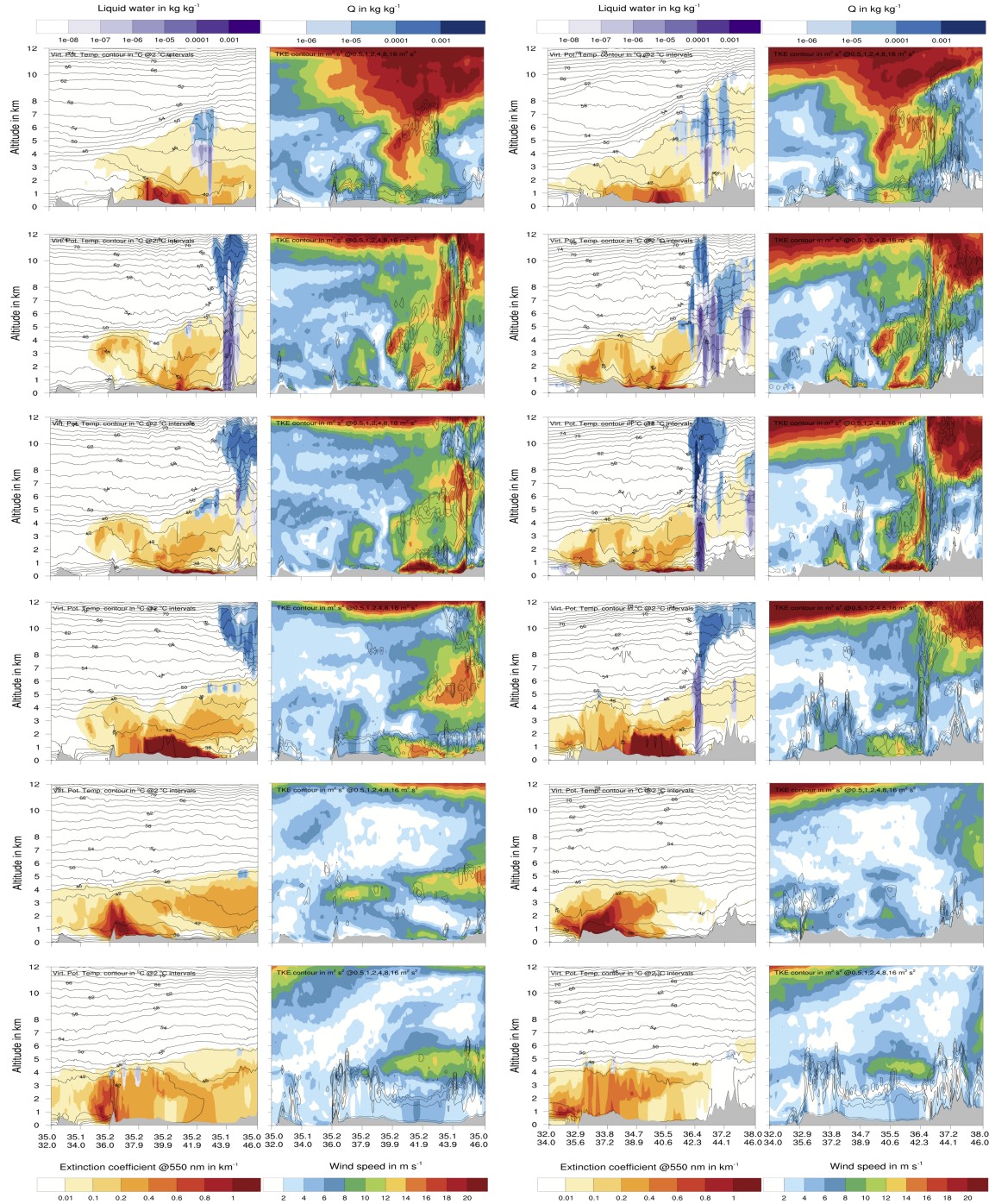

**Figure A4.** Vertical cross-sections through ICON-ART model results. From top to bottom at 08 UTC 06 September; 00, 03, 10 UTC 07 September; 03, 11 UTC 08 September. Left-side two figure column along 35° N circle of latitude from 35° N 32° E to 35° N 46° E. Right-side two figure column along main CPO3 dust plume travel direction from 32° N 34° E to 38° N 46° E.

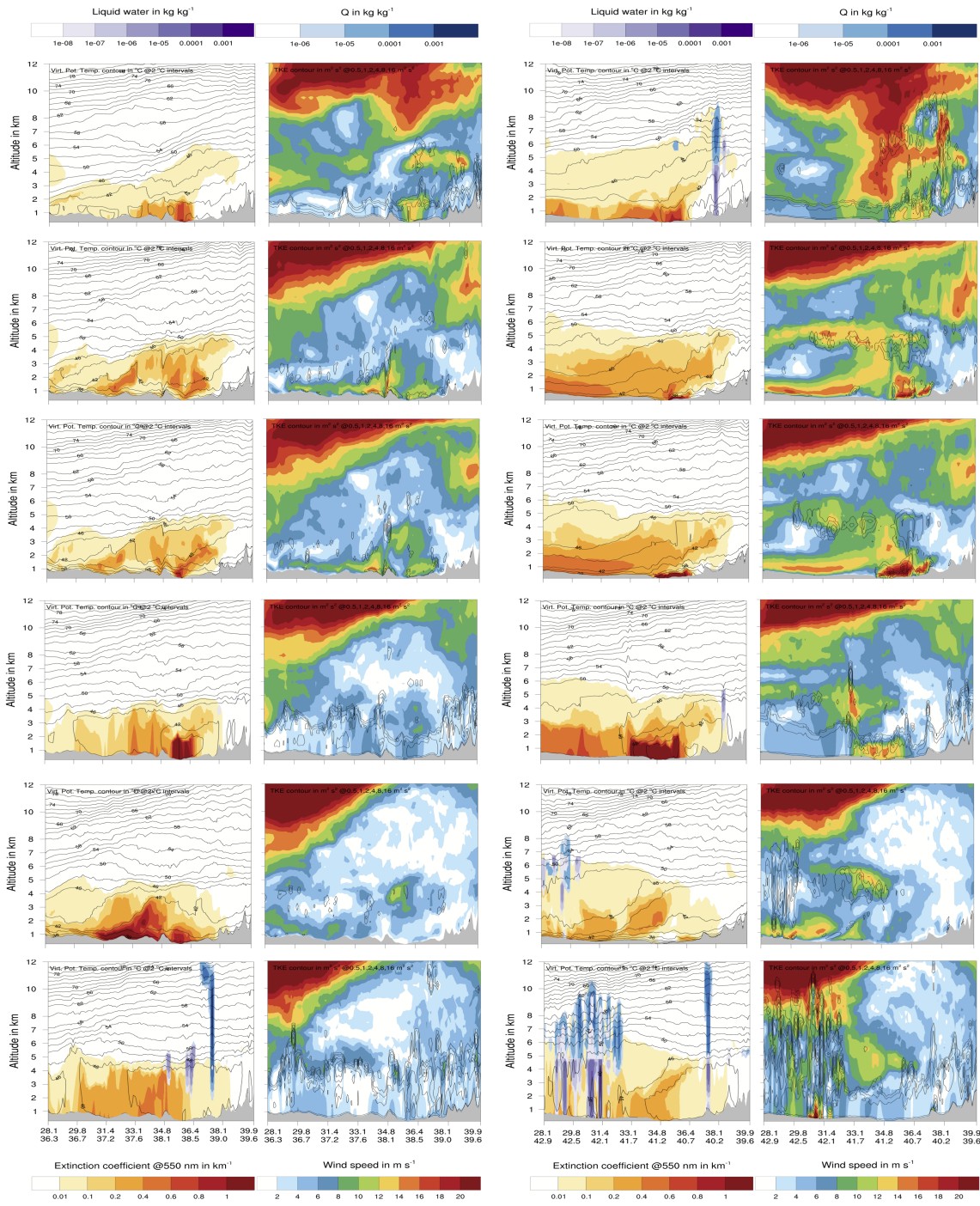

**Figure A5.** Vertical cross-sections through ICON-ART model results, same as figure A4, but along CALIPSO ground track at 23.35 UTC 07 September for left-side two figure column. Right-side two figure column along CALIPSO ground track at 10.35 UTC 07 September.

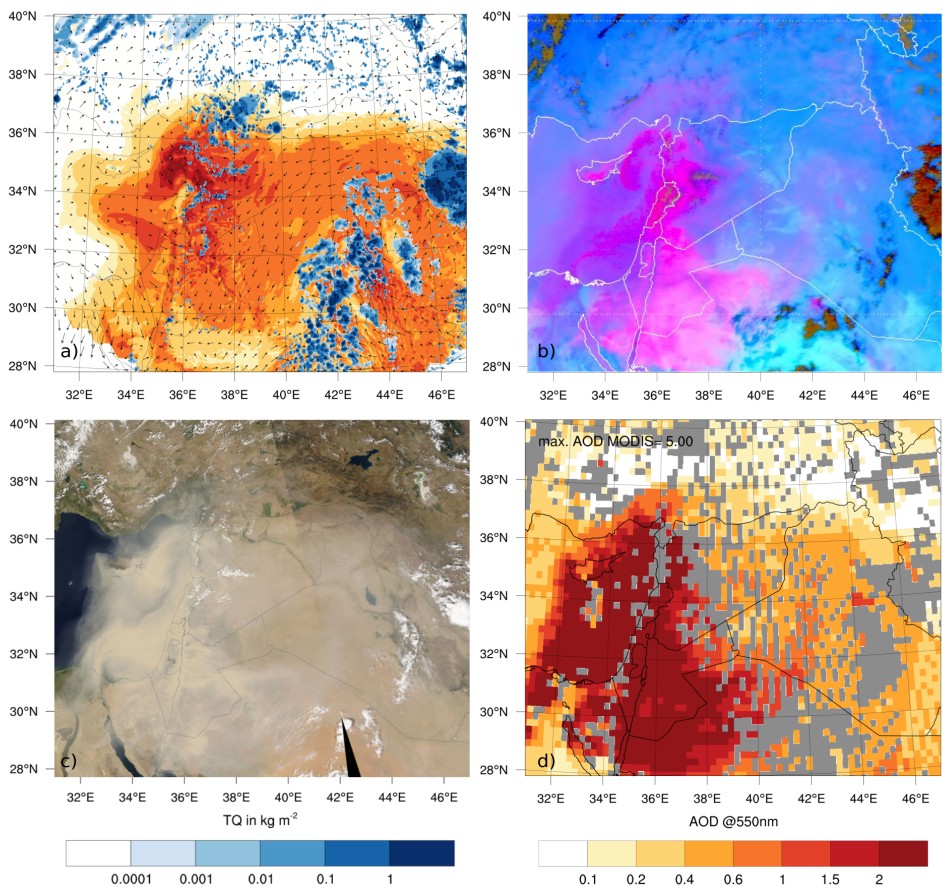

**Figure A6.** ICON-ART model results and satellite observations at 11 UTC 08 September. Displayed are: a) ICON-ART DOD, column integrated hydro-meteor content and wind velocity. b) SEVIRI RGB dust product (Kerkmann et al., 2015). c) Aqua MODIS VIS satellite image, overpass for the left part of the picture at 11.18 UTC (NASA Worldview, 2016). d) Aqua MODIS AOD retrieval using DB2 algorithm (Levy and Hsu, 2015), overpass for the left part of the picture at 11.18 UTC.

authors acknowledge EARLINET for providing aerosol lidar profiles available from the EARLINET webpage. Furthermore, we would like to thank two anonymous referees for their helpful remarks.

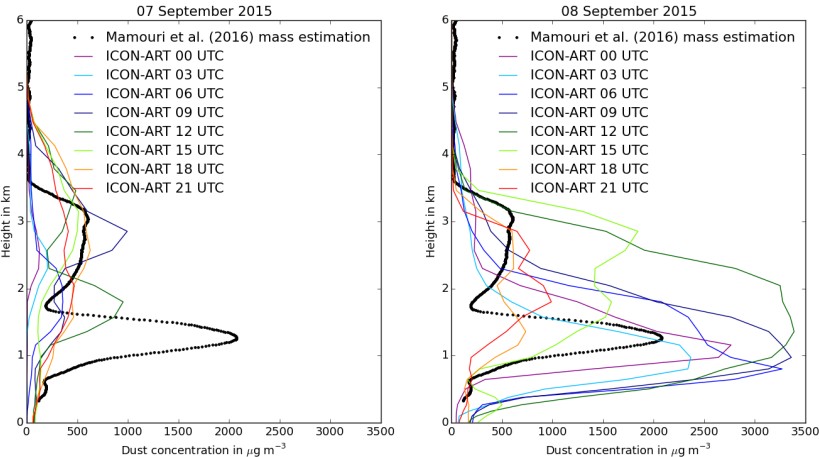

**Figure A7.** Mamouri et al. (2016) mass estimation from lidar observations for EARLINET station in Limassol, Cyprus, for 19 UTC 07 September and ICON-ART model results for 07 and 08 September at $34.7°$ latitude, $35.0°$ longitude. Please note the $2°$ eastward shift of the shown ICON-ART results compared to the Limassol observations due to the spatial offset of the dust plume, see Sec. 3.4.

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
