# Peer review of "Revealing the meteorological drivers of the September 2015 severe dust event in the Eastern Mediterranean"

_Atmospheric Chemistry and Physics, 2017_

## Short Comment (SC1) · 28 Mar 2017

The paper is well written and the results are very exciting. The enormous potential of the ICON model coupled with the state-of-the-art aerosol module ART is documented in a fascinating way.

Nevertheless, I was thinking some comments from my side should be given and may improve the paper a bit.

As a co-author of the papers of Mamouri et al., ACP, 2016, and Solomos et al., 2017, both dealing with this record dust storm in September 2015, and, in addition, as a lidar expert having a long-term cooperation with the lidar group at CUT, Limassol, I would like to recommend the following:

P1, L22: Please check the paper of Nisantzi et al., ACP, 2015, they report Saharan and Middle East lidar observations (2011-2014), perfomed in Cyprus, and provide statistical results. This article could be mentioned in the introduction.

P2, L8-13: We need a short discussion on the existing literature for this September 2015 dust storm, i.e., a short discussion of Mamouri et al., ACP, 2016 and the companion paper of Solomos et al., ACP, 2017 (just finally published on 27 March). This is the normal 'way of life' in science, i.e., to discuss previous work, to discuss what is already known, and what will be the new points of the new article. I speculate that you (the authors) did not read the final version of the Mamouri et al. paper with all the findings concerning mass loadings, dust height distributions, optical depths. . . because there are so many useful observations and findings that corroborate your statements and findings..... The submitted ACPD version of Mamouri et al. is very different from the final one. By the way, in that paper, also the limits of MODIS concerning max AOT retrievals are discussed, and the quality of MODIS data at such high AOT conditions is discussed.

P2, L31: This final sentence of the paragraph has to be 'updated' because there is this Solomos et al. (2017) paper. . . ... or what do you mean with . . . a detailed analysis of the driving atmospheric system . . . has not been published so far. . . . ? A mentuoned, the final version of the Solomos paper is now published.

P6, L10-11: just a short question: Why do you distinguish (always) sedimentation and dry deposition? I am not so familar with the terminology but to my opinion dry deposition includes gravitational settling. But maybe I am wrong.

Page 9, Figure 3 is very nice, but needs to be improved. . . . It is almost impossible to identify Turkey, Cyprus, Israel etc. . . .

Page 15, Figure 6, color scales are missing, but needed. The MODIS analysis stops when the observations indicate: AOD > 5.0 (as written in Mamouri et al, ACP, 2016). In the MODIS figure (Figure 6, bottom, left) all dust regions, where the surface (over

the dark Med sea for example) is not visible anymore, are regions with AOD of 5.0 and more. You may check the MODIS data basis (links are given in the Mamouri et al, 2016 paper). And this in contradictions with the MODIS results in Fig.6... Did you compute these AOD values (map), instead of taking the AOD values from official MODIS data sources?

What does the map (Figure 6, bottom, right) show? We need a color scale. And what about the region just east of Cyprus. The AOD is obviously very high (bottom, left, because the dark Mediterranean Sea is no longer visible, the AOD was rather high, probably 5.0 or even more), but no values in the MODIS AOD map (bottom, right). Impossible, to my opinion! Something is wrong with these MODIS products. Please check!

Figure 7 (right panel): CALIOP obs, 34-36 N, ..... The CALIOP retrieval gets lost at these conditions, the algorithm fails and cannot handle such situations. The dust extinction coefficients exceeded already 500 Mm-1at 3 km height.... and must be about 2000 Mm-1 or more at heights below 2 km to match the MODIS scence (Figure 6,bottom, left , AOD certainly larger than 5).

Page 17: I personally would like to see comparisons of ICON-ART results for the Cyprus region, for the 7-10 September period. But I am sure that huge deviations from our findings (presented in Mamouri et al., 2016) would become visible.

Page 19, Figure 8: Please check the Weizmann Institute AERONET station (a bit east of Tel Aviv) for 9 September (This station measured AOD of 2.4-2.7). What did you find for 9 September for Israel? On 8 September, the dust load was even higher, but there are no AERONET observations, because of too high AOD, which the AERONET algorithm misinterpreted as clouds, I am speculating.

That means, the modeled Jerusalem DOD values are much too low (by a factor 4...)

---

## Referee Comment (RC1) · Anonymous Referee #1 · 3 Apr 2017

The Manuscript entitled "An analysis of the September 2015 severe dust event in the Eastern Mediterranean" describes in great detail the mechanisms of the dust storm episode using the ICON model coupled with a desert dust module (ART). The paper is very well written and I would recommend it to be published on ACP after the following comments have been addressed:

1) Abstract, Line 3: The authors state that "...state-of-the-art dust transport models were unable to forecast the event...". I don't think this is accurate. For example the publication of Solomos et al., 2017 in ACP describes the same episode using another model. I think that the two manuscripts were submitted very close to each other and the authors were not aware of this, even though they give reference to this work when it was still in ACPD. Please correct this statement accordingly. This is present in other areas of the manuscript as well and should also be corrected.

2) Page 3, Line 20: The 3rd research question posed by the authors is "What are the meteorological drivers responsible for pick-up and long-range transport of mineral dust?". It is not clear of they mean in general or in cases such as the event described in the manuscript, because the dust-cycle mechanisms in general are well known and documented. Please rephrase

3) Page 4, Line 15: You state that "...the seamless modelling capabilities of ICON are of crucial importance because inconsistencies in tracer transport and tracer physics at the nest boundaries can be avoided...". Please remove the word "seamless". Also I cannot understand how these inconsistencies are avoided. Please expand.

4) Page 4, Line 22: What do you mean by sedimentation? Does it refer to the sandblasting mechanism for production or the deposition of particles?

5) Page 5, Line 9: It is important to see how the model defines the dust sources between the nests. Are they defined separately for each domain? If so how do you assure there are no continuity problems in the fields? Is this what you mean in Page 4, Line 15 (see comment 3)?

6) Page 5, Line 17: You state the timestep for calling RRTM is 288 seconds. Please provide the timestep of the simulation as well.

7) Page 5, Line 18: "...ART modifies the radiative transfer parameters of the climatological dust distribution...". What do you mean by "climatological"? Do you mean the dust distribution as described by the dust module? Please expand.

8) The same confusion in Line 20: "The parameters returned by ART are the combined values from the local ART dust concentration plus the Tegen climatology". During the simulation dust concentration is calculated using both prognostic dust and climatological values?

9) Page 6, Line 14: "Therefore, the median diameter of each mode is expected to decrease during transport". You mean that during the simulation the size of the particles

changes? Please expand a little as this is very interesting.

10) Page 7 Line 29: "... which represents the soil moisture conditions in the region more realistically...". Do you have a reference or actual data to support that this method provides more realistic values? This is essential as soil moisture dictates dust production. How can you be sure that the underestimation of the dust concentration by the model (as described in later chapters) is not attributed to false soil moisture?

11) Page 9, Figure 3: I would like to see clearer plots, especially the national borders as to know exactly where they refer to. Maybe resizing them?

12) Section 3: Since this is a very detailed description of the event it would be very interesting to see a vertical cross-section of dust concentration and precipitation in the same plot, for different forecast hours. Like what you have in Figures A1-A4, but vertically.

13) Figures 6 and A1-A4. Please add labelbars to the plots (where applicable of course).

14) Section 3.4.1: I would like to see a comparison for Dust Optical Depth from more stations in the computational domain. Is this possible? Maybe using AERONET data? If there is no additional data available then add a sentence in the text stating that.

15) The authors show that the ICON-ART underestimated dust concentrations and give a very thorough explanation as to why this happens. However how can you be sure that this substantial difference is not caused by something simpler like wrong description of the strength of the dust source areas (maybe in reality the areas are more active than described in the model) or, as I stated above, bad definition of the soil moisture? Have you tried some sensitivity runs based on these?

16) Page 28, Line 21: "...this study presents the first successful simulation of the September 2015 severe dust event...". See point (1) in my review.

17) Page 29, Line 10: "...ICON-ART results are one order of magnitude better than

**C3**

those from other models...". This statement is rather odd since you do not present the capabilities of other models in the manuscript to support this. Of course this does not reduce the very good performance of your model in any way.

18) Finally, as far as I can tell, the model does not support the indirect effect of dust particles (cloud and precipitation). Does this affect the performance in this particular case? If the model was able to simulate dust acting as CCN would the results be any better? Just add a small paragraph expanding on this.

In conclusion I believe that the manuscript deserves to be published at the Atmospheric Chemistry and Physics Journal after the issues above have been addressed.

---

## Referee Comment (RC2) · Anonymous Referee #2 · 19 May 2017

We recommend the paper publication with some revision outlined below.

Nice research in this article includes: 1. It identifies & describes clearly the role of a MCS and detailed developments of 3 cold pools (CPO1,2,3) as sources for this unique dust evolution in time & space. 2. It describes an advanced attempt to provide a better model to simulate this dust event that had no precedence 3. It applies measured data close to the source and in Israel which is far from the main plume origin 4. Though the model describes fairly well the main features of the event the authors are not afraid to mention some discrepancies that were found in the details of dust spreading and attempt to provide an explanation. 5. An important physical contribution here: The mineral dust radiation interaction has been implemented in ICON-ART which seems necessary in this event.

However, it is suggested to add some remarks to the paper. In the period of 6-8.9.2015 at least 3 events of dust raising and spreading were evident. The first resulted from a rotating thermal low and the rest from a long red sea through. In Israel, the dust entered as an elevated plume. Then it was mixed downward affected both by the topography and local diurnal cycle of the land sea interaction. This fact is marked in section 3 and 3. 4 and 3.4.1 but we feel that there is a tendency to describe a black and white picture according to the model findings and the features of ground measurements alone. We suggest some modifications:

1. The authors seem to describe the dust penetration to Israel from east to west + a portion coming from the north through the Golan heights (in Sec. 3.4 lines 10 etc. and especially line 15 on). However, in Sede Boker (inland station) the AOT started rising from the 7th afternoon (Sede Boker Aronet). Therefore, we suggest that prior to line 28 (Sec. 3.4) the author will add a comment on the turbulent and complicated nature of the event and will explain that the following study described in 3.4.1 is an attempt to explain the discrepancies they found. They can quote the hour to hour SEVERI satellite showing the complicated patterns of the dust.

2. It is suggested to add the Sede boker Aeronet AOT finding and relate it to the Model AOD and add these results to Fig 8 in 3.4.1

3. It is suggested that the PM10 graph will be given in a logarithmic scale. This will show that part of the dust will be found on the 7th too.

4. The authors tend to explain Ashdod low measurements (fig 8) as a result of sea breeze penetration that brings in clean air. As can be seen from the measurements of the Tera and Aqua and the PM10 measurements of other sea stations the sea was full of dust. There were numerous differences between station to station in Israel even in short ranges a few km apart. So taking only 3 stations for verification cannot reveal the whole picture in such ac complicated event though they seem to fit with the model logic. Therefore, we suggest that fig 8 will be given as a description to the event complexity

without a "black and white description".

5. Fig. 1, caption, add time of satellite pictures.

---

## Author Comment (AC1) · 14 Jul 2017

**Reply to short comment SC1**

Philipp Gasch et al.

*Correspondence to:* philipp.gasch@kit.edu

*The paper is well written and the results are very exciting. The enormous potential of the ICON model coupled with the state-of-the-art aerosol module ART is documented in a fascinating way. Nevertheless, I was thinking some comments from my side should be given and may improve the paper a bit. As a co-author of the papers of Mamouri et al., ACP, 2016, and Solomos et al., 2017, both dealing with this record dust storm in September 2015, and, in addition, as a lidar expert having a long-term*
*cooperation with the lidar group at CUT, Limassol, I would like to recommend the following:*

Below we provide our answers and changes made to the manuscript in response to your comments.

*Q1) P1, L22: Please check the paper of Nisantzi et al., ACP, 2015, they report Saharan and Middle East lidar observations (2011-2014), perfomed in Cyprus, and provide statistical results. This article could be mentioned in the introduction.*

A1) Thank you for your detailed comments and references to existing literature. Our focus is on the transport mecha-
nisms during the September 2015 severe dust event. We did not find statistical results for frequency of dust transport sources or dust transport mechanisms in the mentioned paper beyond the list of dust cases they use for their analysis. They provide a detailed description of dust optical properties in dependence of source region, this is not the main focus of our study. However, as it is relevant to the optical properties section we have added a reference to the paper there with some surrounding discussion on the topic. Furthermore, we have extended our discussion of existing literature in
the introduction, please see also N2.

N1) Sec. 2.1: The spatially invariant mineral composition of dust in ICON-ART means we assume similarity to Saharan dust everywhere. Studies have shown that mineral dust optical properties can depend on the source region (Petzold et al., 2009), which presents a great uncertainty for the radiative forcing as discussed in Myhre and Stordal (2001). For our region of interest, Nisantzi et al. (2015) find differences in the dust particle lidar ratios in a comparison of dust from the Middle East and the
Sahara. Two problems prevent a more detailed description of the mineral dust optical properties for our study. First, there is a lack of observations of the refractive index for our dust source region and the variance within source regions can be considerable (Petzold et al., 2009). Second, to the best of our knowledge a dataset of the earth's crust mineralogical composition for our region is missing so far, making a more detailed availability of refractive indices futile. However, the influence of differences in the refractive indices is small compared to the influence of a varying size distribution (Myhre and Stordal, 2001) and this
latter effect is represented in ICON-ART.

*Q2) P2, L8-13: We need a short discussion on the existing literature for this September 2015 dust storm, i.e., a short discussion of Mamouri et al., ACP, 2016 and the companion paper of Solomos et al., ACP, 2017 (just finally published on 27 March). This is the normal 'way of life' in science, i.e., to discuss previous work, to discuss what is already known, and what will be the new points of the new article. I speculate that you (the authors) did not read the final version of the Mamouri et*

*al. paper with all the findings concerning mass loadings, dust height distributions, optical depths... Because there are so many useful observations and findings that corroborate your statements and findings..... The submitted ACPD version of Mamouri et al. is very different from the final one. By the way, in that paper, also the limits of MODIS concerning max AOT retrievals are discussed, and the quality of MODIS data at such high AOT conditions is discussed.*

A2) We have included a more detailed discussion of Mamouri et al. (2016) in the introduction and more references throughout the text, this was necessary indeed. In addition, we have included a comparison of ICON-ART with the Mamouri et al. (2016) lidar measurements, please see SC1 Q9 A9 N9. Furthermore, we have rewritten parts of the introduction in order to give a more detailed description of previous studies. We now also refer critically to Solomos et al. (2017) in multiple places, however, we believe Solomos et al. (2017) lacks crucial information necessary to
understand this event. Please see RC1 Q1 A1 for further details on this.

[revised manuscript text omitted]

*Q3) P2, L31: This final sentence of the paragraph has to be 'updated' because there is this Solomos et al. (2017) paper..... or what do you mean with...a detailed analysis of the driving atmospheric system...has not been published so far....? A mentuoned, the final version of the Solomos paper is now published.*

A3) We have adjusted our wording to take account of the Solomos et al. (2017) publication.

N3)  Solomos et al. (2017) model the event at convection permitting resolution, however, the spatial extent of their convection permitting domain does not cover the full MCS region. Consequently, their model fails to reproduce the observed CPO outflow structures and connected dust plumes realistically (see discussion in Sec. 3.1 and 3.2).

*Q4) P6, L10-11: just a short question: Why do you distinguish (always) sedimentation and dry deposition? I am not so familar with the terminology but to my opinion dry deposition includes gravitational settling. But maybe I am wrong.*

A4) Dry deposition includes gravitational settling (sedimentation) and deposition due to turbulent diffusion. We corrected our wording to clarify, that we meant deposition due to turbulent diffusion.

N4) Sec. 2 The processes which affect mineral dust number and/or mass concentrations in ART are  gravitational settling (sedimentation), deposition due to turbulent diffusion and wet deposition due to washout.

Sec. 2.1 Due to different processes such as  gravitational settling acting differently on the specific dust mass and number concentrations in ART, the diagnostic median diameter of each mode changes during transport (the standard deviation of each mode is kept constant).

*Q5) Page 9, Figure 3 is very nice, but needs to be improved.... It is almost impossible to identify Turkey, Cyprus, Israel etc....*

A5) We adapted the figure to include less information and better country outlines.

N5) Updated figure 3 included, please see also RC1 Q11 A11 N11.

*Q6) Page 15, Figure 6, color scales are missing, but needed. The MODIS analysis stops when the observations indicate: AOD > 5.0 (as written in Mamouri et al, ACP, 2016). In the MODIS figure (Figure 6, bottom, left) all dust regions, where the surface (overthe dark Med sea for example) is not visible anymore, are regions with AOD of 5.0 and more. You may check the MODIS data basis (links are given in the Mamouri et al, 2016 paper). And this in contradictions with the MODIS results in Fig.6 ... Did you compute these AOD values (map), instead of taking the AOD values from official MODIS data sources? What does the map (Figure 6, bottom, right) show? We need a color scale. And what about the region just east of Cyprus. The AOD is obviously very high (bottom, left, because the dark Mediterranean Sea is no longer visible, the AOD was rather high, probably 5.0 or even more), but no values in the MODIS AOD map (bottom, right). Impossible, to my opinion! Something is wrong with these MODIS products. Please check!*

A6) We have added the colourbars to the figure, although they are the same in every figure which is why we skipped them the first time and noted so in the description underneath. Thank you for noticing that by mistake we did not mark the regions where no MODIS data is available in a different colour than white (although we did say so in the text already: "Unfortunately, no measurements by MODIS are available over sea."). We changed this to grey in the new figures to make it clearly visible and further elaborated in the text. In addition we do not subtract 0.3 for the background aerosol concentration anymore to make it easier for the reader. As before, we continue to state that no data are available from MODIS AOD above the EM in an important section to the east of Cyprus. We do not think we are in a position to speculate why there is no data available in this region or how high the AOD might be. Luckily, we have enough data to compare ICON-ART to observations in the rest of the domain. Furthermore, we have included the maximum AOD values as measured by MODIS. On 07 September, the AOD is still well below the maximum value of 5 where MODIS stops working as you mention and we added a sentence noting the good agreement between MODIS and ICON-ART.

N6) Sec. 3.2: The maximum AOD value observed by MODIS is 2.78, compared to a dust optical depth (DOD) of 2.41 modelled by ICON-ART in good spatial agreement. Differences in the AOD distribution from MODIS and DOD from ART over the EM are attributable to background aerosol (e.g. sea salt, black carbon), which is not represented in our simulation but measured by MODIS. It should be noted that MODIS can suffer from a systematic bias for AODs > 2.5, resulting in an AOD overestimation in the range from 0.5 - 1.5 as shown by Mamouri et al. (2016) through a comparison of MODIS and AERONET data in the region. Our analysis contrasts the simulation results by Solomos et al. (2017, their Fig. 4c), who model AOD values above 20 already before the onset of strong downward mixing of momentum. Furthermore, their modelled bimodal maximum dust distribution was not observed by satellites and no closed cyclonic flow around the heat flow appears to have existed.

Sec. 3.3: There is good agreement in the maximum dust plume optical thickness, with MODIS AOD measurements giving an AOD of 3.71 and the simulation a DOD of 4.15, although again the possible overestimation by MODIS should be kept in mind (Mamouri et al., 2016). Modelled DOD is higher in ICON-ART when compared to MODIS AOD in the eastern part of CPO3 and lower in the western part of CPO3 (Fig. 5d). Unfortunately, no measurements by MODIS are available over the northern Mediterranean Sea where a substantial amount of dust is apparent in the visible satellite image. For the eastern part of CPO3 the MODIS AOD measurements seem doubtful when comparing to the MODIS visible satellite image.

Sec. 3.4: Dust transport into the southern EM is not simulated with the correct magnitude by ICON-ART despite the overall dust plume structure being captured, even when accounting for the MODIS AOD retrieval bias (Mamouri et al., 2016). MODIS measures AOD values consistently between $2-4$ over Israel, the Palestinian Territories and Jordan, and values above 5 over the southern EM. In this area, the contribution of the different plumes transported into the region along the Mediterranean coast from the north and across the Dead Sea Rift Valley from the north-east is especially complex due to the steep orography.

*Q7) Figure 7 (right panel): CALIOP obs, 34-36 N,.... The CALIOP retrieval gets lost at these conditions, the algorithm fails and cannot handle such situations. The dust extinction coefficients exceeded already 500 Mm-1at 3 km height.... and must be about 2000 Mm-1 or more at heights below 2 km to match the MODIS scence (Figure 6,bottom, left , AOD certainly larger than 5).*

A7) We agree (on page 16, line 10 in our discussion manuscript we say: "Altitudes below 2 km are marked as no signal regions in the CALIPSO feature mask due to the attenuation of the lidar signal (not shown).") As your statement agrees with this, we will keep it, thank you for supporting our point of view. As before, we continue to state that the MODIS AOD retrieval seems doubtful in the eastern part of CPO3 and therefore agree with your statement on MODIS as well. However, the measured AOD which is given by MODIS is only in the range of 1-2 (not above 5 as suggested in your comment). As this cannot explain the visible satellite picture, we keep our statement, thank you again for confirming (page 14, line 15: "For the eastern part of CPO3 the MODIS AOD measurements seem doubtful when comparing to the MODIS visible satellite image").

N7) Nothing changed.

*Q8) Page 17: I personally would like to see comparisons of ICON-ART results for the Cyprus region, for the 7-10 September period. But I am sure that huge deviations from our findings (presented in Mamouri et al., 2016) would become visible.*

A8) Our focus is on the onset of dust storm during $07$ and $08$ September. Unfortunately no direct observations are available to us for $08$ September from Cyprus. Thank you for the suggestion to compare our results with the Mamouri et al. (2016) observations, we have added a comparison of ICON-ART to the lidar observations to our manuscript. Due to the strong dust concentration gradient in the region and in order to compare the same air mass as measured by the lidar, please note that we are comparing the measurements to a $2°$ eastward shifted position in the model as the dust plume shows an approximate $2°$ eastward deviation. This is also noted and explained extensively in the text now. Considering the complex situation and the long $47$ hour forecast time without data assimilation we deem the mentioned offset of 2° and a 3 hour delay in dust plume arrival to be a very good representation of reality. The double layered plume structure is clearly visible without specifically tuning the model for these results or updating the soil and land-use datasets as was done in Solomos et al. (2017). In ICON-ART the dust plume reaches Cyprus from east, as is also confirmed by the EUMETSAT animation, not from south as stated by Solomos et al. (2017). Furthermore, the spatial agreement is much better around Cyprus (figure A4 and reply to RC1 Q1). The temporal evolution with a first, higher dust plume and a second lower, mightier and much denser dust plume is readily identifiable from model results (compare figure 4, Mamouri et al. 2016). An even more detailed discussion of this result would further increase the length of the manuscript and Cyprus is not our region of interest in this study, as we are interested in the complex dust transport towards the southern EM. If you have any further questions or would like to see more results please feel free to contact us directly.

N8) Sec. 3.4: A comparison of simulated  DOD with satellite observations for 11 UTC 08 September shows that the model represents the observed dust plume structure in the northern part of the EM (Fig. A6). The highest dust concentrations are present between Cyprus and Syria, although the dust plume has advanced approximately 2° further west in observations, reaching Cyprus. This shift can be explained by the northward deviation and less intense channelling of the CPO3 as well as the long forecast time. ICON-ART DOD values are one order of magnitude higher  and show better spatial agreement than other global dust forecast simulation results in the northern part of the EM (see Sec. 1 ). Taking into account the 2° longitudinal offset in ICON-ART, the vertical structure of the dust plume arrival represents observations by Mamouri et al. (2016) (Fig. A7). A first, elevated plume extending from $2-4$ km with concentrations up to $1000\,\mu g\,m^{-3}$ is noticeable during 07 September in both lidar measurements and model results. Mamouri et al. (2016) observe the arrival of the main dust plume past 19 UTC 07 September with concentrations up to $2000\,\mu g\,m^{-3}$ at $0.75-1.5$ km height. ICON-ART shows the dust plume arrival past 21 UTC 07 September with concentrations up to $3000\,\mu g\,m^{-3}$ at $0.5-2$ km height. During 08 September, dust concentrations increase up to $3500\,\mu g\,m^{-3}$ and the plume thickness grows further, extending from 0.5 km up to 3 km height in the model.

Conclusions: The transport to the northern part of the EM and Cyprus is modelled with DOD values above 2 and in good spatial agreement with satellite observations  at a 2° longitudinal offset towards the east. A comparison with lidar observations in Cyprus (Mamouri et al., 2016) shows very good agreement in vertical dust distribution.

*Q9) Page 19, Figure 8: Please check the Weizmann Institute AERONET station (a bit east of Tel Aviv) for 9 September (This station measured AOD of 2.4-2.7). What did you find for 9 September for Israel? On 8 September, the dust load was even higher, but there are no AERONET observations, because of too high AOD, which the AERONET algorithm misinterpreted as*

*clouds, I am speculating. That means, the modeled Jerusalem DOD values are much too low (by a factor 4...).*

A9) Our focus is on the onset of the dust storm during 06-08 September in order to investigate the previously not known generating meteorological drivers. Therefore, we did not simulate 09 September. Including 09 September would add further length to the study as dust deposition processes would have to be analysed in detail. This is beyond the scope of this work but promising and interesting for the future, especially in comparison with ceilometer observations in the region. As a reply to RC1 Q14 A14 we have added the AERONET station in Sede Boker, which measured on all days, to the results. We state multiple times that our simulations results for transport into the southern EM are not too low by a factor of 4 but on the order of one magnitude (e.g. page 18 line 9, page 19 line 3, page 24 line 12, page 29 line 6 in the discussion manuscript) due to the complex dust transport and emission processes taking place which we refer to and investigate in detail. We continue to do so, however, for Sede Boker the deviation is actually 'only' by a factor of 4.

N9) Please see RC1 Q14 A14 N14 and RC2 Q2 A2 N2.

---

## Author Comment (AC2) · 14 Jul 2017

**Reply to anonymous referee RC1**

Philipp Gasch et al.

*Correspondence to:* philipp.gasch@kit.edu

*The Manuscript entitled "An analysis of the September 2015 severe dust event in the Eastern Mediterranean" describes in great detail the mechanisms of the dust storm episode using the ICON model coupled with a desert dust module (ART). The paper is very well written and I would recommend it to be published on ACP after the following comments have been addressed:*

5    We would like to thank the referee for thoroughly reading our manuscript and providing meaningful questions. Below we provide our answers and changes made to the manuscript in response to your helpful comments.

*Q1) Abstract, Line 3: The authors state that "...state-of-the-art dust transport models were unable to forecast the event. . .". I don't think this is accurate. For example the publication of Solomos et al. (2017) in ACP describes the same episode using another model. I think that the two manuscripts were submitted very close to each other and the authors were not aware of*
10   *this, even though they give reference to this work when it was still in ACPD. Please correct this statement accordingly. This is present in other areas of the manuscript as well and should also be corrected.*

A1) Thank you for this comment, we think it needs a detailed discussion and clarification. By using the term state-of-the-art dust transport models we referred to the global dust forecast models which provide continuous dust forecasts on an everyday basis. From our point of view ICON-ART, being a global dust forecast model, is comparable to these
15   models with the added feature of allowing flexible grid refinements. The driver of our dust simulations is also a global ICON-ART dust forecast with 40 km grid spacing, which gives similar results to the other models when used without the local grid refinement. We have changed the text and now call them 'operational' in order to make our intended statement more clear. We have included references to the Solomos et al. (2017) publication and their modelling study, however, we would like to limit our references to the Solomos et al. (2017) publication in the main analysis section,
20   as our results are not comparable or even contradict theirs to a great extent. We think that our results can provide a much more detailed and accurate description compared to Solomos et al. (2017), which is necessary to understand this event. We would like to highlight some important differences which are crucial in our opinion, but avoid doing so in our manuscript as it will take up too much space there.

1. Solomos et al. (2017) do not use a global dust forecast model and they do not use their model in normal forecast
25   mode but assimilate additional radiosonde data. In addition, they tune their model and dust emission scheme without giving the specific settings ("RAMS–ICLAMS in this study is not used in forecasting mode, but rather as a tool for the a posteriori analysis and explanation of the event. This means that the configuration of several model parameters, such as the nested grid structure, convective parametrization schemes, and dust source strength, is

guided by the available observations.", Solomos et al. 2017 ). We do not assimilate any additional data or modify ICON-ART model physics for our simulation of the event, but use the same version as used for other studies.

2. They do not identify the synoptic situation as an active Red Sea trough situation which explains the highly unusual dust transport direction and is responsible for the unique character of the event due to its very early occurrence at the beginning of September.

3. Their highest resolution, convection permitting domain with 2 km grid spacing covers only the Turkey-Syria-Iran-Iraq border region (Fig. 1, Solomos et al. 2017), whereas our simulation covers the whole region at convection permitting resolution (please see the attachment to this reply, 1). Due to the large spatial extent of the meso-scale convective system, their domain is far too small to capture the full MCS development and its related flow structures (compare Fig. 6 in this manuscript). Because of this, they are unable to identify the different event stages and the individual, consecutive cold-pool outflows. Consequently they do not achieve a meaningful dust distribution, both in spatial coverage as well as magnitude, as a comparison with the satellite data provided in our work reveals. This results in them being unable to link the different dust plumes to the meteorological drivers correctly. They do not mention or investigate the transport direction of the main dust plumes (CPO2 and CPO3 in our work) towards the south-west across Syria, into Jordan and further onwards to the southern EM at all. We would like to describe some of the discrepancies which we found in more detail:

- They present simulated peak AODs above 20 next to large areas of AODs in the range of 1-3 (Fig. 4, Fig. 5, Fig. 12, Solomos et al. 2017). An AOD above 20 as well as its spatial extent are highly unrealistic when compared to satellite observations. AODs below 1 are not identifiable from their color scale, further complicating the analysis as in reality large areas were covered by AODs of less than 1. They do not compare their spatial model results to spatially quantifiable observations (e.g. MODIS AOD, MODIS VIS). The SEVIRI images shown by them are highly distorted map projections and cannot be translated into any quantitative measure, therefore they lack the information necessary to decipher the event.

- No bimodal maximum with AOD values greater 20 was observed in the dust plume structure around the heat low on 06 September 2015 above Syria and no 'closed cyclonic flow' existed as is stated by them (Fig. 4c, Solomos et al. 2017, compare Fig. A1 in this manuscript as well as EUMETSAT animation of the event, the observed maximum was 2.78 with a drastically different horizontal structure). As this presents the beginning of their simulation this result is problematic.

- Their simulated cold-pool spreads towards the north. As we show in this manuscript this is not the observed direction of travel (which is westward / south-westward) and the spatial cold-pool extent is far too small (compare Fig. 7b, Solomos et al. 2017). As cold-pools are the main drivers of the event, this finding is problematic.

- No widespread areas of AOD values greater 15 were observed over eastern Syria at 00 UTC 07 September 2015. In addition, the location of clouds is misrepresented in their results (over Syria, Iraq and Turkey in Solomos et al.

2017, compared to over Iraq, Iran in EUMETSAT observation. Compare Fig. 5a Solomos et al. 2017, and A2 in this manuscript, due to the distorted EUMETSAT picture this is disguised in their publication). They do not provide their modelled wind fields for the important event stage at 00 UTC 07 September (Fig. 5, Solomos et al. 2017). In addition, the structure of the dust plume apparent from CALIPSO is not visible in their model results. As this presents a crucial event stage, this finding is problematic.

- They neither provide an overview of the different cold-pool outflows contributing to the event nor do they decipher the timing of the cold-pools. They do not provide an analysis of the cold pool or dust plume structures during 07 September 2015, which is the main event day, nor do they provide a quantitative comparison on this day. The only plot they provide during 07 September 2015 is Fig. 10 (compare Fig. 6 in this manuscript). There, they compare a highly distorted (see direction of Eastern Mediterranean and position of Iraq), north-centred EUMETSAT image to a different Lambertian projection from RAMS which in addition does not display the AOD quantity shown before. In their analysis are lacking:

(a) An analysis of the dust plume structure with the marked features (compare Fig. 6 in this manuscript).

(b) A quantitative comparison of the spatial dust plume structure to MODIS VIS or AOD.

(c) Their modelled wind fields or a measure to identify the cold-pool outflow borders.

(d) A representation or comparison of the modelled MCS or cloud structure in general.

The above items are crucial for the understanding of this event. The comparison to CALIPSO in Fig. 10c and d further highlights the problems of their simulation. The maximum dust concentrations are achieved north of 36° N in Solomos et al. (2017), where in reality a clear local minimum was observed, this is both visible in the map display as well as the vertical cross-section (compare Fig. 6 and Fig. 7 in this manuscript).

- On 08 September the spatial dust plume distribution around Cyprus does not represent observations with AOD values above 10 next to large regions of 1 (Fig. 12 in Solomos et al. 2017). The Solomos et al. (2017) local minimum with AOD values of 1 to the east of Cyprus was actually where the maximum was observed (see A4 this manuscript). Furthermore, there is an increase to unrealistic AOD values above 10 towards Syria and Lebanon, which was also not observed. From the EUMETSAT animation, MODIS VIS and ICON-ART model results it is clear that the dust transport direction towards Cyprus was not from the south as stated twice in the publication but from the east. There is no discussion of this problematic model result given in the publication; on the contrary the simulation results are compared to measurements while saying that the model captures the situation correctly. A time series comparison between model and station measurements for the temporal evolution of e.g. PM10 is lacking, even though measurements are available to the authors in Cyprus (Mamouri et al., 2016).

4. Solomos et al. (2017) provide physical explanations of the mechanisms leading to the cold pool development which appear unrealistic. The temperature difference between rain droplets and ambient air is not known as dominant driver of cold-pool formation in literature (Knippertz et al., 2009; Marticorena, 2014) and it seems

unlikely that this could be the case as is also noted by one of their reviewers. What can strengthen cold-pool formation is the descent of frozen hydro-meteors into sub-saturated air masses (besides a deep, dry-adiabatic mixed layer, high rain water mixing ratios and small raindrop sizes), however, Solomos et al. (2017) do not investigate this. In addition, droplet- ambient air temperatures of -20°C exist in regions where no clouds were observed (e.g EM) ("A number of atmospheric parameters that determine the formation of the cold pool are shown in Fig. 7a–d. As seen in Fig. 7a, the iso-temperature line of -20 °C between rain droplets and ambient air temperature clearly defines the cold pool area.", Solomos et al. 2017).

In comparison, ICON-ART shows a much better agreement spatially, temporally and magnitude wise. It is able to reproduce all major structures observed by satellite in great detail. We do not think that our simulation results are the best possible result which can be achieved and we know that further improvements will be made in the future, especially with respect to the magnitude of dust emission and concentrations. However, our simulation captures important meteorological event stages and drivers leading to its unique character and structure which is proven by the extensive comparison with measurement data. By doing so, we are able to provide a detailed description of the event course, its stages and responsible drivers. Concluding from the points listed above, we do not think that Solomos et al. (2017) were able to achieve this and we would like to limit our references as we think that there is very little agreement between the studies. In order to make the reader aware of these discrepancies, we added several critical references to the work of Solomos et al. (2017). As before, we continue to refer to Solomos et al. (2017) for their coherent analysis of the Mesopotamia soil degradation over the past decade.

N1)

Sec. 1: Adding to the extraordinariness,  operational global dust transport models were unable to forecast the event as  also noted by Mamouri et al. (2016).  All predictions provided through the World Meteorological Organization's Sand and Dust Storm Warning Advisory and Assessment System (SDS-WAS, http://sds-was.aemet.es)  initialised at 12 UTC 07 September failed to simulate significant dust concentrations in the EM region for 12 UTC 08 September. The simulated values of the dust optical depth in the EM are between $0.1 - 0.4$ in the multi-model mean with a standard deviation of $0.1 - 0.2$. The forecast failure is highly problematic due to the severe impact of the event.

Sec. 1:  Solomos et al. (2017) model the event at convection permitting resolution, however, the spatial extent of their convection permitting domain is chosen too small and does not cover the full MCS region. Consequently, their model fails to reproduce the observed CPO outflow structures and connected dust plumes correctly (see discussion in Sec. 3.2 and 3.3).

Sec. 3:  For the global grid, ICON-ART  produces results comparable to those from other global models. However, due to its flexible nesting capability, it allows for convection permitting simulations for the finest resolution. As is shown in this section, ICON-ART is thereby able to resolve the meteorological drivers of the event in great detail. The results show that the dust event consists of multiple stages and is created by the interaction of different meteorological systems.

Sec. 3.2: Our analysis contrasts the simulation results by Solomos et al. (2017, their Fig. 4c), who model AOD values above 20 already before the onset of strong downward mixing of momentum. Furthermore, their modelled bimodal maximum dust distribution was not observed by satellites and no closed cyclonic flow around the heat flow appears to have existed.

Sec. 3.2: (a The above findings again contradict those of Solomos et al. (2017, their Fig. 7b), who in their model results find

5 a northward travel direction of a small cold-pool structure. Based on the good agreement between ICON-ART and satellite observations this result is implausible. Furthermore, the intensity and spatial extent of their modelled CPO is much too small.

Sec. 3.3 When comparing our results to those of Solomos et al. (2017, their Fig. 10), again large differences become apparent, both in spatial dust plume structure as well as vertical dust distribution. The driving MCS and related CPOs and their clearly marked borders discussed above are not identifiable in their model results and they do not provide a detailed analysis of the

10 flow structures.

*Q2) Page 3, Line 20: The 3rd research question posed by the authors is "What are the meteorological drivers responsible for pick-up and long-range transport of mineral dust?". It is not clear of they mean in general or in cases such as the event described in the manuscript, because the dust-cycle mechanisms in general are well known and documented. Please rephrase*

A2) We rephrased the sentences to be

15 N2) (3) What are the meteorological drivers responsible for pick-up and long-range transport of mineral dust during this event?

*Q3) Page 4, Line 15: You state that ". . .the seamless modelling capabilities of ICON are of crucial importance because inconsistencies in tracer transport and tracer physics at the nest boundaries can be avoided. . .". Please remove the word "seamless". Also I cannot understand how these inconsistencies are avoided. Please expand.*

20 A3) We removed the word seamless and rephrased the paragraph to explain in more detail which inconsistencies we are talking about.

N3) For the tracer transport simulations the seamless modelling capabilities of ICON are of crucial importance because the same physical parametrization packages can be used from a global to regional scale. Thereby, inconsistencies in tracer transport and tracer physics at the nest boundaries concentrations arising from differences in tracer advection and physical

25 parametrizations between the driving model and the high-resolution model can be avoided, which is a major problem for other modelling systems.

*Q4) Page 4, Line 22: What do you mean by sedimentation? Does it refer to the sandblasting mechanism for production or the deposition of particles?*

A4) By Sedimentation we mean gravitational settling. We have modified it accordingly.

30 N4) The processes which affect mineral dust number and/or mass concentrations in ART are sedimentation, dry deposition gravitational settling (sedimentation), deposition due to turbulent diffusion and wet deposition due to washout.

*Q5) Page 5, Line 9: It is important to see how the model defines the dust sources between the nests. Are they defined separately for each domain? If so how do you assure there are no continuity problems in the fields? Is this what you mean in Page 4, Line 15 (see comment 3)?*

A5) We are using a physical parametrization for mineral dust emission which depends on different external datasets. As now stated above (see our answer to comment Q3, A3), this parametrization is the same for each domain. The horizontal grid on which the information from the external datasets is aggregated to varies from nest to nest which is accounted for by the tile approach we use. Additionally, the meteorological input parameters for the parametrizations (like friction velocity, soil moisture content) can show differences due to differences in the horizontal resolution. The different representation of for example convective systems at different resolutions can therefore also lead to different emission fluxes. As we have seen the best agreement of meteorological parameters at convection-permitting resolution, we decided to focus on these results and added a sentence stating so.

N5) In the following, a detailed analysis of the development stages and responsible atmospheric drivers, which lead to the severe dust event, is provided. We focus on the results from the convection permitting domain, as it yields remarkable improvements compared to the global domain.

Q6) Page 5, Line 17: You state the timestep for calling RRTM is 288 seconds. Please provide the timestep of the simulation as well.

A6) We moved this statement to the 'Model set-up' section and added the other model timesteps.

N6) In our setup for the finest resolution, the advection/fast physics time step is 18 seconds with a sub-stepping of the dynamics at 4.5 seconds. RRTM is called every 288 seconds for the finest resolution.

Q7) Page 5, Line 18: ". . .ART modifies the radiative transfer parameters of the climatological dust distribution. . .". What do you mean by "climatological"? Do you mean the dust distribution as described by the dust module? Please expand.

A7) Thank you, this section needed revision. Please see our answer to Q8.

N7) Please see N8).

Q8) The same confusion in Line 20: "The parameters returned by ART are the combined values from the local ART dust concentration plus the Tegen climatology". During the simulation dust concentration is calculated using both prognostic dust and climatological values?.

A8) What we tried to say is the following: We include only the mineral dust on-line radiative feedback in ART as a part of this study. The radiative effect of other aerosol species, such as sea salt or stratospheric aerosol, is non-negligible but not simulated on-line by ART as a part of this study. Therefore, for these aerosol species the constant climatological values from ICON are used. However, for different studies, ART is capable of simulating these species.

N8) Without ART, ICON uses a climatological distribution of aerosols (e.g. mineral dust, sea salt, stratospheric aerosol) to include their radiative effect. When using ART, any of these aerosol species can be calculated on-line and therefore its radiative effect can be included with much better accuracy. For aerosol species not simulated by ART the climatological values are still used and taken from ICON. Therefore, the  radiative transfer parameters  provided by ART to the RRTM are combined values from the local ART aerosol concentration plus the ICON climatology, which is used only for the aerosol species not simulated. For example, in our study we simulated mineral dust using ART, and therefore can include the on-line mineral dust radiative feedback. For the sea salt and stratospheric aerosol radiative effect, however, the climatological values from ICON are used.

The radiative transfers parameters needed consist of the optical depth, single scattering albedo and asymmetry parameter . In order to obtain the on-line mineral dust radiative feedback, the local radiative transfer parameters are calculated using the dust optical properties  and the local dust mass concentration at every grid-point and for every level as detailed in Stanelle et al. (2010). The

5  radiative transfer parameters are calculated in ART and provided to the RRTM, where they feedback on the atmospheric state in ICON.

*Q9) Page 6, Line 14: "Therefore, the median diameter of each mode is expected to decrease during transport". You mean that during the simulation the size of the particles changes? Please expand a little as this is very interesting.*

A9) In ICON-ART, dust is described through collections of particles, the modes. For each mode the integral values of
10 specific number and mass are the prognostic variables. The distribution of specific number and mass with particle size during transport is described using log-normal distributions for each mode with the diagnostic median diameter of the mass distribution and constant geometric standard deviation as parameters (Mode A, d = 1.5 $\mu$m, $\sigma = 1.7$; Mode B, d = 6.7 $\mu$m, $\sigma = 1.6$; Mode C, d = 14.2 $\mu$m, $\sigma = 1.5$). During transport, specific number and mass can change independently from each other, resulting in a median diameter change. In more detail: As the standard deviation is kept constant during
15 transport the diagnosed median diameters of number $\overline{d}_{0,l}$ and mass concentration $\overline{d}_{3,l}$ are always directly related to each other through

$$\ln\overline{d}_{3,l} = \ln\overline{d}_{0,l} + 3 \cdot \ln^2 \sigma_l.$$

The median diameter can be diagnosed from the prognostic variables using

$$\overline{d}_{0,l} = \sqrt[3]{\frac{\widehat{\Psi_{3,l}}}{\frac{\pi}{6}\rho_p exp(\frac{9}{2}\ln^2\sigma_l)\widehat{\Psi_{0,l}}}},$$

20 where $\rho_p$ denotes the density of the mineral dust particles given as 2500 kg m$^{-3}$ and the rest of the quantities is given as per our manuscript. The prognostic variables of specific number and mass mixing ratio can develop independently from each other as some parametrizations influence the distributions differently, e.g. the sedimentation velocities for number and mass distributions are different. This is why specific number concentration and mixing ratio are both prognostic variables in ART, leading to a change in median diameter during transport as at all times both distributions are linked
25 through the above equation. For example, sedimentation of mineral dust is included through simulating a constantly downward directed vertical advection with a size- dependent sedimentation velocity. Because larger particles have a greater sedimentation velocity and therefore settle faster, the size distribution shifts towards smaller particles during transport. We included some changes as part of reply to SC1 Q4 A4, but did not include the above description as we think it is rather technical.

30 N9) The processes which affect mineral dust number and/or mass concentrations in ART are  gravitational settling (sedimentation), deposition due to turbulent diffusion and wet deposition due to washout.

*Q10) Page 7 Line 29: ". . .which represents the soil moisture conditions in the region more realistically...". Do you have a reference or actual data to support that this method provides more realistic values? This is essential as soil moisture dictates*

*dust production. How can you be sure that the underestimation of the dust concentration by the model (as described in later chapters) is not attributed to false soil moisture?*

A10) No measurements are available to confirm this, our assumption was based on the observation that after a hot and dry summer no precipitation occurred previous to the event. Therefore, we decided to adjust the soil moisture to the surrounding conditions. By this, we reduce the correction applied to dust emission due to soil moisture to a minimum because we want to make sure it has little influence in order to exclude this as a possible reason for underestimation as you mention. However, other factors such as soil type and land cover information are certainly of great importance for dust emission and possibly misrepresented due to the lack of data in the region, this is discussed in the section 'Model setup'. We have adjusted the respective passage by excluding the words 'unrealistic' and 'more realistically' and rephrasing.

N10) The IFS initialization data for soil moisture was modified in a region along the Syrian-Iraqi border which showed  high soil moisture values and spatial  inhomogeneities without preceding rain or changes in soil properties. Therefore, in a region from $37.5°N - 41.5°N$ and $32.5°E - 35°E$ the soil moisture index in the four layers provided by the IFS is set to the average value of the region between $36.5°N - 38°N$ and $32°E - 34°E$. This is done in order to prevent a possible effect of soil moisture on dust emission in this region, where dust emission is likely to be under-estimated due to the recent changes in land use conditions (see Sec. 2). The region modified is an important dust source region and emission fluxes for mineral dust increased due to the reduction of the soil moisture content.

*Q11) Page 9, Figure 3: I would like to see clearer plots, especially the national borders as to know exactly where they refer to. Maybe resizing them?*

A11) We reduced the plot information content, there is no more display of geopotential height and surface pressure as this was not discussed in the text. Additionally national and continent borders are now thicker and the figure is resized.

N11) Included new figure 3.

Synoptic situation on 06 September at 18 UTC as simulated by ICON-ART for the global domain. Shown are  the a) 300 hPa, b) 600 hPa and c) 900 hPa level.  Black lines denote the height of the respective pressure level in geopotential metres.  Wind speed is colour coded and wind velocity is shown as vectors.

*Q12) Section 3: Since this is a very detailed description of the event it would be very interesting to see a vertical cross-section of dust concentration and precipitation in the same plot, for different forecast hours. Like what you have in Figures A1-A4, but vertically.*

A12) Thank you for this suggestion, we have included cross-sections for every point in time discussed in our work in the appendix. There are four new cross-sections which we have chosen, both the CALIPSO overpass tracks (Fig. A7) and two additional ones to provide insights into the east-west event structure and dust transport (Fig. A6). The first new cross-section runs from $35°N$ $32°E$ to $35°N$ $46°E$ along the $35°N$ circle of latitude, thereby providing insight into

the east-west transport over Syria towards the northern EM. The second new cross-section runs from $32°N$ $34°E$ to $38°N$ $46°E$ along the main south-west dust plume travel direction (an extension of our previously existing Golan height cross-section), thereby providing insight into dust transport towards the southern EM. Besides the already discussed plume structure due to different meteorological drivers, the cross-sections reveal that during nighttime the cold-pool outflow is confined to a shallow layer of approximately 1 km close to the surface with wind speeds above $20$ m s$^{-1}$. Dust concentrations are highest in the lowest hundred meters, due to a lack of turbulent mixing the dust plume does not cover the full cold-pool outflow depth. With sunrise wind speeds are reduced and the dust plume has a greater vertical depth as turbulent mixing increases.

N12) Sec. 3.2: In addition, there are four cross-sections through ICON-ART results along which the event evolution and vertical structure can be tracked for all points in time discussed in this paper. The first cross-section runs from $35°N$ $32°E$ to $35°N$ $46°E$ along the $35°N$ circle of latitude, thereby providing insight into the east-west transport over Syria towards the northern EM (Fig. A4, left). The second cross-section runs from $32°N$ $34°E$ to $38°N$ $46°E$ along the main south-westward dust plume travel direction, thereby providing insight into dust transport towards the southern EM (Fig. A4, right). The third and fourth cross-sections run along both the CALIPSO overpass tracks (Fig. A5).

Sec 3.2: The night-time spread of the CPO3 towards the west is crucial due to its subsequent interaction with the developing boundary layer mixing during daytime. During night-time the CPO is confined to a shallow layer of approximately 1 km close to the surface with wind speeds above $20$ m s$^{-1}$ (Fig. A4, A5). Due to the stable stratification the dust plume does not cover the full CPO depth and dust concentrations are highest in the lowest hundred meters. With sunrise downward mixing of momentum increases. This leads to an increase of dust emissions and a greater dust plume depth which subsequently extends throughout the full CPO.

*Q13) Figures 6 and A1-A4. Please add labelbars to the plots (where applicable of course).*

A13) We added labelbars below the plot valid for all plots as ART DOD and MODIS AOD use the same colours.

N13) Included labelbars.

*Q14) Section 3.4.1: I would like to see a comparison for Dust Optical Depth from more stations in the computational domain. Is this possible? Maybe using AERONET data? If there is no additional data available then add a sentence in the text stating that.*

A14) We have included the AERONET station in Sede Boker, Israel for comparison in Fig. 8 (see also RC2, Q2 A2 N2, Q4 A4 N4). Unfortunately, this is the only station which measured in the region on $07$ and $08$ September. Due to technical problems (personal communication) Rehovot, Israel only became active again on $09$ September, a day which is not part of our analysis as it would greatly inflate the length of the paper. As a side effect (following our reply to RC 2, Q4 A4 N4 which deals with the whole section) we have also replaced the PM10 measurements for Ashdod in the plot and show the ones for Beer-Sheva, which is close to Sede Boker, instead.

N14) Sec. 3.4.1:  A comparison of modelled DOD and AERONET AOD measurements shows a similar development of the optical depth for 07 September, although with an offset of 0.3. The offset is explainable by AERONET measuring the optical depth due to all aerosol species, whereas we only display DOD from ICON-ART, as well as a possible underestimation of dust background concentration in the model. Nevertheless, the main signal appears to be shaped by mineral dust processes captured by ICON-ART. The maximum modelled DOD for Sede Boker is 1.0 on 08 September, compared to 4.1 measured by AERONET. The AERONET values appear realistic, as they are in good agreement with MODIS AOD measurements in the region. Thus, ICON-ART shows an underestimation of DOD by a factor of four.

*Q15) The authors show that the ICON-ART underestimated dust concentrations and give a very thorough explanation as to why this happens. However how can you be sure that this substantial difference is not caused by something simpler like wrong description of the strength of the dust source areas (maybe in reality the areas are more active than described in the model) or, as I stated above, bad definition of the soil moisture? Have you tried some sensitivity runs based on these?*

A15) This is a good point. Our focus was not on simulating the event correctly with respect to dust magnitude in every detail, but first to understand the event and its meteorological drivers, their course and timing. We have not conducted sensitivity runs, as we think that a deeper analysis of different model settings and tuning parameters would be beyond the scope of this manuscript and increase its size further. From our point of view, important parameters which should be analysed in the future are the soil conditions in the dust source region. As stated above, we hope that we limited the influence of soil moisture to a minimum. We would like to emphasize, that the emission of dust in connection with the hydraulic jump should be seen as an additional and interesting feature. We agree, one reason for the underestimation of dust concentrations in ICON-ART is probably connected to the description of soil properties in the active dust source areas in the Mesopotamia region. However, in this part ICON-ART does a fairly good job when compared to satellite observations, and the main differences on the order of one magnitude only become apparent after the dust plume transects the Dead Sea Rift Valley. Therefore, we assume that the emission of mineral dust in connection with the hydraulic jump does play an important role indeed. However, it is certainly below the 90 % by which ICON-ART underestimates dust concentrations in some parts of the southern EM and we have added statements in our manuscript referring to the importance of soil conditions.

N15) Sec. 3.4: Summarizing, the existence of super-critical flow conditions in the region with connected hydraulic jumps is assumed to cause widespread and strong dust emissions on the eastern side of the Dead Sea Rift Valley. This  contributes to the exceptional amount of dust in the southern part of the EM on 08 September. ICON-ART captures the special flow phenomena, albeit not with the correct magnitude and timing. The lack of a sufficiently developed super-critical flow and resulting high near-surface wind speeds prevents dust emission in Jordan and Israel in the model.

In combination with already underestimated dust emissions due to the recent land cover changes and soil degradation in the Mesopotamia region (see Sec. 2 and 3.2), this provides an explanation why dust transport into the southern EM is underestimated by an order of magnitude. Nevertheless, ICON-ART provides  a detailed understanding of previously unknown processes contributing to the historic dust event which makes these findings worthwhile to report.

Conclusions:

For the transport to the southern EM, a hydraulic jump is demonstrated to be of  importance for dust emission in addition to the advection of the dense dust plumes into the region. It is captured by ICON-ART, albeit with reduced

5  intensity compared to observations.  Due to the out-of-date soil conditions in the Mesopotamia dust source region and an underdeveloped hydraulic jump phenomena, dust transport into the  southern EM is underestimated by one order of magnitude by ICON-ART. Modelled DODs are in the range of $0.5 - 1.5$ over Israel and PM10 concentrations reach up to $600$ $\mu$g m$^{-3}$ in Jerusalem. Nevertheless, the characteristic dust transport features are captured. The arrival of the main dust plume during the night of $08$ September is simulated  at

10  1 km height and subsequent downward mixing increases surface dust concentrations.

*Q16) Page 28, Line 21: "...this study presents the first successful simulation of the September 2015 severe dust event. . .". See point (1) in my review.*

A16) We have rephrased our wording.

15  N16)  This study presents a successful simulation of the September 2015 severe dust event at convection permitting resolution.

*Q17) Page 29, Line 10: ". . .ICON-ART results are one order of magnitude better than those from other models. . .". This statement is rather odd since you do not present the capabilities of other models in the manuscript to support this. Of course this does not reduce the very good performance of your model in any way.*

20  A17) We again referred to the global dust forecast models which provide continuous dust forecasts on an everyday basis and have altered the sentence in this sense (their results are available from the World Meteorological Organization dust forecast comparison (http://sds-was.aemet.es), see Q1, A1). On a side note, ICON-ART is also one order of magnitude closer to the observations and has the correct spatial distribution compared to Solomos et al. (2017), which overestimate the observed dust optical depth (AOD above 25 instead of the observed value of 2 above Syria on $06$ September) and is

25  unable to represent the spatial distribution (aforementioned two peak distribution around heat low).

N17) Sec. 1: Adding to the extraordinariness,  operational global dust transport models were unable to forecast the event as  also noted by Mamouri et al. (2016).  All predictions provided through the World Meteorological Organization's Sand and Dust Storm Warning

30  Advisory and Assessment System (SDS-WAS, http://sds-was.aemet.es)  initialised at 12 UTC 07 September failed to simulate significant dust concentrations in the EM region for 12 UTC 08 September. The simulated values of the dust optical depth in the EM are between $0.1 - 0.4$ in the multi-model mean with a standard deviation of $0.1 - 0.2$. The forecast failure is highly problematic due to the severe impact of the event.

Conclusions: As the meteorological drivers are captured in detail, it is expected that the remaining underestimation of dust concentration is attributable to an out-of-date description of soil properties in the region due to the on-going conflict.

*Q18) Finally, as far as I can tell, the model does not support the indirect effect of dust particles (cloud and precipitation). Does this affect the performance in this particular case? If the model was able to simulate dust acting as CCN would the results be any better? Just add a small paragraph expanding on this.*

A18) For the cloud micro-physical processes a the two-moment cloud scheme is used (Seifert and Beheng, 2006). The two-moment scheme utilizes a parametrization developed by Seifert and Beheng (2001) which predicts number and mass concentrations for six different hydro-meteor species. These are cloud droplets, rain drops, cloud ice, snow, graupel and hail. For this parametrization an extension was developed by Rieger (2016), which includes the aerosol effect on cloud formation through using the current, local aerosol mass and number concentrations from ART. It is based on parametrizations published by Phillips et al. (2013) for the heterogeneous ice nucleation spectrum and by Barahona and Nenes (2009) for the cirrus regime with competition between homo- and heterogeneous freezing. The parametrization is not used in this study, as the inclusion of aerosol - cloud microphysics interaction creates a new set of research questions and the focus in this study is on the mineral dust radiation interaction. The combined effects of the mineral dust radiation and cloud microphysics interaction are investigated and quantified in a separate publication for a different event (Rieger et al., 2017). As a side note, trials have been conducted for this study as well, the results did not show any marked differences with a clear sign and therefore were stopped again in order to focus on the mineral dust radiation interaction. The most visible result of including the aerosol - cloud microphysics interaction is a change in the structure of the meso-scale convective system with altered rainfall positions. However, much larger differences results from usage of 1-moment versus 2-moment cloud microphysics scheme. Nevertheless, for further investigations the inclusion of aerosol - cloud microphysics interactions would be interesting.

N18) Sec. 2.2: For the cloud micro-physical processes a the two-moment cloud scheme is used (Seifert and Beheng, 2006), as this was found to lead to more realistic features of the meso-scale organized convection. The two-moment scheme utilizes a parametrization developed by Seifert and Beheng (2001) which predicts number and mass concentrations for six different hydro-meteor species. These are cloud droplets, rain drops, cloud ice, snow, graupel and hail. For this parametrization an extension was developed by Rieger (2016), which includes the aerosol effect on cloud formation through using the current, local aerosol mass and number concentrations from ART. The aerosol - cloud microphysics interaction is not included in this study as it creates a new set of research questions and the focus in this study is on the mineral dust radiation interaction. The combined effects of the mineral dust radiation interaction and its impact on cloud microphysics are investigated and quantified in a separate publication for a different event (Rieger et al., 2017).

C) Please note the additional changes which we have included in the paper as part of our improvements, most of them concern the English phrasing of sentences, figure labelling or units. More substantial ones are listed below.

C1) We changed the title to be:

 Revealing the meteorological drivers of the September 2015 severe dust event in the Eastern Mediterranean

C2) The dust sources are located in northeastern Africa, not northwestern Africa as stated before:

As a result, the most important remote dust source regions for the EM are situated in  northeastern Africa and the southern Arabian peninsula (Ganor, 1991; Kubilay et al., 2000).

C3) We retracted a statement in the introduction, as it is valid above oceans which is not the case in our study:

C4) We have added a sentence on the need for further research on the radiative effects in CPOs in Sec. 3.5:

A more intense and faster spreading CPO can have multiple reasons and further research is necessary in order to quantify the different contributions

C5) Added statement in the conclusions referring to questions posted in the beginning, in addition changed question heading to be non-bold:

Summarizing, we are able to answer the research questions presented in the beginning as follows

C6) We have added a statement on the outlook and overarching implications of this study to the conclusions:

In conclusion, this comprehensive case study has demonstrated the need to explicitly represent deep moist convection in dust storm forecasting. While Pantillon et al. (2016) propose a simple parametrisation to represent the climatological effects of haboobs in coarser resolution models, the forecasting of severe events like the one investigated here can hardly be successful without explicit convection. Given the substantial impact of the event and the potential benefit of an early warning, forecasting centres around the world should consider running higher-resolution dust forecasts for the most vulnerable regions. More research is also needed into the multi-scale interactions between Red Sea troughs, heat lows and convection. Moreover, the role of hydraulic jumps for dust emission and transport in the Dead Sea valley appears an interesting subject for further study, which would greatly benefit from a denser observational network.

**1   Figures**

[Figure]

**Figure 1.** Model domain comparison. Marked in red is the ICON-ART convection permitting domain and as a black frame the section used for our analysis plots. The white frame shows the extent of the RAMS convection permitting model domain used by Solomos et al. (2017).

**References**

Barahona, D. and Nenes, A.: Parameterizing the competition between homogeneous and heterogeneous freezing in ice cloud formation–polydisperse ice nuclei, Atmos. Chem. Phys., 9, 5933–5948, 2009.

Ganor, E.: The composition of clay minerals transported to Israel as indicators of Saharan dust emission, Atmospheric Environment. Part A. General Topics, 25, 2657–2664, 1991.

Knippertz, P., Ansmann, A., Althausen, D., Müller, D., Tesche, M., Bierwirth, E., Dinter, T., Müller, T., VON HOYNINGEN-HUENE, W., Schepanski, K., et al.: Dust mobilization and transport in the northern Sahara during SAMUM 2006–a meteorological overview, Tellus B, 61, 12–31, 2009.

Kubilay, N., Nickovic, S., Moulin, C., and Dulac, F.: An illustration of the transport and deposition of mineral dust onto the eastern Mediterranean, Atmospheric Environment, 34, 1293–1303, 2000.

Mamouri, R.-E., Ansmann, A., Nisantzi, A., Solomos, S., Kallos, G., and Hadjimitsis, D. G.: Extreme dust storm over the eastern Mediterranean in September 2015: satellite, lidar, and surface observations in the Cyprus region, Atmospheric Chemistry and Physics, 16, 13 711–13 724, doi:10.5194/acp-16-13711-2016, http://www.atmos-chem-phys.net/16/13711/2016/, 2016.

Marticorena, B.: Dust Production Mechanisms, in: Mineral Dust, edited by Knippertz, P. and Stuut, J.-B. W., chap. 6, pp. 93–120, Springer, Dordrecht, Netherlands, 2014.

Pantillon, F., Knippertz, P., Marsham, J. H., Panitz, H.-J., and Bischoff-Gauss, I.: Modeling haboob dust storms in large-scale weather and climate models, J. Geophys. Res.-Atmos., 121, 2090–2109, 2016.

Phillips, V. T., Demott, P. J., Andronache, C., Pratt, K. A., Prather, K. A., Subramanian, R., and Twohy, C.: Improvements to an empirical parameterization of heterogeneous ice nucleation and its comparison with observations, J. Atmos. Sci., 70, 378–409, 2013.

Rieger, D.: Der Einfluss von natürlichem Aerosol auf Wolken, Ph.D. thesis, Karlsruhe Institute of Technology, Karlsruhe, 2016.

Rieger, D., Steiner, A., Bachmann, V., Gasch, P., Förstner, J., Deetz, K., Vogel, B., and Vogel, H.: Impact of the 4 April 2014 Saharan dust outbreak on the photovoltaic power generation in Germany, Atmospheric Chemistry and Physics Discussions, 2017, 1–31, doi:10.5194/acp-2017-441, http://www.atmos-chem-phys-discuss.net/acp-2017-441/, 2017.

Seifert, A. and Beheng, K.: A two-moment cloud microphysics parameterization for mixed-phase clouds. Part 1: Model description, Meteorol. Atmos. Phys., 92, 45–66, 2006.

Seifert, A. and Beheng, K. D.: A double-moment parameterization for simulating autoconversion, accretion and selfcollection, Atmos. Res., 59, 265–281, 2001.

Solomos, S., Ansmann, A., Mamouri, R.-E., Binietoglou, I., Patlakas, P., Marinou, E., and Amiridis, V.: Remote sensing and modelling analysis of the extreme dust storm hitting the Middle East and eastern Mediterranean in September 2015, Atmospheric Chemistry and Physics, 17, 4063–4079, 2017.

Stanelle, T., Vogel, B., Vogel, H., Bäumer, D., and Kottmeier, C.: Feedback between dust particles and atmospheric processes over West Africa during dust episodes in March 2006 and June 2007, Atmos. Chem. Phys., 10, 10 771–10 788, 2010.

---

## Author Comment (AC3) · 14 Jul 2017

**Reply to anonymous referee RC2**

Philipp Gasch et al.

*Correspondence to:* philipp.gasch@kit.edu

*We recommend the paper publication with some revision outlined below. Nice research in this article includes: 1. It identifies & describes clearly the role of a MCS and detailed developments of 3 cold pools (CPO1,2,3) as sources for this unique dust evolution in time & space. 2. It describes an advanced attempt to provide a better model to simulate this dust event that had no precedence 3. It applies measured data close to the source and in Israel which is far from the main plume origin 4. Though the*

5   *model describes fairly well the main features of the event the authors are not afraid to mention some discrepancies that were found in the details of dust spreading and attempt to provide an explanation. 5. An important physical contribution here: The mineral dust radiation interaction has been implemented in ICON-ART which seems necessary in this event.*

*However, it is suggested to add some remarks to the paper. In the period of 6-8.9.2015 at least 3 events of dust raising and spreading were evident. The first resulted from a rotating thermal low and the rest from a long red sea through. In Israel, the*

10   *dust entered as an elevated plume. Then it was mixed downward affected both by the topography and local diurnal cycle of the land sea interaction. This fact is marked in section 3 and 3. 4 and 3.4.1 but we feel that there is a tendency to describe a black and white picture according to the model findings and the features of ground measurements alone. We suggest some modifications:*

We would like to thank the referee for the comments related to the dust transport towards the southern EM. Below we

15   provide our answers and changes made to the manuscript in response.

*Q1) The authors seem to describe the dust penetration to Israel from east to west + a portion coming from the north through the Golan heights (in Sec. 3.4 lines 10 etc. and especially line 15 on). However, in Sede Boker (inland station) the AOT started rising from the 7th afternoon (Sede Boker Aronet). Therefore, we suggest that prior to line 28 (Sec. 3.4) the author will add a comment on the turbulent and complicated nature of the event and will explain that the following study described in 3.4.1 is*

20   *an attempt to explain the discrepancies they found. They can quote the hour to hour SEVERI satellite showing the complicated patterns of the dust.*

A1) We have rewritten the paragraph discussing the complicated nature of the dust transport towards the EM and state so in the conclusions as well. In addition, we have added a comparison with the Limassol, Cyprus lidar following a suggestion by SC2 Q8 A8 N8 and cross-sections along the transport direction following RC1 Q12 A12 N12.

[revised manuscript text omitted]

*Q2) It is suggested to add the Sede boker Aeronet AOT finding and relate it to the Model AOD and add these results to Fig 8 in 3.4.1*

A2) We have added the Sede Boker AERONET AOD to the figure. As it can be expected, this shows the same problems as are visible in the PM10 measurements already, with an underestimation of dust by a factor of $4$. On $07$ September the course of the AERONET AOD is replicated by ICON-ART, however with an offset of 0.3. This can be explained by other aerosols besides mineral dust (sea salt, carbon) which are measured by AERONET but not included in our ICON-ART simulation for this study. As a consequence of showing Sede Boker and your justified criticism of our Ashdod discussion, we also changed the PM10 measurements from Ashdod to Beer-Sheva, as this is in closer proximity to Sede Boker and therefore comparable, please see Q4+A4.

N2) Please see N4 for changes.

*Q3) It is suggested that the PM10 graph will be given in a logarithmic scale. This will show that part of the dust will be found on the 7th too.*

A3) We are now giving the graph with a logarithmic scale. As you expected, the development of the dust concentrations is now observable for $07$ September already. We have extended the discussion to contain this behaviour. The features discussed in the text remain the same and we have adjusted the discussion to create a less "black-and-white description", see next question.

N3) Please see N4 for changes.

*Q4) The authors tend to explain Ashdod low measurements (fig 8) as a result of sea breeze penetration that brings in clean air. As can be seen from the measurements of the Tera and Aqua and the PM10 measurements of other sea stations the sea was full of dust. There were numerous differences between station to station in Israel even in short ranges a few km apart. So taking only 3 stations for verification cannot reveal the whole picture in such ac complicated event though they seem to fit with the model logic. Therefore, we suggest that fig 8 will be given as a description to the event complexity without a black-and-white description.*

A4) We agree that the dust distribution on 08 September was complex in the southern EM and that it cannot be adequately captured by a comparison with three stations alone. However, by displaying the chosen stations we tried to describe features which were not only visible in these stations but also the surrounding ones with similar characteristics (Northern Israel low land, Jerusalem mountain ridge, coastal which is now switched to arid). As a side-effect of showing Sede Boker AERONET values we now also display Beer-Sheva PM10 measurements. Your comments on the Ashdod measurements are correct, and we have retracted our statements. Of course, a more sophisticated comparison and statistical analysis is needed in the future, but in order to do so the dust emission and transport characteristics into the region need to be improved first. We believe, that in this context more research regarding dust processes is needed in the region

(see also our new concluding remarks, RC2 C6). We now mention the complexity of the dust distribution and that the comparison is not complete using three stations alone as you suggested. Also, we tried to reduce the black-and-white description by wording our statements more carefully.

N4) Sec. 3.4.1:

[revised manuscript text omitted]

30   September and 11.18 UTC 08 September 2015 (NASA Worldview, 2016).

C) Please note the additional changes which we have included in the paper as part of our improvements, most of them concern the English phrasing of sentences, figure labelling or units. More substantial ones are listed below.

C1) We changed the title to be:

 Revealing the meteorological drivers of the September 2015 severe dust event in the Eastern Mediterranean

35   C2) The dust sources are located in northeastern Africa, not northwestern Africa as stated before:

As a result, the most important remote dust source regions for the EM are situated in  northeastern Africa and the southern Arabian peninsula (Ganor, 1991; Kubilay et al., 2000).

C3) We retracted a statement in the introduction, as it is valid above oceans which is not the case in our study:

C4) We have added a sentence on the need for further research on the radiative effects in CPOs in Sec. 3.5:

A more intense and faster spreading CPO can have multiple reasons and further research is necessary in order to quantify the different contributions

C5) Added statement in the conclusions referring to questions posted in the beginning, in addition changed question heading to be non-bold:

Summarizing, we are able to answer the research questions presented in the beginning as follows

C6) We have added a statement on the outlook and overarching implications of this study to the conclusions:

In conclusion, this comprehensive case study has demonstrated the need to explicitly represent deep moist convection in dust storm forecasting. While Pantillon et al. (2016) propose a simple parametrisation to represent the climatological effects of haboobs in coarser resolution models, the forecasting of severe events like the one investigated here can hardly be successful without explicit convection. Given the substantial impact of the event and the potential benefit of an early warning, forecasting centres around the world should consider running higher-resolution dust forecasts for the most vulnerable regions. More research is also needed into the multi-scale interactions between Red Sea troughs, heat lows and convection. Moreover, the role of hydraulic jumps for dust emission and transport in the Dead Sea valley appears an interesting subject for further study, which would greatly benefit from a denser observational network.

---

## Referee Report (RR1)

**Review of the manuscript entitled: "Revealing the meteorological drivers of the September 2015 severe dust event in the Eastern Mediterranean" by Gasch et al.**

This review (second round) has two parts. I (Albert Ansmann) will focus on the response of the authors to my request to provide a proper Introduction (Sect.1) into the field of research regarding this extreme (record breaking) dust event, i.e. to provide a proper overview on the published work concerning the September 2015 dust event. I asked my colleague, the modeling expert Stavros Solomos from the National Observatory Athens to critically review the revised version. His comments are given in Part 2.

Part 1 (Albert Ansmann):

I appreciate very much that the authors did a lot to improve the first version of the manuscript. The paper will become an excellent contribution to the atmospheric science literature. Nevertheless, I am not satisfied with the Introduction. The authors did not fully follow my suggestions (Q2 in their reply letter). The introduction must therefore be further improved to my opinion.

The main focus of the Gasch et al. paper is modeling of the September 2015 super dust event. The title of the paper clearly indicates that. An overview of the published literature regarding observations and modeling efforts of this super dust event and the main findings described in these papers must be given. In the case of observations this is done. Only one peer reviewed paper is available here (Mamouri et al., 2016). The rest is more or less grey literature. Alpert et al. (2016) is an EGU abstract.

In the case of modeling efforts, the requested review is not given. Thus, the Introduction must be improved here significantly. There is the paper of Solomos et al. (2017) on this super dust storm, which goes into deep details. The findings of this paper must be summarized in the Introduction and afterwards the motivation for the Gasch et al. paper should be outlined in view of the already known facts and analysis results of Solomos et al. (2017).

The main (exiting) questions regarding this dust storm were: Why did the forecasts more or less failed to predict this dust storm? What were the reasons for the development of the enormous dust storm? And these answers can already be found in the Solomos et al. paper. The Gasch et al. paper is a follow-up paper of the Solomos paper. This is very clear! But this not a 'draw back'. By saying what is already known it becomes very easy to say what will be the new contributions of Gasch et al.? What is

missing, what are the new points. And there are many, as the paper nicely shows. This is the well-accepted and logical way of Introductions. However, the authors failed to follow my opinion. The Introduction is not acceptable.

Because of the low resonance to my recommendation (Q2) I have the feeling I should expand my own ideas what the authors should present:

Let me start with the following (defintions): There is a nice paper (Gkikas et al., ACP, 16, 2016, should show up in the references) presenting an extended climatological overview on dust storms in the Mediterranean and they defined: a) STRONG dust storm… when the 500 nm AOD exceeds the climatological mean AOD for given site by 2 STD and is between 2-4 STD, b) EXTREME dust outbreak .., when the AOD exceeds 4 STD. And this September 2015 dust storm exceeded the STD at Limassol by a factor of more than 25 (as written in Mamouri et al., 2016) !!! This expresses best what happened in September 2015 in the Middle East.

Now to the most surprising question (motivating the Solomos paper and the Gasch paper): Why did the dust forecast models fail… in this specific case of this super dust storm?

Solomos et al. (2017) answered this question already in a number of points. They provide a list of reasons for the development of the dust event, based on complex modeling and in-depth comparison with spaceborne remote sensing.

Solomos et al. already showed why all these dust forecast models failed because 'convection permitting resolution' is required. Solomos et al. already discussed the reason(s) for the enormous dust emissions (development of MCS, of cold pool outflows, density currents occurrence, humid air transport from the south, even changes in the surface characteristics, probably a consequence of the long-lasting political crises in the Middle East).

In the conclusions of Soloms et al. the reasons for the dust storm were summarized:

1. the formation of a strong thermal low and of convective outflows over Syria that lifted dust up to 4 km,

2. the intrusion of moist and unstable air masses from the Arabian Sea and the Red Sea that triggered convective activity over the Iraq, Iran, and Syria (Turkey border),

3. the generated outflow boundaries that led to dust deflation and formed a westward-propagating haboob that merged with the previously elevated dust over Syria,

and 4. the increased efficiency of Middle East dust sources in the aftermath of war and the related changes in land use.

I was trying in my first review, to convince you to present exactly this. What is available in case of this super storm (observations, modeling)? What are the facts...?

And based on this information, then your paper comes into play. What are the new points presented in your paper? What is still open. What do you want to demonstrate? And then you can introduce ICON-ART and the incredible potential of this modeling infrastructure, linked to this fantastic and unique ART module.  So, your paper will become a rather valuable  addition to the scientifc literature, no doubt. And this high level paper should have a high-level Introduction as well.

Let me say at the end (because I had this impression when reading the current version of the Introduction)... Science is not an Olympic racing competition (who is the best?). Science is a careful, sensitive, steady and slow accumulation of new knowledge, step by step, paper by paper, ..., and a good Introduction reflects that, provides a review of these papers before coming up with the own new idea and contribution.

Part 2 (Stavros Solomos and Albert Ansmann)

The submitted manuscript presents an ICON-ART simulation of the record breaking dust event of September 2015 over Middle East and the Eastern Mediterranean. The authors use an extended convection-resolving domain to analyze this episode and in our opinion their major findings (i.e. transport of moisture from the Red Sea, formation of a strong thermal low in Syria, three consecutive and eventually merging cold pool outflows) support the results presented in previous studies of the same event by Mamouri et al., 2016 and Solomos et al., 2017.

Their analysis is valid and provides for the first time the ability to examine convective processes over such an extended domain. However we feel that the authors do not fully exploit their model results and that they should provide more insight and quantification of the convective cloud processes, the vertical structure of the developing storms and the dusty outflows. In general we recommend that this manuscript should be accepted for publication in ACP with major revisions. Specific comments follow below:

Specific Comments

**P2. Line 35 – P3. Lines 1-3: "Solomos et al. (2017) model the event at convection permitting resolution, however the spatial extent of their convection permitting domain does not cover the full MCS region. Consequently, their model fails to reproduce the observed CPO outflow structures and connected dust plumes realistically (see discussion in Sec. 3.2 and 3.3)."**

As stated in Solomos et al., 2017:"The key to forecasting these events in atmospheric models is the use of cloud-resolving grid space. However, such high-resolution grid space can only be applied over limited areas due to restrictions in computational power. Forthcoming studies using an extended cloud-resolving grid over the entire Middle East (e.g. Gasch et al., 2017) could provide more detail on the individual atmospheric processes during this episode".

However, the analysis and the general findings presented here are very similar to the ones shown in Solomos et al, 2017. It is somehow contradicting for the authors to say that this previous work lacks crucial information since their major findings are basically the same. ICON-ART may be performing better at some places but by no means RAMS "failed" to reproduce the event. Furthermore ICON-ART also clearly misses several event processes with deviations of up to one order of magnitude. This paragraph should be rephrased.

**P3. Lines 20-21: "It is capable of local grid refinements, in this study the finest nest has a convection permitting grid spacing of 2.5 km."**
**P8. Lines 13-15: "A grid spacing of 2.5 km is generally assumed to be sufficient to permit the development of convection in a non-hydrostatic model. Therefore the convection parametrization, including the parametrization for shallow convection, is switched off for the finest grid."**

Whether a 2.5×2.5 km grid space is adequate for resolving convection is somehow questionable. Especially during the first crucial stages of convective development even this resolution may not be sufficient for accurately representing the initial cloud dynamics and microphysics. The authors should support this argument with literature references.

**P9. Lines 9-10: "The aerosol - cloud microphysics interaction is not included in this study as it creates a new set of research questions and the focus in this study is on the mineral dust radiation interaction."**

Based on previous studies, the effects of dust-cloud microphysics interactions in the formation of similar systems is probably very limited compared to the dynamics of the system (see for example Solomos et al., 2012).

**P9. Lines 16-18: ”The IFS initialization data for soil moisture was modified in a region along the Syrian-Iraqi border which showed high soil moisture values and spatial inhomogeneities without preceding rain or changes in soil properties.”**
Is there any observational evidence to support this change in initial model soil moisture? What would be the difference in modeled dust fields if the authors used the original IFS fields?

**P12. Lines 19-20: “We focus on the results from the convection permitting domain, as it yields remarkable improvements compared to the global domain.”**
How do the intermediate domain results compare to the fine-grid results? The benefits of using a convection permitting grid space (though reasonable) are not properly justified. The authors should compare their high resolution runs with ICON-ART results using for example only 3 nested domains (e.g. 40-20-10-5 km) and discuss the possible improvements.

**P13. Lines 8-11: “Our analysis contrasts the simulation results by Solomos et al. (2017, their Fig.4c), who model AOD values above 20 already before the onset of strong downward mixing of momentum. Furthermore, their modelled bimodal maximum dust distribution was not observed by satellites and no closed cyclonic flow around the heat flow appears to have existed.”**
In Figure A1 (a) and also in SEVIRI images it looks that dust flow is actually following a cyclonic circulation over Syria. The extreme AOD values in RAMS simulations shown in Solomos et al., 2017 appear over a few grid points mostly due to the overlapping of multiple dust layers. These may indeed be unrealistic but the event itself was extreme and furthermore there is no reliable satellite retrieval at such high AODs. Atmospheric models are far from being perfect and as shown in the current manuscript even ICON-ART fails to reproduce the transport of dust at certain areas or presents an underestimation of one order of magnitude compared to dust measurements.

**P13. Lines 18-19: “The reinitialization of ICON-ART with IFS at 12 UTC impairs the CPO2 development due to the termination of convective structures.”**
The authors should justify why it is needed to reinitialize ICON-ART after 18 hours run. What would be the model results if the authors let this simulation continue? Initialization with the IFS at this stage actually means that ICON-ART meteorological

fields are overwritten by the IFS analysis that already assimilates all available observations of the MSC organization.

**P14. Lines 5-8: "The above findings again contradict those of Solomos et al. (2017, their Fig. 7b), who in their model results find a northward travel direction of a small cold-pool structure. Based on the good agreement between ICON-ART and satellite observations this result is implausible. Furthermore, the intensity and spatial extent of their modelled CPO is much too small.**

This is not true and the authors should remove this comment. As explained in Solomos et al., 2017, Figure 7 shows a primary convective cell that travels towards the north and triggers the generation of the larger cold pools which are shown in Figure 8 and indeed travel westward / south-westward. This activity is also evident in SEVIRI images.

**P14. Lines16-27: "Past midnight on 07 September and explosive intensification…….**
**and a greater dust plume depth which subsequently extends throughout the full CPO"**
In our opinion, this section (accompanied by the A4, A5 cross-section plots) shows a clear picture of the developing situation and moreover exhibits the benefits of using such an extended convection resolving domain. However it is not clear what is shown in each of these plots, the captions do not include information about the plotted quantities and the quality of the images should be improved (if possible adding also an indication of the cross-section locations over a SEVIRI horizontal plot). The horizontal evolution of the episode is more or less evident from SEVIRI but the vertical analysis can only be obtained through modeling simulations as shown here. Improving this information and also adding these plots in the main manuscript will improve the analysis and the overall understanding of the event.

**P15. Lines 1-2: "Downstream the line-shaped rainfall distribution near surface wind speeds increase strongly as the CPO3 reaches the surface (Fig. 5b). Wind speeds above 12 m s$^{-1}$ are modelled inside the CPO3 region."**
The formation of a gusty density current in front of the rain curtain must be further analyzed with a vertical plot of the microphysical and dynamical structure of the storm evolution at this stage. This is the crucial stage for the generation of the haboob and more information must be provided on the contribution of the various cloud properties to the strength of the generated CPO.

**P15, Line8: "The maximum DOD value is 4.15, it is reached in an area close to the leading edge of CPO3 which shows the highest values of DOD."**

The authors should extend their contour scale up to 5 in Figure 5. The same applies to Figure 6 and other AOD images in the manuscript where the maximum values are clearly above 2.

***P15, Line 30: "Unfortunately, no measurements by MODIS are available over the northern Mediterranean Sea"***
May be "northeastern parts of Mediterranean Sea" is more appropriate.

***P15, Line 32: "For the eastern part of CPO3 the MODIS AOD measurements seem doubtful when comparing to the MODIS visible satellite image."***
In our opinion the MODIS AOD retrievals do not fully represent the severity of the event. The extreme PM measurements in Cyprus and Israel (away from the sources) and the total attenuation of CALIPSO backscatter imply much larger values of AOD close to the areas of intense dust activity.

***P18, Lines 8-9: "The vertical structure of the dust plume can be investigated at this point in time with the help of a CALIPSO overpass which occurred at 10:35 UTC."***
The qualitative comparison presented here does not provide valuable information on the unique characteristics of this event. We can offer to the authors our quantitative CALIPSO analysis from Solomos et al., 2017 so as to compare ICON-ART results and conclude on the validity of their simulation.

***P18, Lines 15-22: The satellite passes the region where the merged one and two day old sea-breezes ….. CALIOP reports high values of attenuated backscatter in this region whereas ICON-ART simulates a minimum due to the clean air characteristics of the sea breeze"***
The explanation given by the authors on the discrepancy between ICON-ART and CALIPSO profiles suggest the mobilization of dust particles by Mediterranean sea-breeze winds penetrating somewhat 800 km inland the Arabian Peninsula. Are these winds strong enough to mobilize dust? This is something hard to believe by the reader and the authors must further support their argument with data.

***P19, Line 3-6: "Sec. 3.3 When comparing our results to those of Solomos et al. (2017, their Fig. 10),again large differences become apparent, both in spatial dust plume structure as well as vertical dust distribution. The driving MCS and related CPOs and their clearly marked borders discussed above are not identifiable in their model results and they do not provide a detailed analysis of the flow structures."***
Yes the two models differ. Regarding the vertical dust structure the authors should try to compare the same quantities in Figure 7 so they can reach to some quantitative

results for their simulations. We can offer our CALIPSO analysis so as to compare ICON-ART results and conclude on the validity of their simulation.

***P.20, Lines 4-6: "The high surface wind speeds and turbulent mixing inside the CPO3s result in enormous dust emissions during daytime, consequently the dust is transported within the full boundary layer height up to 5 km (Fig. A4, A5).***
How enormous? The authors should give a range of the modeled dust concentrations inside the CPOs.

***P.20, Lines 14-15: "ICON-ART DOD values are one order of magnitude higher and show better spatial agreement than other global dust forecast simulation results in the northern part of the EM (see Sec. 1)."***
This is not exactly true. If you take a look for 8 September 2015 12:00 UTC in the SDS-WAS portal (https://sds-was.aemet.es/forecast-products/dust-forecasts/forecast-comparison) several models indicate AOD values up to 1.2 over the area of interest.

***P.20, Lines 15-21: "Taking into account the 2° longitudinal offset in ICON-ART, the vertical structure of the dust plume arrival represents observations by Mamouri et al. (2016) (Fig. A7) ….. During 08 September, dust concentrations increase up to 3500 g /m3 and the plume thickness grows further, extending from 0.5 km up to 3 km height in the model."***
Comparing a Cyprus station with a model grid point 200 km towards the Mediterranean Sea does not make a lot of sense for aerosol studies. On 8 September the lidar was not operating so there is no point to include the comparison shown in the right plot of A7 Figure. It is not clear what the authors mean by : "A first, elevated plume extending from 2-4 km with concentrations up to 1000 µg / m3 is noticeable during 07 September in both lidar measurements and model results" and how this is supported by Figure A7. Also the authors state that: "During 08 September, dust concentrations increase up to 3500 µg /m3 and the plume thickness grows further, extending from 0.5 km up to 3 km height in the model". But the near-surface concentrations in Figure A7 are below 250 µg/m3 on this day while the surface stations in Cyprus recorded concentrations up to 10000 µg/m3 and AOD values greater than 5 indicating a model underestimation of several orders of magnitude. In general we recommend that the authors should remove this entire paragraph since it does not contribute to the explanation of the episode.

***P21. Lines 18-20: "The maximum modelled DOD for Sede Boker is 1.0 on 08 September, compared to 4.1 measured by AERONET. The AERONET values appear***

*realistic, as they are in good agreement with MODIS AOD measurements in the region. Thus, ICON-ART shows an underestimation of DOD by a factor of four."*

If the measured AOD is 4.1 in Israel, then the authors should reconsider their much lower DOD values at the areas of intense haboob activity. Also, nobody ever validated ground-based data with satellite observations. The authors should rephrase or remove this sentence.

*P21. Lines 21-22: "When comparing PM10 measurements, a larger difference between model results and observations on the order of one magnitude becomes apparent."*

To our understanding, this large underestimation implies that near the sources the event was much stronger than simulated by ICON-ART and even MODIS AOD retrievals are not able to capture its severity in terms of optical thickness. It looks like some important meteorological or surface mechanism is still missing in the proposed analysis.

*P23. Line 2: "The underestimation by a factor of four between model and measurement is consistent with the AERONET measurements in Sede Boker."*

This sentence does not make sence.

*P23. Line 29: "leading to a compressions"*

Leading to compression

*Figure 9, Caption.*

The authors should number properly the panel plots as a,b,c,d and explain what is shown there.

*Figure 11.*

The authors should provide statistical metrics for their comparisons.

*P. 27, Lines 19-20, 33-35 "As a result, possible mineral dust emissions due to the super-critical flow cannot be captured by ICON-ART."*

*"In combination with already underestimated dust emissions due to the recent land cover changes and soil degradation in the Mesopotamia region (see Sec. 2 and 3.3), this provides an explanation why dust transport into the southern EM is underestimated by an order of magnitude. Nevertheless, ICON-ART provides a detailed understanding of previously unknown processes contributing to the historic dust event which makes these findings worthwhile to report."*

We agree with the authors that the detailed description of this flow feature is valid and adds to the overall description of this record episode. The hydraulic jump shown in

Figure 10 occurs due to the steep topographic gradient between Golan Heights and the Sea of Galilee. However as the authors also mention, the modeled dust values were already too low before the system reached this area and in our opinion the extreme PM10 measurements in Israel cannot be explained by local production.

***P28. Lines 4-5:"The validation of the mineral dust distribution and transport characteristics with satellite and station measurements show overall good agreement between ICON-ART and the observations, especially during the early stages of the event."***
Quantitative comparisons in previous sections indicate a severe underestimation of modeled dust and obviously the quantification of dust concentrations is crucial for radiative transfer calculations. Nevertheless we believe that this section (3.5) is still useful but we recommend that the authors should examine the vertical CPO structure at various stages of development (stratification of dust layers, distribution of TKE and speed inside the haboobs) with and without the interactive dust radiative transfer. Investigation of such vertical cross-sections could really provide some insight on the impact of radiative active dust in CPO dynamics.

***P.32. Line 11:" This study presents a successful simulation of the September 2015 severe dust event at convection permitting resolution."***
The simulation clearly underestimates the dust concentrations and there is no quantitative comparison for the vertical dust concentrations. There is really no need to claim a simulation as "successful". Instead we would propose that the authors rephrase this sentence to something like: "This study presents a simulation of the September 2015 severe dust event at convection permitting resolution that allows the reproduction of the main atmospheric processes during in this event."

***P.32. Lines 13-15: "the inland penetrating Eastern Mediterranean sea-breeze and the widespread occurrence of super-critical flow conditions and subsequent hydraulic jumps are suggested as important drivers for dust emission."***
This is not supported by the presented analysis. Even if these processes exist their effect is minimum compared to the huge convective outflows.

***P32. Lines 21-22: "(1) Is the forecast of the dust event improved by running convection permitting simulations? The convection permitting simulation of the dust event with ICON-ART improves the forecast quality decisively."***
This is not supported by the presented analysis. In order to prove this argument the authors should compare their results with ICON-ART results at lower resolution. For example it is not justified if a 4 grid configuration with parameterized convection at 40-

20-10-5 km cannot reproduce the event. A comparison between these two runs could be easy to do and will show how "decisively" is the forecast quality improved.

**P30. Lines 24-25: "As the energy reaches the surface nevertheless, reductions in surface temperature due to mineral dust are smaller than those found by two other studies."**
Which other studies? Are these studies comparable to an extreme haboob event?

**P32. Lines 27-28:"A comparison with lidar observations in Cyprus (Mamouri et al., 2016) shows very good agreement in vertical dust distribution."**
This is not supported by the presented analysis and a comparison of a Cyprus station with a model grid point 2° east has no meaning for aerosol studies. Also on 8 September the lidar was not operating so the right plot in Figure A7 is not needed.

**P32. Line 30: "and without data assimilation"**
Actually when the authors reinitialize their model with IFS at 12 UTC on 6 September, they practically assimilate all available observations of the already evolving event in their simulation. From an operational forecasting point of view, the real value would be to show the results of ICON-ART from the previous day run.

**P33. Lines 1-2, 5: "For the transport to the southern EM, a hydraulic jump is demonstrated to be of importance for dust emission in addition to the advection of the dense dust plumes into the region"**
**"Modelled DODs are in the range of 0.5-1.5 over Israel and PM10 concentrations reach up to 600 μg/ m3 in Jerusalem."**
The dust concentrations are already underestimated before the plumes approach Israel. An increase from 600 to 6000 μg/m3 cannot be explained by local emissions.

*References:*

Mamouri, R.-E., Ansmann, A., Nisantzi, A., Solomos, S., Kallos, G., and Hadjimitsis, D. G.: Extreme dust storm over the eastern Mediterranean in September 2015: satellite, lidar, and surface observations in the Cyprus region, Atmos. Chem. Phys., 16, 13711-13724, https://doi.org/10.5194/acp-16-13711-2016, 2016.

Solomos, S., Kallos, G., Mavromatidis, E., and Kushta, J.: Density currents as a desert dust mobilization mechanism, Atmos. Chem. Phys., 12, 11199–11211, doi:10.5194/acp-12-11199-2012, 2012.

Solomos, S., Ansmann, A., Mamouri, R.-E., Binietoglou, I., Patlakas, P., Marinou, E., and Amiridis, V.: Remote sensing and modelling analysis of the extreme dust storm hitting the Middle East and eastern Mediterranean in September 2015, Atmos. Chem. Phys., 17, 4063-4079, https://doi.org/10.5194/acp-17-4063-2017, 2017.